# Divergent community assembly processes and multifunctionality contributions of abundant and rare soil bacteria during a 53-year restoration in the Tengger Desert, China
Qingqing Hou [1,4], Rui Xia[1,4], Bodong Yuan[1], Muhammad Aqeel[1], Ying Sun[1], Longwei Dong[1], Abdul Manan[1], Fan Li[1], Yan Deng[1], Xusheng Guo[2], Guili Wu[1], Jinzhi Ran[1], Weigang Hu [1] ✉, Jihua Wu[1], Xinrong Li[3] & Jianming Deng [1] ✉

Soil microbial communities play vital roles in driving ecosystem restoration. However, understanding of the successional dynamics of abundant and rare bacterial subcommunities and their relationships with ecosystem multifunctionality during restoration, particularly in desertified ecosystems, remains limited. Here, we examined the succession of abundant, intermediate, and rare bacterial subcommunities over a 53-year restoration chronosequence following the implementation of straw checkerboard barriers in the Tengger Desert, China. Our findings revealed that the establishment of straw checkerboard barriers significantly increased the richness of abundant, intermediate, and rare taxa over time. However, our results indicated a divergence in ecological processes underpinning the successional dynamics of soil bacterial communities. Stochastic processes and homogeneous selection primarily governed the assembly of abundant and rare subcommunities, respectively, as evidenced by fundamental differences in their niche breadth. More importantly, we further uncovered a dual mechanism underlying the relationships between soil bacterial communities and ecosystem multifunctionality. Abundant taxa were integrally associated with multiple nutrient cycling-related functions simultaneously, likely mediated through coordinated environmental responses or potential interspecies connections, whereas rare taxa were more linked to individual functions independently. These findings deepen our understanding of the successional dynamics of soil microbial communities and the microbe-ecosystem multifunctionality relationships in desert restoration.

Drylands, home to more than 38% of the global population, provide a wide array of ecosystem services[1,2]. However, these regions are increasingly vulnerable to desertification due to increased aridity driven by climate change and anthropogenic pressures, jeopardizing global ecological security and sustainable socio-economic development[1,3,4]. In response, ecological restoration efforts have been widely implemented globally, effectively enhancing biodiversity as well as ecosystem functions and services[5–7]. Over the past seven decades, large-scale artificial restoration initiatives in China, including the Three-North Shelterbelt Development Program and the Grain for Green Program, have significantly contributed to the regreening of drylands[8–10]. Among the ecological restoration strategies widely employed in these projects, straw checkerboard barriers (SCBs) stand out as one of the

[1]State Key Laboratory of Herbage Improvement and Grassland Agro-ecosystems, College of Ecology, Lanzhou University, Lanzhou, China. [2]School of Life Sciences, Lanzhou University, Lanzhou, China. [3]Shapotou Desert Research and Experiment Station, Northwest Institute of Eco-Environment and Resources, Chinese Academy of Sciences, Lanzhou, China. [4]These authors contributed equally: Qingqing Hou, Rui Xia. ✉e-mail: huweigang@lzu.edu.cn; dengjm@lzu.edu.cn

most cost-effective and demonstrably highly effective in stabilizing shifting sand and curbing wind erosion[11,12].

Soil microbes, as highly abundant and diverse organisms, participate in multiple ecological processes related to biogeochemical cycles and are crucial drivers for maintaining multiple ecosystem functions and services simultaneously (i.e., ecosystem multifunctionality)[13–16]. Existing studies have confirmed that changes in soil microbial diversity and community structure significantly contribute to the restoration of degraded ecosystems[17–19]. In natural ecosystems, the distribution of species in microbial communities is commonly unbalanced, characterized by a small number of abundant taxa alongside numerous rare taxa[20,21]. Research has shown distinct spatial distribution patterns between abundant and rare taxa[22–24]. However, the temporal variations of abundant and rare subcommunities during long-term restoration efforts remain inadequately understood, limiting our comprehension of microbe-driven ecosystem processes and functions.

Abundant and rare taxa may play distinct roles in promoting ecosystem restoration owing to their divergent ecological and functional characteristics[25,26]. Abundant taxa can utilize a wide range of resources, and are highly active in driving biogeochemical cycles due to their broad niche occupancy and metabolic versatility[20,23,27]. As a result, they can significantly contribute to the rapid restoration of multiple functions in degraded ecosystems[28]. In contrast, rare taxa have specific habitat preferences and narrow niche breadth and limited competitive ability, possess diverse genetic and metabolic lineages[29]. This empowers rare taxa to promptly respond to environmental disturbance, however, their inherently high diversity may reinforce the resilience and long-term stability of ecosystem functioning through functional redundancy and complementary effects[30,31]. Thus, analyzing abundant and rare taxa separately offers critical insights into divergent restoration goals and serves as guidance for the development of targeted restoration strategies. Traditional studies have often focused on abundant taxa, using them as proxies for the entire microbial community[32,33]. However, recent studies have emphasized the importance of rare taxa in maintaining community structure and ecosystem functions across various ecosystems, such as alpine grasslands[34], acidic soils[35], and farmlands[36]. Moreover, rare taxa perform specific functions that facilitate the turnover of certain elements, such as sulfate reduction[37]. Nonetheless, the fundamental mechanisms and relative importance of abundant and rare taxa in driving the restoration of desertified ecosystems insufficiently elucidated.

Uncovering the underlying community assembly processes is essential for comprehending the maintenance of biodiversity and its relationships with ecosystem functions[38–40]. For instance, stochastic assembly processes promote the positive effects of the diversity of rare taxa on ecosystem multifunctionality in farmlands[25]. In general, different assembly processes shape abundant and rare subcommunities[41,42]. Previous studies have shown that stochastic processes have a greater impact on structuring abundant subcommunities, whereas rare taxa are primarily mediated by deterministic processes in reforestation[43], farmland[23] and dryland ecosystems[22]. However, other studies have reported contrasting findings[27,44,45]. Together, these results suggest that the assembly processes of abundant and rare subcommunities may be contingent upon the specific ecosystem types. Furthermore, there is increasing evidence that ecological restoration can drive a time-dependent shift in the balance between stochastic and deterministic processes in microbial succession[39,46,47]. For instance, previous studies have indicated that within successional stages microbial communities are initially governed by stochastic processes, but the importance of deterministic processes intensifies as succession proceeded[39]. However, the relative importance of stochastic and deterministic processes in structuring abundant and rare subcommunities and their temporal variation during the long-term restoration of desertified ecosystems remain largely untested.

In this study, we measured soil bacterial communities using amplicon sequencing of the 16S rRNA gene from 55 soil samples collected along a 53-year restoration chronosequence containing 11 stages following the establishment of SCBs on the southern fringe of the Tengger Desert in China (Supplementary Fig. 1). We defined abundant, intermediate and rare taxa based on the average relative abundance of individual taxa across the entire dataset using multivariate cutoff level analysis (MultiCoLA; Supplementary Figs. 2 and 3)[48]. This study sought to address three important issues: (i) How do the richness and composition of abundant, intermediate, and rare soil bacterial subcommunities respond to the establishment of SCBs over time? (ii) What are the relative contributions of stochastic and deterministic processes in shaping abundant, intermediate and rare subcommunities, and how do they shift over time during restoration? (iii) Do abundant and rare taxa play distinct roles in maintaining ecosystem multifunctionality? To address these issues, we examined the temporal diversity and composition patterns of abundant, intermediate, and rare subcommunities. A null-modeling-based quantitative framework was used to discern the relative importance of the ecological processes structuring the three subcommunities. We further measured 17 ecosystem functions related to productivity, decomposition rate, and soil nutrient pools, which are representative of the storage and cycling of matter and energy. Subsequently, the relationships of the three subcommunities with individual ecosystem functions and multifunctionality were assessed at both the community and species levels.

## Results

### Temporal dynamics in diversity and composition of soil bacterial subcommunities

The rarefaction curves for most of our samples reached a steady plateau, signifying that the majority of bacterial amplicon sequence variants (ASVs) were successfully recovered, providing comprehensive coverage of the bacterial richness in our dataset (Supplementary Fig. 4). A total of 601 ASVs (2.40%) were classified as abundant taxa, accounting for 55.54% of the total sequences, and 4507 ASVs (17.97%) were classified as intermediate taxa, accounting for 33.59% of the total sequences (Fig. 1b, c). In contrast, although a larger proportion of ASVs (19,969, 79.63%) were identified as rare taxa, their relative abundance accounted for only 10.87% of the total sequences (Fig. 1b, c). Based on the sample occupancy rate for each ASV, 53.24% of the abundant taxa were found in > 50% of the samples, whereas none of the rare taxa appeared in > 35% of the samples, and 70.92% of the rare taxa appeared in only one sample (Fig. 1d). At the phylum level, the rare subcommunities (56 phyla) encompassed more taxonomic groups than the abundant (18 phyla) and intermediate (30 phyla) subcommunities (Fig. 1e). Actinobacteria, Proteobacteria, and Chloroflexi were the dominant phyla across all three biological groups. Both the Levins' index and the tolerance index were significantly higher for abundant and intermediate ASVs compared to rare ASVs (both $P = 2.2e{-}16$; Fig. 1f, g).

ASV richness of both abundant and intermediate taxa exhibited marked increases with restoration duration, reaching asymptotic stability after approximately 15 years (both $P = 1.0e{-}6$; Fig. 2a, b; Supplementary Table 1), whereas rare taxa richness showed a sustained, linear increase over time ($P = 0.002$; Fig. 2c; Supplementary Table 1). Although rare taxa included more ASVs across all samples, intermediate ASVs contributed greater richness at sample level ($P = 2.2e{-}16$; Supplementary Fig. 5a). NMDS ordination and ANOSIM based on Bray–Curtis distance revealed distinct differences in the composition of abundant, intermediate and rare subcommunities among samples with varying restoration durations (all $P = 0.001$; Fig. 2d–f). SIMPER analysis indicated that the cumulative contributions of abundant and intermediate subcommunities to the composition difference of the whole community were significantly higher than those of rare subcommunities ($P = 2.2e{-}16$; Supplementary Fig. 5b; Supplementary Table 2).

Moreover, the composition differences of abundant, intermediate, and rare subcommunities increased significantly with longer restoration duration interval (all $P = 0.001$; Fig. 3a–c). Compared to rare subcommunities, the composition of abundant subcommunities exhibited lower temporal variability (Fig. 3d–f). The β-diversity partitioning further revealed that species replacement (turnover), rather than richness difference, contributed for the majority of community composition variation (Fig. 3d–f), as shown by the heatmaps of the relative abundance of ASVs (Supplementary Fig. 6).

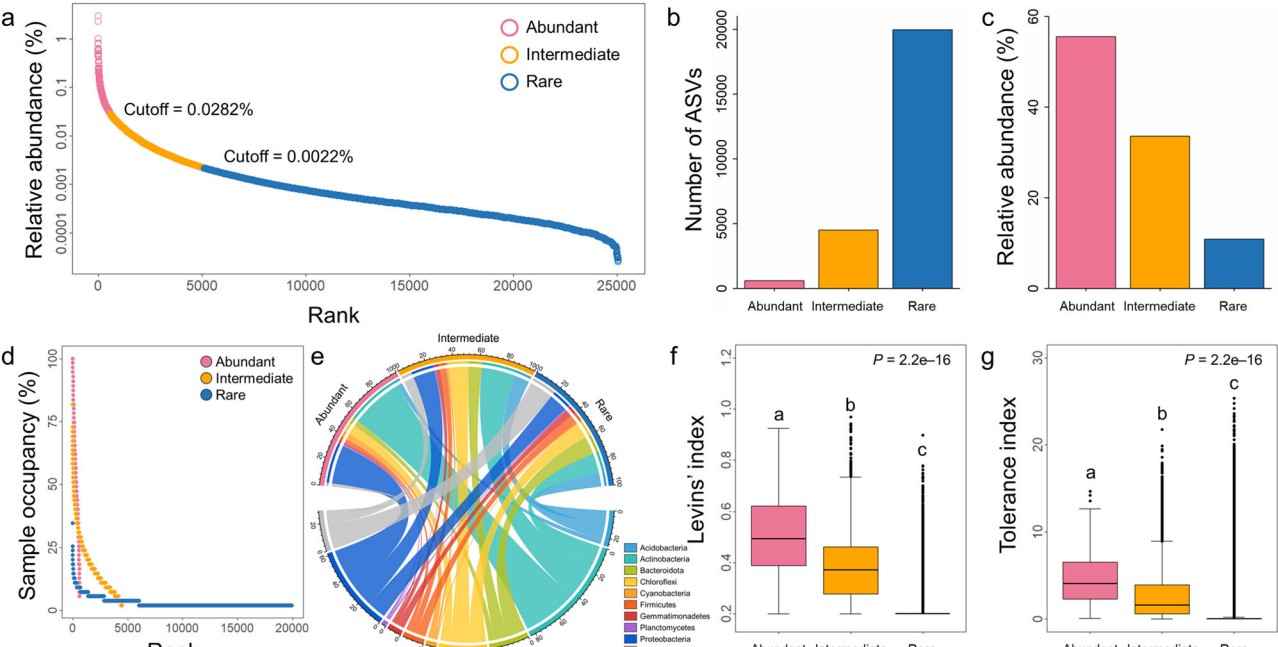

**Fig. 1 | Definition and composition of abundant, intermediate, and rare taxa.**
**a** Cutoff values for defining abundant, intermediate, and rare taxa based on the average relative abundance across all samples. **b, c** Number of ASVs and relative abundance for the three biological groups across all samples. **d** The proportion of samples occupied for each of abundant, intermediate, and rare ASVs. **e** Taxonomic distribution of the three biological groups at phylum level. The thickness of each ribbon in the circos plot represents the relative abundance of the three biological groups assigned to different phyla. **f, g** Boxplots of the niche breadth for each ASV within abundant ($n = 601$), intermediate ($n = 4507$), and rare taxa ($n = 19,969$) estimated using the Levins' index and the tolerance index, respectively. The different letters above the boxes represent significant differences determined by the Kruskal–Wallis test followed by Dunn post-hoc test (Bonferroni correction).

Furthermore, we evaluated the temporal variability of the relative abundance of each ASV and found that the substantial proportion (80.74%), predominately rare taxa, remained relatively stable along restoration duration (Supplementary Fig. 7). In contrast, most of the abundant ASVs (72.71%) exhibited either positive (34.28%) or negative (38.43%) responses in relative abundance under the influence of the SCB establishment.

## Community assembly processes and co-occurrence network of soil bacterial subcommunities

Leveraging the null-modeling-based quantitative framework, we elucidated that distinct ecological processes underpinning the assembly of abundant and rare subcommunities (Fig. 4). Specifically, the assembly of abundant subcommunities was mainly modulated by stochastic processes (69.3%), especially dispersal limitation (45.19%), while variable selection exerted a moderate deterministic influence (26.6%) on the abundant subcommunities (Fig. 4a). However, the assembly of intermediate (70.37%) and rare (73.53%) subcommunities was mainly governed by deterministic processes, with variable (43.43% for intermediate subcommunities) and homogenous selection (54.14% for rare subcommunities) being the main factors, respectively (Fig. 4b, c). On the other hand, the temporal variations in the assembly processes of abundant, intermediate and rare subcommunities were further delineated. Across all restoration stages, homogenizing dispersal and undominated processes had a greater influence on the temporal assembly of abundant subcommunities, while homogeneous selection dominated intermediate and rare subcommunities (Fig. 4d–f). Notably, homogenizing dispersal was more accentuated during the initial stages of restoration for the assembly of both abundant and intermediate subcommunities.

The co-occurrence network consisted of 742 nodes and 53,046 edges, in which 25.84% of abundant taxa nodes, 43.20% of intermediate taxa nodes and 30.96% of rare taxa nodes were detected (Fig. 5a). Within the network, 63.00% of pairwise correlations among bacterial ASVs were positive, indicating that their relative abundances generally changed synchronously with restoration duration, likely reflecting shared environmental preferences.

The topological properties of abundant and intermediate ASVs, including degree, closeness centrality, and betweenness centrality, were significantly higher than those of rare ASVs ($P = 2.2e{-}16$), while their clustering coefficient was notably lower than that of rare ASVs ($P = 6.0e{-}9$; Fig. 5b). A total of 142 ASVs were identified as keystone nodes in the network, comprising 3 module hubs (all abundant ASVs) and 139 connectors (with 1 abundant, 57 intermediate, and 81 rare ASVs) (Supplementary Fig. 8). By constructing the networks separately, we further found that abundant taxa exhibited higher network robustness compared to intermediate and rare taxa ($P = 2.2e{-}16$; Supplementary Fig. 9). In addition, we constructed correlation subnetworks at each restoration stage and found a progressive increase in network complexity over time, as evidenced by noticeable elevations in the degree and betweenness centrality of the nodes (Supplementary Figs. 10 and 11). Throughout all restoration stages, abundant and intermediate taxa exhibited higher degree, closeness centrality and betweenness centrality and lower clustering coefficient than rare taxa, which was similar to the results of the co-occurrence network across samples (Supplementary Fig. 11).

## Relationships between soil bacterial subcommunities and ecosystem multifunctionality

A significant and positive relationship between soil bacterial ASV richness and ecosystem multifunctionality was found for abundant ($P = 1.0e{-}10$) and intermediate taxa ($P = 3.7e{-}7$), but not for rare taxa ($P = 0.427$; Fig. 6a). Consistent results were observed for plant productivity, decomposition rate, and soil nutrient pool indices (Supplementary Fig. 12). The great majority of individual functions were significantly and positively associated with ASV richness of abundant and intermediate taxa, whereas only 4 functions were associated with rare taxa (Supplementary Fig. 13). Additionally, the average degree of abundant, intermediate, and rare taxa within subnetworks at each restoration stage showed significant positive correlations with ecosystem multifunctionality (Supplementary Fig. 14). To complement these bivariate correlation analyses, we employed a structural equation model (SEM) to infer the hypothesized direct and indirect pathways connecting restoration duration, plant cover, soil pH, bacterial ASV richness, average degree and

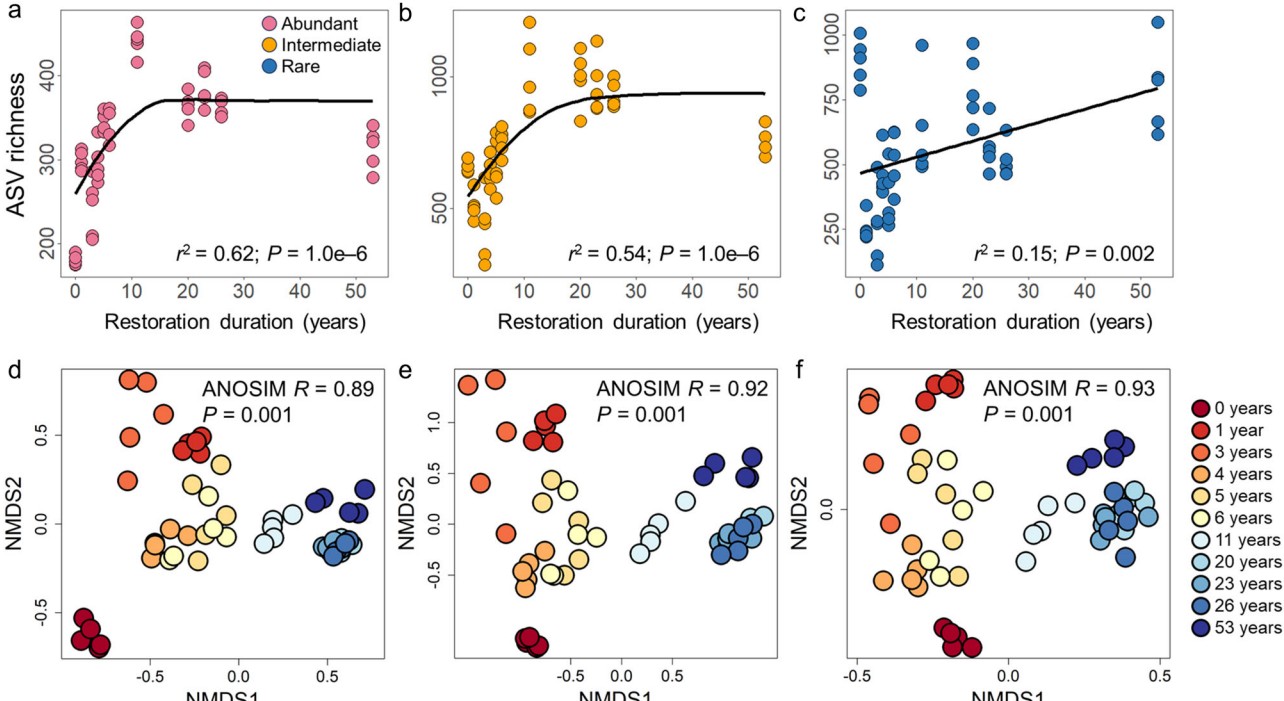

**Fig. 2 | Temporal changes in the richness and composition of abundant, intermediate, and rare subcommunities. a–c** Relationships between restoration duration and the richness of abundant, intermediate, and rare taxa. The black fitted lines are from linear and nonlinear regression. The solid and dotted lines represent statistically significant ($P \le 0.05$) and nonsignificant ($P > 0.05$) relationships, respectively. **d–f** NMDS ordination of abundant, intermediate, and rare subcommunities based on Bray–Curtis distance. ANOSIM was performed to test the differences in abundant, intermediate, and rare subcommunities among restoration durations.

multifunctionality (Fig. 6b). Our results revealed that restoration duration was positively associated with multifunctionality both directly and indirectly, mediated through enhanced plant cover and reduced soil pH. Increased plant cover significantly augmented bacterial ASV richness, particularly for abundant and intermediate taxa. Notably, bacterial ASV richness maintained a direct and positive association with multifunctionality ($P = 0.032$), while the relationship between average degree and multifunctionality was not statistically significant ($P = 0.688$). Moreover, the Mantel test showed that compositional changes in abundant and intermediate subcommunities were more strongly associated with multifunctionality than those in rare subcommunities (Supplementary Fig. 15).

At the ASV level, we further explored the relationships between the relative abundance of each ASV and individual ecosystem functions and multifunctionality indices, and found that the ASVs associated with functions and multifunctionality exhibited higher proportions and stronger correlations in abundant and intermediate subcommunities than in rare subcommunities (Supplementary Fig. 16). By examining the number of ASVs associated with varying numbers of functions, we observed the proportion of ASVs linked to multiple functions increased for abundant subcommunities ($P = 0.005$) but decreased for intermediate ($P = 0.020$) and rare subcommunities ($P = 3.3e-5$; Fig. 7). This indicated that abundant ASVs are more likely to be associated with multiple functions simultaneously, while rare ASVs are generally linked to individual functions. To further elucidate the potential mechanisms underpinning the differential contribution of distinct taxa to multifunctionality, we investigated the relationships between their co-occurrence network degree and magnitude of association with multifunctionality (Fig. 5c–e). Our results revealed that the abundant ASVs with higher degree manifested stronger correlations with multifunctionality ($P = 6.0e-6$), whereas inverse relationship was observed for rare ASVs ($P = 3.1e-5$).

## Discussion

Uncovering the succession patterns of high-abundance and rare taxa and their relationships with ecosystem functions during restoration is essential for comprehending the roles of soil microbial communities in rehabilitating the functions and services of desertified ecosystems. This study leveraged a 53-year restoration chronosequence of desertified ecosystems to demonstrate the similar diversity patterns but distinct assembly processes of abundant and rare bacterial subcommunities. More importantly, our results also unveiled marked divergences in the relationships of abundant and rare taxa with multifunctionality. Abundant taxa exhibited strong, simultaneous associations with multiple ecosystem functions, likely mediated through their co-occurrence patterns with other ASVs, whereas rare taxa were primarily linked with individual ecosystem functions independently.

One of the primary objectives of restoration efforts is to enhance the biodiversity of degraded ecosystems[5]. Our results indicated that the establishment of SCBs significantly increased the richness of abundant, intermediate and rare bacterial taxa. The increase can potentially be attributed to the improvement of vegetation and soil conditions, which provides more space and resources, thereby creating additional niches that support the survival and reproduction of more diverse species[49,50]. The majority of rare taxa possessed a high level of habitat specificity, and were only found in a single or limited number of samples, which may be attributed to their specific habitat preferences and narrow metabolic versatility[23,51]. Although the relative abundance of these rare ASVs showed negligible changes over time, their community composition displayed substantial variations, driven primarily by species-specific ecological niches across restoration stages (Fig. 3f)[22]. In contrast, a significant proportion of abundant ASVs, alongside certain intermediate ASVs, exhibited either positive or negative responses in relative abundance to restoration duration, revealing pronounced population fluctuations during restoration. Such dynamics of their relative abundance could be attributed to the differences in their tolerance and resistance in response to environmental fluctuations. For example, a greater number of abundant ASVs belonging to Firmicutes exhibited higher relative

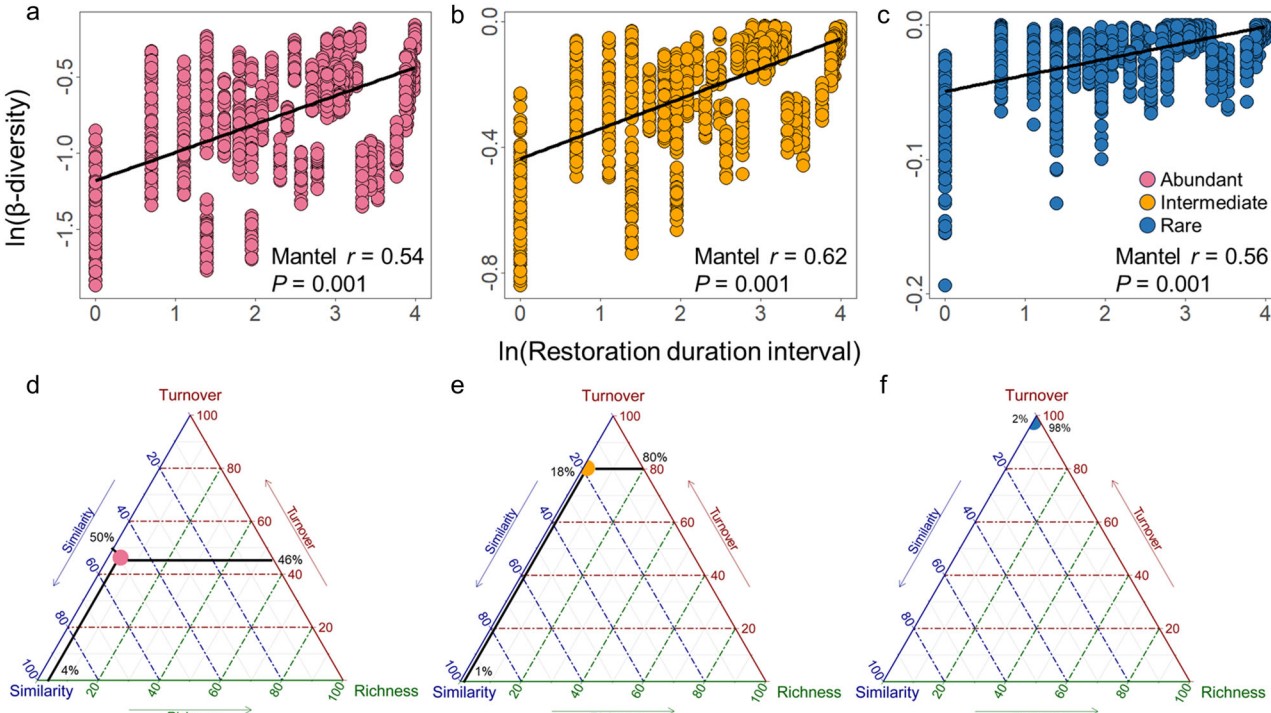

**Fig. 3 | Temporal variations in β-diversity and the visualization of its compositional components. a–c** Relationships between restoration duration dissimilarity and β-diversity of abundant, intermediate, and rare subcommunities based on the Mantel test. The black fitted lines are from linear regression. **d–f** Triangular plots of the components of β-diversity of abundant, intermediate, and rare subcommunities. The three axes represent compositional similarity and the two components of β-diversity (i.e., species turnover and richness difference) based on Bray–Curtis dissimilarity. Each point represents a triplet of average values from the corresponding similarity, turnover, and richness difference matrices, constrained such that their sum equals 1. The average similarity, species turnover, and richness difference were 50%, 46%, and 4% for abundant subcommunities; 18%, 80%, and 1% for intermediate subcommunities; and 2%, 98%, and 0% for rare subcommunities.

abundance during the early stages of restoration, due to their high tolerance to extreme drought conditions[46].

Our findings indicated that the successional dynamics of abundant and rare taxa were governed by contrasting community assembly processes during restoration, which aligns with previous studies[23,43]. Abundant taxa, typically ecological generalists with broad niche breadth, can exploit a wider range of resources. Therefore, their distribution patterns are primarily shaped by stochastic processes (e.g., dispersal limitation or ecological drift), as generalists tend to be less influenced by environmental selection[52–54]. In contrast, rare taxa, as specialists, have narrower range of habitats and specific environmental fitness, making their community assembly predominantly deterministic, particularly through homogeneous selection[20,52,55]. This is because homogeneous environmental conditions exert strong selective pressure on rare taxa, resulting in their perpetually low relative abundance and convergent community composition[42,56]. Additionally, the implementation of artificial restoration efforts and the establishment of plants introduce variable selection by altering the soil physicochemical properties throughout restoration process[39,42]. Thus, variable selection served as a secondary yet critical process orchestrating the compositional turnover of both abundant and rare subcommunities[42]. Intermediate taxa (comprising some ASVs that fluctuate between rare and abundant states) exhibit heightened sensitivity to these environmental changes, as their ecological fitness is strongly influenced by temporal shifts in environmental conditions[56]. Notably, the importance of homogenizing dispersal on both abundant and intermediate taxa progressively diminished as restoration proceeds. This trend is partially attributed to the prevalence of wind erosion and environmental homogeneity during the early stages of restoration, which facilitated the stochastic dispersal of species[46,57].

In this study, the increased richness of high-abundance taxa may reflect the diversification ecological niches shaped by variable selection with restoration duration, which promotes stability and functionality of

ecosystems, thus the richness of high-abundance taxa was positively correlated with ecosystem multifunctionality. In contrast, the richness of rare taxa showed no significant or negative (see Fig. 6b) relationship with multifunctionality, likely due to their population constraints and narrow ecological niches. At the ASV level, high-abundance taxa were generally associated with multiple functions concurrently, while most rare taxa were associated only with specific functions. This divergence can be attributed to the fact that, despite greater overall richness, rare subcommunities are less metabolically versatile than high-abundance taxa, which typically dominate niche space and possess broader metabolic repertoires that underpin key nutrient cycling functions[23,28]. Supporting this interpretation, a recent study demonstrated that bacterial taxa with broader niche breadth maximize the completeness and diversity of metabolic pathways essential for growth and resource acquisition, such as carbon fixation, ATP synthesis, and carbohydrate and nitrogen metabolism[58]. These findings further emphasize that mass ratio effects, rather than diversity effects, may play a more important role in maintaining key ecological processes and functions throughout the multi-decadal restoration of desertified ecosystems[59,60]. However, some studies have shown that rare taxa play unique ecological roles, particularly in specific ecological processes involving cycling and transformation of sulfur-containing compounds[37]. In addition, rare taxa act as a "seed bank" capable of becoming dominant under favorable conditions[61], thus playing an important role in ecosystem response to environmental changes.

Co-occurrence network analysis has been emerged as a powerful tool for deciphering interspecies connections that significantly shape biodiversity patterns and ecosystem functioning[62–64]. In this study, abundant taxa exhibited elevated degree and centrality, serving as module hubs within the network, whereas rare and intermediate taxa served as keystone connectors that bridged distinct modules. This delineates discrete yet distinct ecological roles for these taxa in maintaining network structure and stability. Although our SEM analysis

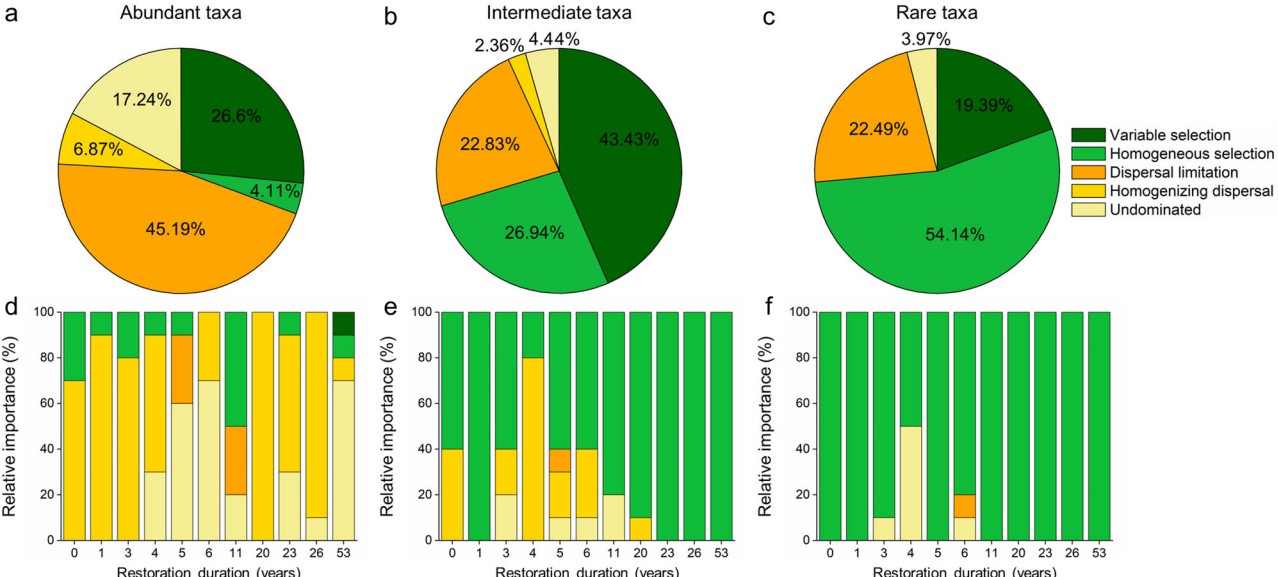

**Fig. 4 | Relative importance of ecological processes mediating the assembly of abundant, intermediate, and rare subcommunities. a–c** Relative importance of the assembly processes of abundant, intermediate, and rare subcommunities throughout the entire restoration process. **d–f** Relative importance of the assembly processes of abundant, intermediate, and rare subcommunities across different restoration stages. The relative importance of these five ecological processes was estimated using null model analysis.

revealed no significant correlation between the average subnetwork degree and ecosystem multifunctionality, subsequent ASV-level analysis indicated that the relative abundances of both high-degree abundant ASVs and low-degree rare ASVs exhibited stronger correlations with multifunctionality. Collectively, these insights point toward a dual mechanistic framework through which bacterial communities regulate ecosystem multifunctionality via complementary roles of these key taxa. Specifically, highly connected, abundant taxa presumably due to shared environmental preferences functioned as module hubs, exhibiting extensive co-occurrence patterns with other ASVs. These taxa thereby act as key contributors to multiple ecological processes via mass ratio effects, functional integration, and metabolic versatility, enabling comprehensive and efficient resource utilization to maintain high functional performance[25]. Conversely, rare taxa occupied unique niches and perform specific functions, resulting in their independent association with multifunctionality that relies little on interspecies connections or shared environmental responses[25,37].

This study provides practical insights for advancing dryland restoration. First, conventional afforestation and grass-planting effectively enhance plant cover and the richness of high-abundance microbial taxa (Fig. 6b). However, additional regulation of soil properties (e.g., soil pH) is necessary to meet the niche requirements of rare microbial taxa, and thereby promoting the functional redundancy of microbial communities during ecological restoration. Such enhancements are instrumental in bolstering ecosystem multifunctionality and ensuring long-term ecological stability[29]. Furthermore, the keystone ASVs identified may serve as reliable bioindicators for monitoring dryland restoration, given their pivotal role as ecosystem engineers and their ability to orchestrate microbial communities in performing ecosystem functions[65]. Therefore, successful restoration may be achieved with the introduction of these ASVs, either by targeted isolation and cultivation in artificial environments or via the deployment of commercially available microbial inoculants.

Despite the valuable insights provided by our study into the distinct characteristics of abundant and rare taxa, some limitations warrant attention and should be addressed in future research. For instance, microbial co-occurrence networks reflect common selection patterns driven by shared environmental gradients, rather than providing conclusive evidence of actual ecological interactions[63,66]. In light of this, further studies should

utilize cutting-edge omics technologies, and consider integrating co-occurrence and metabolic networks to infer more robust interactions among microbial species and elucidate the potential underlying ecological mechanisms[67,68]. Additionally, the null-modeling-based quantitative framework used to assess the relative contributions of microbial community assembly processes is inherently relies on assumptions regarding phylogenetic signals, as well as the abundance and distribution of individuals within a community[69,70]. Applying this framework to different subcommunities in our study may introduce mathematical biases, potentially obscuring nuanced inter-group interactions[56]. Despite these methodological caveats, our results still provide a robust empirical foundation for understanding how assembly processes differentially structure microbial subcommunities during ecological restoration.

In summary, we have characterized diversity patterns, community assembly processes, and relationships with multifunctionality of abundant and rare bacteria along a 53-year restoration chronosequence of desertified ecosystems. The richness of abundant, intermediate, and rare taxa significantly increased with restoration duration. Species replacement emerged as the main driver of community variations in both abundant and rare subcommunities. Stochastic processes dominated the succession of abundant taxa, owing to their broad ecological niches, while homogeneous selection primarily structured rare subcommunities, which persisted at consistently low abundances. Meanwhile, our results revealed distinct relationships of abundant and rare taxa with ecosystem multifunctionality. Abundant taxa were simultaneously associated with multiple ecosystem functions through coordinated environmental responses or potential interspecies connections, whereas rare taxa tended to be linked to individual ecosystem functions independently. This dual mechanism highlights the complementary role of both microbial groups in maintaining ecosystem multifunctionality during restoration. Our findings contribute to a deeper understanding of the interplay between soil microbial communities and multifunctionality, offering profound insights to guide the formulation of restoration and conservation policies in drylands.

## Methods
### Field survey and sampling
The study area is located on the southern edge (37°35′55″N, 103°50′31″E) of the Tengger Desert in northwest China. The mean elevation, mean annual

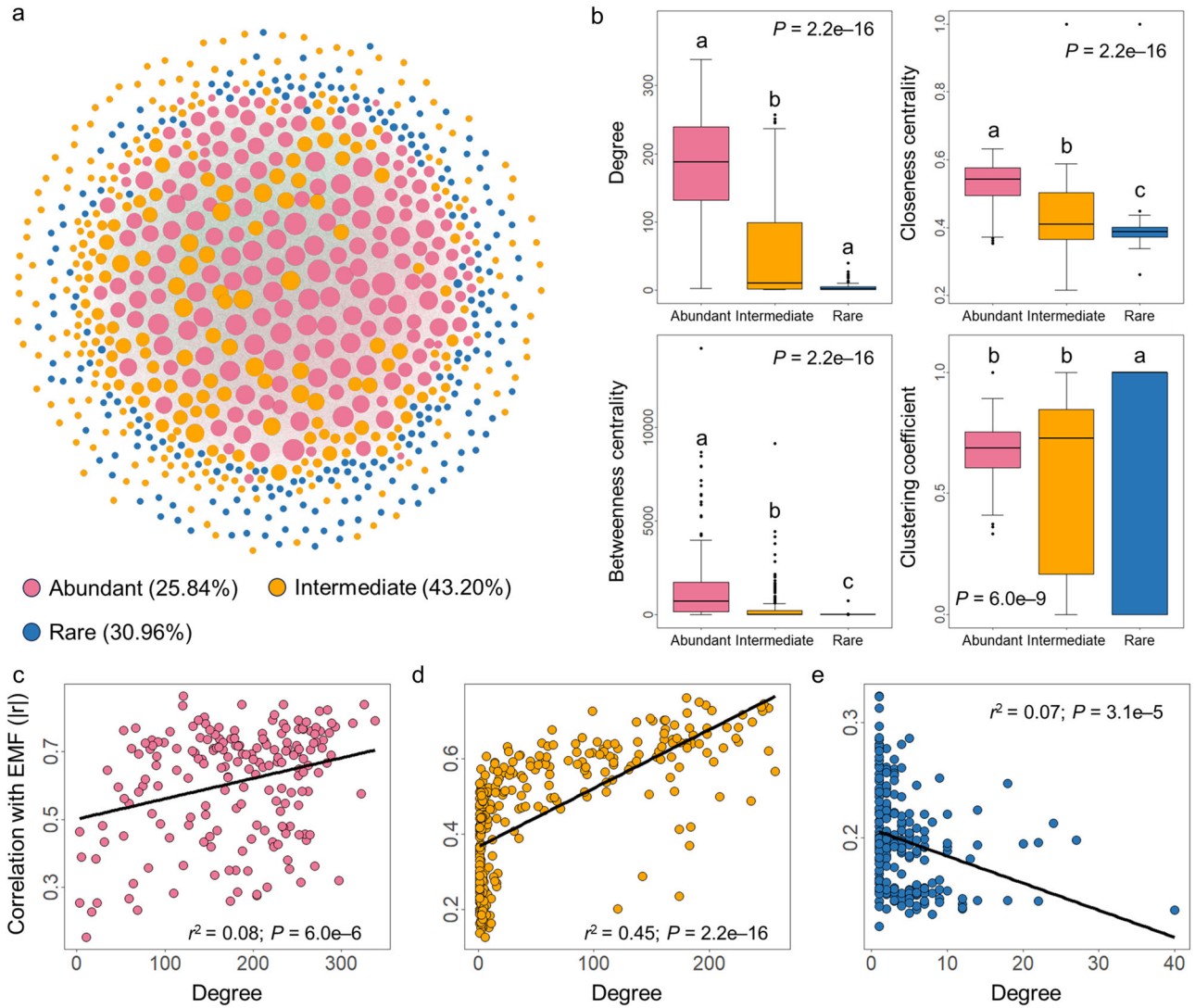

**Fig. 5 | Co-occurrence network of abundant, intermediate, and rare taxa. a** The co-occurrence network constructed with abundant, intermediate, and rare taxa. The size of each node is proportional to its degree. The values below the network represent the number proportion of nodes in each group. **b** Boxplots of node-level topological properties of abundant (*n* = 192), intermediate (*n* = 321), and rare taxa (*n* = 230), including degree, closeness centrality, betweenness centrality and clustering coefficient. The different letters represent significant differences determined by the Kruskal–Wallis test followed by Dunn post-hoc test (Bonferroni correction). **c–e** Relationships between the degree of ASVs and their correlation strength (|*r*|) with multifunctionality for abundant, intermediate, and rare taxa. The black fitted lines are from linear regression. EMF ecosystem multifunctionality.

temperature and mean annual precipitation are 1680 m a.s.l., 8.7 °C and 179 mm, respectively. This region is seriously affected by land degradation and desertification caused by wind erosion and is therefore defined as a priority area for implementing ecological conservation and restoration projects in China[8]. Since the 1950s, SCBs, as a cost-effective restoration technology, have been extensively established in this region to protect transportation routes, alleviate soil erosion, and restore desertified ecosystems[11]. SCBs can promote soil nutrient accumulation and facilitate the development of biocrusts[71]. Our recent study has demonstrated that the establishment of SCBs triggers a progressive convergence of plant and soil microbial communities toward those characteristic of non-desertified natural states, albeit complete restoration remains unrealized even after 53 years[46]. During this process, plant-soil microbe correlations and elevated soil organic carbon content emerged as key drivers of community succession and ecological restoration. In practice, the establishment of SCBs involves arranging wheat straw in a checkerboard pattern composed of numerous squares in mobile sandy land, with one half of straw buried in the sand and the other half exposed[12]. Plant communities on mobile sandy land are

dominated by *Agriophyllum squarrosum*, with a cover of around 2%, while plant communities in non-desertified natural ecosystems are dominated by *Artemisia ordosica*, with a cover of around 45%[46].

In this study, a well-documented restoration chronosequence spanning 53 years through the establishment of SCBs was identified. Field data were collected between August and September 2017 from a mobile sandy land plot with a restoration duration of 0 years (i.e., an unrestored desertified ecosystem) and 10 plots restored by SCB treatment, each with different restoration durations (i.e., 1, 3, 4, 5, 6, 11, 20, 23, 26 and 53 years; Supplementary Fig. 1)[46]. It should be noted that to obtain a longer chronosequence, the 53-year restoration plot was located in the Shapotou restoration area, the earliest site in China to implement SCBs for sand fixation. The other restored plots were situated within the main sampling area of Jingtai County (Supplementary Fig. 1b). Nevertheless, all study plots were positioned along the southern edge of the Tengger Desert, ensuring consistent environmental conditions. Specifically, a 12 × 12 m quadrat was established in each plot. In each quadrat, five 12-m-long transects (spaced 1.2 m apart) were arranged to measure plant cover and harvest above-ground plant materials following

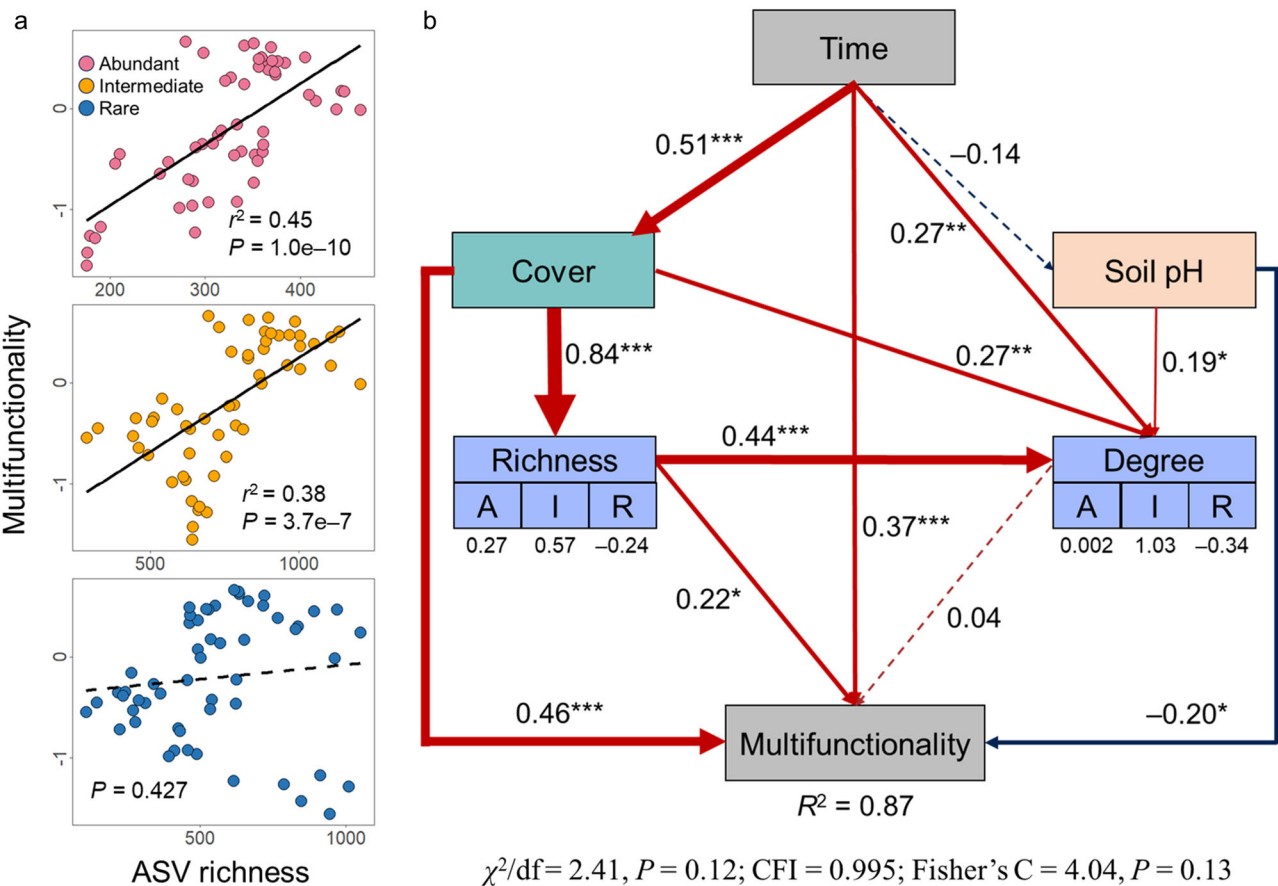

**Fig. 6 | Relationships of abundant, intermediate, and rare taxa richness with ecosystem multifunctionality. a** Relationships between multifunctionality and richness of abundant, intermediate, and rare taxa. The black fitted lines are from linear regression. The solid and dotted lines represent statistically significant ($P \leq 0.05$) and nonsignificant ($P > 0.05$) relationships, respectively. **b** Structural equation model depicting the hypothesized direct and indirect relationships among restoration duration, plant cover, soil pH, bacterial ASV richness, average degree and multifunctionality. The richness and degree of abundant, intermediate, and rare taxa were each consolidated into a composite variable using the standardized coefficients (as shown in the values below, representing their weights) from a linear regression model. Numbers adjacent to arrows (path coefficients) represent standardized effect sizes. Significant $P$ values are represented by *** when $P < 0.001$, ** when $P < 0.01$ and * when $P < 0.05$. Arrow width is proportional to the magnitude of standardized path coefficients. Red and blue arrows represent positive and negative relationships, respectively, with solid and dashed arrows denote statistically significant and non-significant relationships, respectively. $R^2$ is the proportion of variance explained by the model. Goodness-of-fit of SEM is evaluated by the Chi-square test, CFI, and Fisher's C test. Time, restoration duration.

the line-intercept protocol; five $1.2 \times 1.2$ m subplots (spaced at least 4 m apart) were randomly selected to collect soil samples[46,72]. In each subplot, 5–7 soil cores (0–10 cm depth) were collected randomly, and then thoroughly mixed to form a composite sample in the field. A total of 55 soil samples were collected from the 11 quadrats. The samples were sieved (<2 mm) and divided into three parts. One part was dried for the determination of physicochemical properties, another part was stored at 4 °C for measuring enzyme activities, and the last part was stored at –80 °C for DNA extraction (within one week). Soil pH was determined using a pH meter in a 1:2.5 mass:volume soil and water suspension.

**Molecular analyses**

The genomic DNA of each soil sample ($N = 55$) was extracted from 0.5 g of defrosted soil using the PowerSoil® DNA Isolation Kit (Mo Bio Laboratories, Carlsbad, CA, USA). The V3 − V4 hypervariable region of the bacterial 16S rRNA gene was amplified with the universal primer pair 338 F (5′-ACTCCTACGGGAGGCAGCAG-3′) and 806 R (5′-GGAC-TACHVGGGTWTCTAAT-3′)[73,74]. Subsequently, the 16S rRNA amplicons underwent paired-end sequencing ($2 \times 300$ bp) on an Illumina MiSeq platform (Illumina, San Diego, CA, USA) at the Majorbio Bio-pharm Technology Co., Ltd. (Shanghai, China) following standard protocols. The raw sequences were processed by removing adapter, trimmed, quality filtered, and deduplication using Cutadapt (v.1.18)[75] and fastp (v.0.23.4)[76,77].

The resulting high-quality sequence data was analyzed using the Quantitative Insights into Microbial Ecology (QIIME2 v.2024.10) pipeline[78]. The DADA2 plugin was then employed to denoise and generate feature sequence and table of amplicon sequence variants (ASVs)[79]. The taxonomic identity of each ASV was determined using the SILVA database (v.138)[80]. Sequences assigned to mitochondria, chloroplasts, and archaea were excluded from downstream analysis. Compared to operational taxonomic units (OTUs) clustered at 97% similarity, the ASVs approach allows precise identification of differences in each amplicon sequence, thereby reduces the omission of rare taxa and provides higher resolution at the species level[81]. A total of 2,505,038 sequences and 25,077 ASVs were obtained. To standardize richness comparison among samples at an equal sampling depth[73], the ASV table was rarefied to the lowest number of sequences (24,911) found within an individual sample for richness calculations.

**Classification of abundant, intermediate and rare taxa**

To minimize arbitrariness in classification, we employed MultiCoLA based on the entire dataset to define abundant and rare taxa (Supplementary Fig. 2)[48]. Initially, ASV dataset was ranked in ascending order on their sequences count. Low-abundance ASVs were removed based on the continuous percentage (1–95%) of the total number of sequences to obtain abundant (i.e., rare ASVs were removed) and rare (i.e., rare ASVs were retained) truncated datasets (Supplementary Fig. 2a, b). Pairwise distance

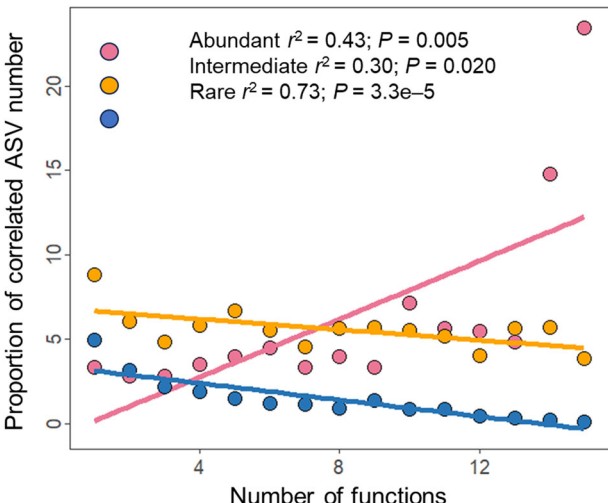

**Fig. 7 | Relationships between the number of functions and the number proportion of function-related ASVs.** The fitted lines are from linear regression.

matrices were then calculated for the original and truncated datasets using the Bray–Curtis dissimilarity index. Subsequently, community structure and the main patterns of community variation (using non-metric multi-dimensional scaling) of the original and truncated datasets were compared by applying the non-parametric Spearman rho correlation coefficient and the Procrustes method using the R package "vegan" (Supplementary Fig. 2c–f)[48]. While the abundant dataset maintained a high degree of structural consistency with the original dataset, the rare dataset demonstrated marked divergence in community composition. Finally, we precisely define thresholds for delineating abundant and rare truncated datasets based on the Spearman and Procrustes correlation values, and calculated the corresponding relative abundances. The results showed that the abundant truncated dataset retained nearly identical community structure and variation compared to the original dataset until 45% (corresponding to an average relative abundance of 0.028%) of rare ASVs were removed, while the rare truncated dataset showed no obvious divergence from the original dataset when the threshold exceeded 11% (corresponding to an average relative abundance of 0.002%) (Supplementary Fig. 2). As a consequence, ASVs with an average relative abundance > 0.028% across all samples were defined as abundant taxa, those with an average relative abundance < 0.002% as rare taxa, and those with an average relative abundance between 0.002% and 0.028% as intermediate taxa (Fig. 1a). These three biological groups (abundant, intermediate and rare subcommunities) were then used in the subsequent analyses.

**Application of the space-for-time substitution approach**

In this study, we strategically utilized the space-for-time substitution approach to assess ecological dynamic across plots with varying restoration durations for the following compelling reasons. First, given the inherent difficulty of obtaining long-term data from the same plot, the space-for-time substitution approach is a widely acceptable and scientifically credible alternative in temporal studies of ecological succession[82,83]. Second, all selected plots shared similar treatment, initial conditions and environmental settings[46]. Finally, we evaluated the relationships between bacterial community composition (including entire dataset as well as abundant, intermediate, and rare taxa) based on Bray–Curtis distance, geographic distance, and restoration time interval between samples using the Mantel test with the R package "vegan" to evaluate spatial autocorrelation and time effects (Supplementary Fig. 3)[84]. These results indicated that differences in bacterial community composition were significantly associated with restoration time interval, but not with geographical distance (Supplementary Fig. 3). This suggests that the observed variations in bacterial communities were primarily determined by the age of the established SCBs, rather than by spatial

autocorrelation, despite the 53-year restoration plot in Shapotou was spatially separated from others.

**Ecosystem functioning measurement**

Seventeen plant and soil variables (hereafter referred to as "functions") were used to assess ecosystem multifunctionality in this study, including plant productivity-related functions (aboveground biomass, plant carbon, plant nitrogen, and plant phosphorus), decomposition rate-related functions (soil DNA concentration, the activities of sucrose, β-glucosidase, cellulase, urease, alkaline phosphatase and catalase), and soil nutrient pool-related functions (soil organic carbon, ammonium, nitrate, total nitrogen, available phosphorus and total phosphorus). These functions are good proxies of the storage and cycling of matter and energy, and are widely used in multifunctionality studies conducted in dryland ecosystems[14,72,73]. To obtain an average multifunctionality index for each sample, we first normalize ($\log_{10}$-transformed) and standardized each of functions using the Z-score transformation. The Z-scores of the 17 functions were then averaged to obtain a multifunctionality index[72]. Additionally, separate multifunctionality indices were also calculated using only the functions from plant productivity (PPI), decomposition rate (DRI) and soil nutrient pool (SNPI).

The plant materials were dried for 24 h at 80 °C and weighed to obtain aboveground biomass (AGB), and subsequently ground to a fine powder using a high-speed grinder for the analyses of plant nutrients. Soil organic carbon (SOC) and plant carbon (PC) were determined using the Walkley–Black chromic acid wet oxidation method. Soil total nitrogen (STN) and plant nitrogen (PN) were measured by the Kjeldahl sulfuric acid-digestion method. Soil ammonium ($NH_4^+ - N$) and nitrate ($NO_3^- - N$) were measured by a TOC-TN analyzer from 2 M KCl extracts. Soil total phosphorus (STP) and plant phosphorus (PP) were quantified using a molybdate colorimetric test after perchloric acid digestion, whereas soil available phosphorus (SAP) was assessed following a 0.5 M NaHCO₃ extraction. Soil sucrase activity (Suc) was measured with the substrate sucrose and expressed as µg glucose $g^{-1}$ dry soil $h^{-1}$. Soil β-glucosidase activity (β-Glu) was determined with the substrate p-nitrophenyl-β-D-glucopyranoside and indicated in units of µg p-nitrophenol $g^{-1}$ dry soil $h^{-1}$. Soil cellulase activity (Cel) was measured with the substrate sodium carboxymethylcellulose and reported in units of µg glucose $g^{-1}$ dry soil $h^{-1}$. Soil urease activity (Ure) was measured with the substrate urea and expressed as µg ammonia $g^{-1}$ dry soil $h^{-1}$. Soil alkaline phosphatase activity (ALP) was assayed using p-nitrophenyl phosphate as substrate and indicated in units of µg p-nitrophenol $g^{-1}$ dry soil $h^{-1}$. Soil catalase activity (Cat) was determined by potassium permanganate titration and expressed as µl (0.1 mol $L^{-1}$) $KMnO_4$ $g^{-1}$ dry soil $h^{-1}$. DNA concentration was measured using Nano-DropTM 2000 UV-Vis spectrophotometer (Thermo Fisher Scientific, USA).

**Statistics and Reproducibility**

All statistical analyses were carried out using R software (v.4.4.1) unless otherwise indicated. To evaluate the niche breadth of each bacterial ASV, we treated 17 measured ecosystem functions as environmental variables and calculated two complementary metrics: Levins' index and tolerance index. The Levins' index estimates niche breadth in terms of the distribution of ASVs across different environment conditions using the R package "MicroNiche"[85], whereas the tolerance index account for both the distribution range of ASVs and the variation in their relative abundances along environmental gradients (i.e., environmental tolerance), using the R package "ade4"[86]. Differences in niche breadth among abundant, intermediate, and rare taxa were evaluated using the Kruskal–Wallis test with a Dunn post-hoc test (Bonferroni correction) using the R package "dunn.test". We evaluated the responses of ASV richness of abundant, intermediate, and rare taxa across the restoration duration by fitting linear and nonlinear (quadratic, logarithmic, and restoration model) regression models, and selected the optimal model for each dataset based on the lowest AIC value. The restoration model developed by Poorter et al.[87] can capture the typical restoration trajectory of ecological attributes, exhibiting an initial increase

followed by stabilization, and was fitted using nonlinear least squares regression. Differences among ASV richness of abundant, intermediate, and rare taxa were further evaluated using the Kruskal–Wallis test with a Dunn post-hoc test (Bonferroni correction). Various statistical analyses related to community composition were carried out using the R package "vegan". Temporal variations in abundant, intermediate, and rare subcommunities were visualized by NMDS ordination based on Bray–Curtis distance. Analysis of similarity (ANOSIM) was utilized to evaluate the differences in subcommunities among samples with different restoration durations. Furthermore, to quantify the cumulative contributions of abundant, intermediate, and rare taxa to the overall community composition, we performed the similarity percentage (SIMPER) analysis based on Bray–Curtis distance. The Mantel test was used to evaluate the relationships between restoration duration interval and the β-diversity of abundant, intermediate, and rare subcommunities. The β-diversity was then partitioned into balanced variation in abundance (turnover, i.e., species replacement) and abundance gradient (richness) components based on Bray–Curtis distance using the R package "betapart"[88]. Additionally, we evaluated the variability in the relative abundance of each ASV over the restoration duration based on Pearson correlation to test whether their abundance remained relatively stable across samples.

The relative importance of the assembly processes governing abundant, intermediate, and rare subcommunities throughout the entire restoration process and across different restoration stages were further evaluated using a null model analysis[69,70,89]. Briefly, we calculated the β-nearest taxon index (βNTI) and the Bray–Curtis-based Raup–Crick ($RC_{bray}$) using the R packages "picante" and "ape", to quantify the relative importance of deterministic (i.e., variable selection and homogenous selection) and stochastic (i.e., dispersal limitation, homogenizing dispersal, and undominated) processes. A conspicuous deviation ($|\text{βNTI}| > 2$) indicates that the community turnover is primarily controlled by deterministic processes. Furthermore, $\text{βNTI} < -2$ or $> +2$ indicates that homogenous selection or variable selection caused the similarity or difference between a pair of communities, respectively. In contrast, $|\text{βNTI}| < 2$ indicates that stochastic processes drive the observed turnover. Specifically, $RC_{bray} > 0.95$ and $RC_{bray} < -0.95$ indicate the relative contribution of dispersal limitation and homogenizing dispersal, respectively, whereas $|RC_{bray}| < 0.95$ represents the influence of the "undominated" processes mainly composed of weak selection, weak dispersal, diversification, and ecological drift. In summary, for all pairwise community comparisons, the relative importance of variable selection and homogeneous selection were estimated by calculating the fractions of pairwise βNTI values above $+2$ and below $-2$, respectively. Within this set of pairwise community comparisons not governed by selection (i.e., those with $|\text{βNTI}| < 2$), the relative importance of dispersal limitation and homogenizing dispersal were estimated as the fractions of all pairwise comparisons with $RC_{bray} > +0.95$ and $< -0.95$, respectively, while the fraction of all pairwise comparisons with $|RC_{bray}| < 0.95$ estimates the relative importance of the "undominated" processes.

A co-occurrence network consisting of abundant, intermediate, and rare ASVs was constructed to reveal their shared abundance dynamics along restoration duration or potential interspecies connections[62]. We focused solely on the ASVs that exhibited significant changes in relative abundance with restoration duration, as they reflect temporal variations in the composition of soil microbial subcommunities. SparCC method was used to construct microbial co-occurrence network using the R package "SpiecEasi"[90], and only correlations with absolute values of SparCC correlation coefficients ($\rho$) > 0.3 and $P < 0.05$ were retained. In the resulting network, nodes represented bacterial ASVs, and edges represented correlations among nodes that reflect parallel changes in the abundance of ASVs across samples[64]. The network was visualized using Gephi software (v.0.9.2). We calculated the key topological properties for each node, including degree, closeness centrality, betweenness centrality and clustering coefficient, using the R package "igraph". These metrics are widely employed in microbial network analysis to assess the importance of nodes, and are potentially associated with community stability and function[91,92]. Degree

refers to the number of edges connected to a node. Nodes with high degree co-vary with many other nodes, often acting as key drivers of community function, resource distribution, and resilience. Closeness centrality measures the proximity of a node to all other nodes in a network via the average shortest path length, reflecting the speed and efficiency with which a node can propagate changes or respond across the network. Betweenness centrality quantifies the number of shortest paths passing through a node. Nodes with high betweenness centrality can effectively regulate the flow of information or metabolites, and are crucial for maintaining network stability. The clustering coefficient measures the degree to which a node's neighboring nodes are interconnected, calculated as the ratio of the actual to all possible connections among adjacent nodes. A higher clustering coefficient indicates that neighboring nodes are more tightly interconnected, suggesting localized clustering. Thus, nodes with higher degree, closeness centrality, betweenness centrality and clustering coefficient are regarded as key hubs for information transfer and connectivity, playing a crucial role in maintaining the complexity and functionality of the network[65,92,93]. Subsequently, we assessed the differences in these topological properties among abundant, intermediate, and rare nodes using the Kruskal–Wallis test with a Dunn post-hoc test (Bonferroni correction). The keystone ASVs were further determined based on the within-module connectivity ($Z_i$) and among-module connectivity ($P_i$), including network hubs ($Z_i > 2.5$ and $P_i > 0.62$), module hubs ($Z_i > 2.5$ and $P_i \leq 0.62$), and connectors ($Z_i \leq 2.5$ and $P_i > 0.62$)[94,95]. As a supplement, we constructed networks for abundant, intermediate, and rare taxa separately, and compared their stability. The stability of the network was characterized by robustness, which is defined as the proportion of remaining nodes after removing 50% of random nodes at 1000 iterations[96,97]. Furthermore, we constructed subnetworks at different restoration stages and calculated the key topological properties of each node using the same standards mentioned above to evaluate how the network structure and the importance of nodes change over restoration duration.

Ordinary least-squares linear regressions were performed to assess relationships between multifunctionality indices and ASV richness for abundant, intermediate, and rare taxa. Pearson's correlation coefficients ($r$) between individual functions and ASV richness were calculated. Subsequently, a structural equation model (SEM) was constructed to infer the hypothesized direct and indirect relationships between restoration duration, plant cover, soil pH, bacterial ASV richness, network properties (i.e., average degree in subnetworks at each restoration stage), and multifunctionality across samples using the R package "piecewiseSEM". These relationships in the SEM were modeled using linear models. To avoid model overfitting, ASV richness and degree of abundant, intermediate, and rare taxa were each consolidated into a composite variable using linear regression models prior to SEM construction, using standardized coefficients serving as their weights. The abundant and intermediate taxa had positive weights, while the rare taxa had a negative weight for the composite richness and subnetwork degree. We then employed the chi-square test ($\chi^2$), comparative fit index (CFI) and Fisher's C test to evaluate the goodness-of-fit statistics of the model. Mantel tests were conducted to evaluate the correlations between multifunctionality and β-diversity, as well as its components, for abundant, intermediate, and rare subcommunities.

To delve deeper into the mechanisms by which different taxa maintain multifunctionality, we conducted ASV-level analyses. Initially, we quantified both the proportion and absolute correlation strength ($|r|$) of ASVs significantly associated with individual ecosystem functions and multifunctionality indices for abundant, intermediate, and rare taxa. Subsequently, we specifically calculated the number proportion of ASVs associated with varying numbers of functions for abundant, intermediate, and rare taxa, and evaluated their linear relationships[25]. In addition, using the aforementioned co-occurrence network, we examined the relationships between the degree of ASVs and their correlation strength ($|r|$) with multifunctionality. This analysis aimed to uncover extent to which shared responses or potential interspecies indirect connections contribute to multifunctionality[25].

**Reporting summary**

Further information on research design is available in the Nature Portfolio Reporting Summary linked to this article.

## Data availability

The datasets that support the main findings of this study are publicly available on figshare [https://doi.org/10.6084/m9.figshare.26778400][98]. The raw bacterial sequence data have been deposited in the NCBI SRA database under accession code PRJNA877077.

## Code availability

The R code used to generate the main results of this study is publicly available on figshare [https://doi.org/10.6084/m9.figshare.26778400][98].

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

## Acknowledgements

We would like to thank the Shapotou Desert Research and Experiment Station (Chinese Academy of Sciences) for the support in field sampling. This research was funded by the National Key Research and Development Program of China (2023YFF0805602), the National Natural Science Foundation of China (32225032, 32001192, 32271597), the Innovation Base Project of Gansu Province (2021YFF0703904), the Natural Science Foundation of Gansu Province (23JRRA1157, 22JR5RA525), the Key Research and Development Program for Top Leading Talents in Gansu Province (23ZDKA0010), and Research Grant Support from Lanzhou City (127000–563224111).

## Author contributions

J.D. and W.H. conceived the study. Field data were collected by Q.H., W.H., Y.S., X.L., and A.M. Laboratory measurements were performed by Q.H., R.X., Y.S., L.D., A.M., F.L., Y.D. and B.Y. Statistical analysis and data integration were carried out by Q.H., R.X., B.Y., and W.H. with suggestions and help from J.R. and J.D. The first draft was written by Q.H., J.R,. and J.D. M.A., G.W., J.W., X.L. and X.G. contributed to the writing of the manuscript. All authors read and approved the final manuscript.

## Competing interests

The authors declare no competing interests.
