## [Transparent Peer Review file · Communications Biology]

Divergent community assembly processes and multifunctionality contributions of abundant and rare soil bacteria during a 53-year restoration in the Tengger Desert, China

Corresponding Author: Professor Jianming Deng

Version 0:

Reviewer comments:

Reviewer #1

(Remarks to the Author)

The authors present a study using 16S sequencing of 55 samples from 11 sites, which have been treated with straw checkerboard barriers (SCBs) over the past 53 years, to explore whether rare or abundant microbial taxa are more critical in driving ecosystem restoration. While I commend the authors for their data analysis skills and the amount of information they were able to extract from a relatively small dataset, I have several concerns that suggest the study is overstated in its current form and requires significant further clarification and justification.

1. Title and Scope: The title of the manuscript overstates the scope of the study, particularly given that it focuses on a localized area. As a study conducted on a local scale, it is essential to clearly specify the geographical context (e.g., include the location or refer to "arid ecosystems"). While small-scale studies certainly have value, the high-impact language used in the title is not aligned with the scale of the study. I strongly recommend tempering the title and framing the findings within a more appropriate context.

2. Interpretation of Results and Causality: A key concern is the interpretation of the results. One of the main research questions posed by the authors is whether rare taxa play a more significant role than abundant taxa in driving ecosystem restoration. However, the manuscript is based on correlation analyses, and the use of language that implies causality is misleading. Correlation does not equate to causation, and the experimental design and statistical analysis employed here do not support the claim that "high-abundance taxa drive a greater number of ecosystem functions through their interactions," nor does it demonstrate whether ecosystem functions drive the interactions of high-abundance taxa. A more cautious interpretation of these findings is necessary.

3. Study Design and Restoration Approach: A detailed description of the study site and restoration methods is lacking. Given that the study employs a space-for-time substitution design (which is technically a pseudoreplication), it is possible that the observed patterns are driven more by the proximity of the sites than the restoration method itself. Additionally, it is important to reference previous research on straw checkerboard barriers, including their effects on soil properties and vegetation communities, to help readers understand their potential ecological impacts. This would provide a stronger foundation for hypothesizing about the role of microbes in ecosystem restoration.

4. Definition of Abundant, Intermediate, and Rare Taxa: The classification of taxa into abundant, intermediate, and rare categories needs further justification. While I appreciate the authors' efforts to reduce arbitrariness, the current definitions appear biologically inconsistent. Several issues arise:

1. The categorization is based on the average relative abundance of OTUs across all samples. This approach could misclassify OTUs that are rare in early successional stages but become dominant later—these taxa are potentially the most ecologically significant, yet they are likely categorized as intermediate.

2. OTUs are not a taxonomic level, and the comparison of an order with a species can be problematic, as orders would naturally be more abundant than species. This discrepancy should be addressed by standardizing the analysis to a common taxonomic level.

3. Rarefying data can bias results against rare taxa, and I am not convinced it is appropriate in this context. Additionally, singletons could be considered rare taxa, but they seem to be excluded from the analysis, further biasing the results.

5. Network Analysis and Rare/Abundant Taxa: I also question the suitability of network analysis for examining abundant and rare taxa. Network analysis is based on correlations between taxa, and the patterns observed could be driven by shared

environmental factors that shape taxa distributions within the same sample, rather than by ecological interactions. The occurrence of abundant taxa in more samples naturally lead to stronger correlations, but this does not necessarily reflect biologically meaningful interactions. Therefore, the application of network analysis in this context requires more careful consideration and a discussion of its limitations.

Conclusion: In summary, while the authors have presented a technically sound analysis, the manuscript overstates its findings, and significant revisions are necessary before it can be considered for publication. Clarification of the study design, justification of taxa definitions, and a more cautious interpretation of the results are essential. Additionally, the title and claims made in the manuscript should be tempered to better reflect the study's local scale and correlational nature.

Reviewer #2

(Remarks to the Author)

This study investigates successional dynamics over a 53-year restoration chronosequence following the implementation of straw checkerboard barriers (SCB) in the Tegger Desert, China. The authors demonstrate that SCB increased the diversity of abundant taxa over time, with negligible effects on rare taxa. Their findings emphasize the pivotal role of abundant taxa in shaping bacterial community composition, as well as enhancing the complexity and stability of co-occurrence networks. Variable selection emerged as the predominant mechanism driving the assembly of both abundant and rare taxa. Notably, the study highlights that abundant taxa facilitated multiple nutrient cycling functions through interactions, while rare taxa contributed independently to single functions.

I commend the authors for undertaking the challenging task of analyzing the dynamics of rare taxa, defined here as taxa with low abundance and limited representation across samples in amplicon datasets. This study reflects significant effort, from meticulous sampling for amplicon sequencing to the thorough characterization of environmental and functional soil properties.

That said, one aspect that detracts from the study's value is the reliance on predictive databases (FAPROTAX) to infer functional capacities from OTUs. While I acknowledge the economic and technical challenges associated with metagenomic or metatranscriptomic analyses, predictive software introduces inherent limitations that may reduce the robustness of the functional inferences presented in the manuscript. This reviewer would rather remove these predictions and focus on the amplicon data and the rest of the analyses.

Additionally, analyzing rare taxa requires substantial sequencing depth, yet the authors do not provide any statistics on sequencing depth. To address this, the authors should include accumulation (rarefaction) curves for highly abundant, mid-abundant, and rare taxa to ensure proper diversity representation. Without this analysis, the observed lack of association between rare taxa and multifunctionality could be attributed to poor representation of rare communities. Such underrepresentation could cascade across all analyses, including the co-occurrence-based network analysis. The authors mention that rare taxa appear in only 10% of the samples, which they link to the low connectivity of this group in the reconstructed network. However, this claim warrants further support through the inclusion of diversity plateau evidence.

The authors state in the abstract and discussion that high- and mid-abundance taxa contribute to network stability, while rare taxa do not. However, network stability was not directly measured. Instead, only certain network topological properties, such as degree, were assessed and correlated with multifunctionality. Clarifying and justifying the connection between network stability and the selected topological metrics is essential to support the manuscript's conclusions.

Turning to methodology, several key aspects lack sufficient detail. For instance, soil sampling—a critical component of the study—was relegated to the supplementary materials. These details should be brought back into the main text. Similarly, the network reconstruction strategy is not fully described. The authors do not specify the software used, or the rationale behind the selection of the correlation threshold used for network reconstruction. Moreover, only one network combining the three groups (high, mid-abundance, and rare taxa) was reconstructed. This raises questions about the data underlying the boxplots in Figure 5. Were the networks randomized to calculate these metrics, or were multiple networks reconstructed under different conditions to enable replication? Or was centrality and other parameter calculate per node and used for comparison? Including this information is essential for the reproducibility of the study.

Another methodological concern is the statistical approach. The authors use the Mann-Whitney U test to compare three groups, but this test is designed for pairwise comparisons. Using it to compare three groups would require conducting multiple pairwise comparisons, which could inflate the Type I error rate (false positives) unless corrected (e.g., using a Bonferroni adjustment). Even with corrections, this approach is less efficient and not ideal for multi-group comparisons. A more appropriate statistical test, such as Kruskal-Wallis with a post-hoc analysis, should be applied to allow comparisons across all three groups.

In conclusion, this study has the potential to contribute meaningfully to the field of microbial ecology. However, addressing the methodological and analytical concerns outlined above is necessary to enhance the rigor and reliability of the findings.

Specific comments:

Lines 124-125, are confusing in the current format – authors seem to provide redundant information regarding the prevalence of rare taxa.

Line 168 – define the network properties – i.e degree is the measure of the number of links associated with a node, etc.

Improve methods

Figure 6. Legend must include a definition of the multifunctionality metrics – what is PC, PN, PP, etc.

Version 1:

Reviewer comments:

Reviewer #1

(Remarks to the Author)

I appreciate the authors' effort in refining their manuscript by moderating their claims, adding details about the experimental design, and addressing limitations. However, the major concerns I raised in the previous review still require substantial revisions through additional analyses or supporting materials rather than just wording adjustments.

1. Justification of pseudo-replication

A detailed experimental design map is needed to properly address concerns about pseudo-replication. While you have provided more details on the sampling plan and the known effects of straw checkerboard barriers (SCB) on soil properties, your justification that "the space-for-time substitution approach is widely employed in temporal studies of ecological succession and is generally regarded as reliable and scientifically sound" is more applicable to plant succession than to soil microbial communities, given the different spatial scales involved.

As I suggested previously, please include a map that clearly illustrates the spatial scale of your study. Additionally, I recommend consulting Soil DNA chronosequence analysis shows bacterial community re-assembly following post-mining forest rehabilitation by Peddle et al. (Figure 5) as an example of how to demonstrate that observed microbial community variance is not merely an artifact of spatial distance.

2. Definition of abundant, intermediate, and rare taxa

The classification of taxa based on the average abundance across all samples appears arbitrary and does not account for taxa that shift from rare to abundant over the course of succession. You identified ~140 taxa that follow this pattern and categorized them as intermediate, suggesting that some taxa start as rare but become abundant by the end of restoration. Shouldn't these taxa receive more discussion?

Additionally, the "sample-specific rarity cutoff" remains unclear. According to Figure 1a, OTUs exceeding 0.3% in some samples constitute only a small portion of the most abundant OTUs, yet a large fraction of them ultimately fall into the intermediate category. This suggests that inter-sample abundance variability is common and non-negligible.

To better justify your classification, I suggest calculating the abundance category (abundant/intermediate/rare) for each OTU in each sample and then determining the percentage of samples in which each OTU falls into each category. Alternatively, quantifying inter-sample abundance variability for each taxon could demonstrate whether most taxa remain stable across samples, supporting the use of an average-based classification.

While I still believe that tracking taxa with significant temporal shifts would provide deeper ecological insights, I acknowledge that this might require more extensive analyses beyond the scope of this revision.

3. Taxonomic resolution in abundance classification

Since OTUs can represent various taxonomic levels (phylum, class, family, genus, species), comparing abundance across these levels is problematic. While OTU-based analysis is common, its combination with rare/abundant classification is less so. A "class" with an unknown order could be classified as abundant, while an OTU at the species level could be categorized as rare.

To address this, I recommend reporting taxonomic resolution for each class (% of OTUs annotated to genus vs. phylum for abundant/intermediate/rare categories). If the resolution is consistent across categories, your approach should be robust. However, if there are significant discrepancies, you might consider restricting the analysis to OTUs with consistent taxonomic resolution (e.g., genus-level) to reduce ambiguity.

4. Methodological concerns regarding rarefaction, network analysis, and community assembly indices

This is my most critical concern. There needs to be a clear justification that rarefaction, network analysis, and community assembly indices are appropriate for a dataset that has been divided into abundant and rare taxa. While these methods are widely used in microbial ecology, they are typically applied to entire communities rather than split into categories. I am concerned that the observed patterns may be driven by mathematical artifacts rather than ecological processes.

Rarefaction disproportionately affects rare taxa, introducing stochasticity. The higher stochasticity observed in rare taxa's community assembly could be an artifact of rarefaction rather than an ecological signal. I suggest rarefying only for richness calculations and using unrarefied data for all other analyses.

Network analysis may be biased because abundant taxa are more frequently detected across samples, making them more likely to co-occur with other taxa. This can artificially inflate their degree (number of connections) and centrality (importance

in the network), leading to an overestimation of their ecological significance. To test for such biases, I recommend performing a supplementary analysis on a completely randomized microbial dataset, categorizing abundant/rare groups as you did, and checking whether similar patterns emerge.

Community assembly indices may also be affected, as dominant phyla could skew deterministic signals, while rare taxa—being absent in many samples—could inflate stochasticity. More clarification is needed on how your dichotomization might introduce mathematical biases in these methods.

Additionally, given the strength of your temporal dataset, why not analyze network structure and community assembly across different restoration stages? This would provide more valuable insights into microbial succession in the context of restoration ecology.

Detailed comments

Title: Why not specify "Tengger Desert" directly?

Introduction, Paragraphs 1 & 2: These sections would benefit from a clearer justification for why separating abundant and rare taxa is critical to understanding or improving ecological restoration.

One possible approach is to emphasize the ecological rationale: abundant taxa may be crucial for short-term functional recovery, while rare taxa contribute to ecosystem resilience through functional redundancy. Thus, studying them separately provides insights into different restoration goals.

Alternatively, you could frame the abundant-rare dichotomy as a trade-off in restoration strategies. Should restoration prioritize inoculating soils with rare taxa to enhance stress resilience, or focus on rebuilding abundant taxa to accelerate ecosystem productivity?

Introduction, Paragraph 3 (Line 69): Did you investigate temporal variation in community assembly? The current figures do not seem to incorporate a temporal component.

Introduction, Paragraph 4: This section effectively explains the importance of dryland restoration and SCBs, but it does not clearly justify why microbial communities—especially abundant and rare taxa—should be considered separately. A reasonable revision would be to merge this paragraph with Paragraph 1 and move SCB-related details to the Methods section. Paragraphs 2 and 3 should also be reframed with a stronger focus on restoration.

I will stop here, as the Results and Discussion sections may require significant revisions if you incorporate the additional analyses I suggested.

Reviewer #2

(Remarks to the Author)

Dear authors, thank you for your detailed answers to this reviewer's comments and questions. The manuscript has improved largely.

I have some small edits that I suggest in the following lines:

Lines 123-125: Please acknowledge the need for deeper sequencing depth for rare groups. It is not a negative aspect but something that must be acknowledged to inform future studies.

Line 172: Change "positive connections" to "positive correlations"

Line 173: Instead of "indicating higher prevalence of synergistic" say "indicating higher putative synergistic relationships"

Remember, correlation networks provide a view of correlations and provide insights for further hypotheses development.

Version 2:

Reviewer comments:

Reviewer #1

(Remarks to the Author)

General comments:

This round of revision has significantly improved the manuscript. I now have enough context to not recommend rejection. However, the manuscript still requires substantial revision to meet the standards of Communications Biology. Most importantly, I strongly recommend incorporating structural equation modeling (SEM) to synthesize your analyses and deliver a clear, integrative take-home message. Right now, the manuscript reads like a set of preliminary analyses without a compelling narrative. The discussion remains surface-level, especially in areas where your data could support deeper, testable insights. The network analysis, in particular, feels underdeveloped—please start by clearly defining its purpose in your study.

The data itself is highly promising, but I encourage you to take a step back and reflect more deeply on the knowledge gap you're addressing and the practical value of your findings. This is a rare 53-year restoration chronosequence—what lessons can you offer future restoration practitioners? Have microbial communities fully recovered with the current restoration approach? If not, how does your work help address that gap? If recovery is complete, which subcommunities restored more slowly, and what does that imply? Based on your assembly mechanism results, how could restoration outcomes be improved?

Detailed comments:

Line number based on 27703_2_revised_manuscript_marked_up_915090_sv5p5t_convrt

Lines 106–109: This section discussing shelterbelt forest restoration (SCBs) reads disconnected from the surrounding content. I suggest moving it earlier in the introduction (e.g., around Lines 38–41), potentially replacing or following the Three-North Shelterbelt example. Alternatively, consider moving this content to the Methods section. In the remainder of the

introduction, use a more general term like “restoration” rather than focusing specifically on SCBs.

I will jump to method before results:

Lines 461–463: Reference 44 is highly relevant. Please summarize its key findings explicitly here, rather than vaguely stating that it “triggers changes.”

Line 474 & Supplementary Figure 1: Why don't you include the 53-year restoration plot in panel a? Including it would improve clarity and completeness.

Line 530: Please briefly explain the MultiCoLA method and cite the original reference. Without this context, Supplementary Figure 2 is difficult to interpret. Also clarify the rationale for selecting the two cutoff values (0.028% and 0.002%).

Line 552: A scatter plot showing geographic distance versus Bray–Curtis dissimilarity would be helpful. I definitely not meant to accusing you are deliberately hiding something but it's possible that the 53-year plot is geographically distant from all others, potentially acting as an outlier and lowering the overall correlation. Given that geographic distance and restoration age seem correlated (per Supplementary Figure 1), it's surprising to see such a discrepancy in their respective effects.

Lines 555–556: Consider rephrasing to: “These results indicate that differences in bacterial community composition were significantly associated with restoration time intervals, but not with geographical distance (Supplementary Table 1).” This wording better reflects that you are discussing beta diversity, not absolute composition.

Line 597: Please indicate which software or platform and version was used for data analysis.

Lines 598–599: A reference is missing here. Also, specify whether any R packages were used to calculate the niche indices. More importantly, explain why these two indices are necessary. For instance, the `levins.Bn` function in the `MicroNiche` package requires you to define “environments.” What environments were compared here?

If you treated each sample as a unique environment (as implied), I suggest reconsidering the use of the Levin's index. If your “abundant taxa” are defined by both high abundance and widespread occurrence, the Levin's index based on sample occurrence may be redundant. If your goal is to show that abundant taxa occupy broader niches, consider using your ecosystem function measurements instead. For example, you could cluster samples into distinct “environmental” types, then compute the niche breadth based on that clustering. A similar logic likely applies to your tolerance index—please clarify its computation and rationale.

Lines 604–606: The combined use of regression and Kruskal–Wallis tests is unclear. Are you aiming to predict temporal trends or assess differences between time points? These methods serve distinct purposes. I recommend emphasizing the temporal trend. Consider fitting and comparing different regression models (e.g., linear, quadratic, logarithmic, spline-based) and reporting R^2 or AIC values to support your choice of model.

Line 647: While I'm relatively new to network analysis and Gephi, I would appreciate a clearer ecological interpretation of the topological properties you report. For instance, overall network metrics such as centralization of betweenness (rather than individual node values) have been used to infer microbial community stability—see Long et al. (2025, *Global Change Biology*, Fig. 1c) and Chen et al. (2024, *Communications Earth & Environment*, Fig. 1b). These may provide more meaningful contrasts than node-level metrics. Otherwise, please explain the ecological significance of each metric more clearly in the Methods section.

(Upon further reading, I noticed similar graphs are included in the supplementary material. In that case, my revised suggestion is to add more explanation in the Methods section regarding why these specific network metrics were chosen and what ecological insights they are intended to provide.)

Line 668 and final analysis: Ending with a simple correlation analysis feels somewhat underwhelming given the depth of your earlier explorations. I encourage you to develop a hypothesis-driven (hypotheses built on your results above) structural equation model (SEM) to synthesize your findings and strengthen the paper's impact. A good example is provided by Chen et al. (2024, *Communications Earth & Environment*), which effectively ties together multiple ecological layers using SEM. For instance, your SEM could include restoration age and soil chemical properties as input variables, microbial attributes (e.g., richness of rare taxa, network properties of abundant taxa) as mediating variables, and ecosystem functions (e.g., multifunctionality, enzyme activity, or plant growth) as outcome variables. Your earlier results suggest that richness of rare taxa and the average degree of abundant taxa may be key mediators, while intermediate taxa could be indirectly linked through soil chemistry. I understand this may require additional work, but I believe this approach would align well with your paper's title on “Divergent multifunctionality contributions” and make your study valuable to a broader ecological and restoration audience.

Results and Discussion Review Comments

L141–165: Consider moving this section to the supplementary materials if you are constrained by word limits. While I've repeatedly asked for more details on how you define abundant, intermediate, and rare microbes, the results here are not central to your main narrative.

L166–167: The trend description is ambiguous. For abundant or intermediate taxa, the increase from year 0 to 10 is evident, but I wouldn't characterize the subsequent trend as a nonlinear increase. Please rephrase for clarity.

L168–170: This sentence is hard to follow and appears internally contradictory. If it doesn't contribute directly to your main points, consider removing it. Alternatively, a clearer version might be: “Although rare taxa include more ASVs across all samples, intermediate ASVs contribute greater richness at the sample level.” Also, avoid the term “alpha-diversity” unless you're using a specific index—just state the name (e.g., richness) for precision.

L173: Comparing contributions to community dissimilarity among the three groups is interesting, but I question the logic of comparing 1 vs. 2, 2 vs. 3, etc. Wouldn't comparisons like 0 vs. 1, 0 vs. 2, ..., 0 vs. 53 be more ecologically informative? That way, you can directly discuss which groups recover early and which remain unrecovered after 53 years.

L175: Include the relevant statistics here or in the figure to substantiate the claim of a significant difference.

L178: Consider using “longer” instead of “greater” for temporal descriptions.

L179: It's unclear how the triangle plot demonstrates temporal variability, and the figure reference may be incorrect. Also, please explain how the triangle plot is constructed—is it based on total beta diversity or some partitioned component?

L184: The current heatmap visualization is ineffective. Please revise the color scheme to better highlight relative abundance differences.

L204–205: Avoid referring only to “abundant and rare subcommunities” here. Omitting “intermediate” can be misleading and imply exclusion from your analysis.

L223: While positive correlations are often interpreted as synergistic interactions, that assumption generally applies within a shared ecological context. In your case, microbes from year 0 and year 53 are unlikely to interact directly. The observed correlations likely reflect shared responses to environmental gradients rather than true biotic interactions. Given your study design, it would be more informative to construct subnetworks for each restoration stage, rather than categorizing nodes by abundance. Supplementary Figures 8 and 9 are more insightful and might be better suited for inclusion in the main text.

L239: With the richness of your dataset, I strongly encourage replacing the final section with a structural equation model (SEM). For example: What are the known effects of SCB (your restoration method) from prior literature? Does it increase soil carbon or vegetation productivity? SEM would allow you to test whether restoration time improves microbial indices indirectly via changes in soil properties or vegetation, and which microbial features best explain ecological functions such as decomposition.

L285–286: The claim “The increase can potentially be attributed to the improvement of vegetation and soil conditions” is exactly the type of hypothesis that SEM could test more rigorously.

L294: Similarly, the idea that patterns are “driven primarily by species-specific ecological niches across restoration stages” can also be empirically evaluated using SEM. It could help confirm whether different ecological processes govern the assembly of the three subcommunities.

L341–342: This statement contradicts your earlier assertion (L309) that abundant taxa, as ecological generalists with broad niches, are shaped mainly by stochastic processes like dispersal limitation. If abundant taxa are insensitive to environmental selection, how can their increase be associated with increased niche differentiation? Please clarify your position.

L350–351: The assumption that abundant taxa are necessarily fast-growing or highly competitive is oversimplified. Fierer (2017) points out substantial variation in traits even among closely located soil microbes. Furthermore, Actinobacteria are often abundant yet are considered stress-tolerant, not necessarily fast-growing or competitive. Please ground your interpretations in trait-based ecology and acknowledge this complexity.

L360: The interpretations in this paragraph are problematic:

1. Your data show significant differences in topological properties (e.g., higher degree for abundant, higher closeness centrality for intermediate, etc.), so the conclusion that they are similar is unsupported.
2. Even if metrics were similar, that doesn't justify the claim that they play equally important roles in maintaining network structure. Concepts like “keystone species” or “modularity” would be more appropriate indicators.
3. Given that the network spans all restoration stages, it's difficult to interpret how node-level metrics (like degree) relate to ecosystem multifunctionality.
4. Finally, the implication that microbes from year 0 and year 53 can interact lacks ecological plausibility.

L379: Acknowledging limitations is useful when you have a clear take-home message. It's meant to help readers understand how broadly your conclusions apply. But in your case, I've raised concerns about your experimental design and whether your analyses are the best way to support your story. So just adding a “limitations” section won't really fix those deeper issues.

Version 3:

Reviewer comments:

Reviewer #3

(Remarks to the Author)

The authors revised the manuscript according to all my comments.

High-abundance rather than rare bacteria dominate diversity patterns, co-occurrence network and ecosystem multifunctionality during a 53-year restoration in a desert

COMMSBIO-24-5478 submitted to *Communications Biology* on 1th Sep 2024

Dear Referees,

We greatly appreciate the time and effort that the editor and the reviewers have spent on handling our manuscript. Thank you very much. We have carefully addressed the questions raised by the reviewers on the previous version, which are now reflected in the revised manuscript that we hereby resubmit.

Below, we have labeled each reviewer's comments, and we address them one by one. Reviewer comments are in regular font with black color while our response is shown in blue color; *Italic texts* refer to new text added in the manuscript. To benefit review, we provide the main text of the paper with a tracking change version to show where we make revision. We also provide a clean version of the main text. Note that line number within reviewer comments refer to the original manuscript, while line numbers in our reply refer to the clean version of the manuscript resubmitted.

Referee expertise:

Referee #1: soil restoration and omics

Referee #2: soil microbiology

Reviewers' Comments:

Reviewer #1 (Remarks to the Author):

The authors present a study using 16S sequencing of 55 samples from 11 sites, which have been treated with straw checkerboard barriers (SCBs) over the past 53 years, to explore whether rare or abundant microbial taxa are more critical in driving ecosystem restoration. While I commend the authors for their data analysis skills and the amount of information they were able to extract from a relatively small dataset, I have several concerns that suggest the study is overstated in its current form and requires significant further clarification and justification.

1. Title and Scope: The title of the manuscript overstates the scope of the study, particularly given that it focuses on a localized area. As a study conducted on a local scale, it is essential to clearly specify the geographical context (e.g., include the location or refer to "arid ecosystems"). While small-scale studies certainly have value, the high-impact language used in the title is not aligned with the scale of the study. I strongly recommend tempering the title and framing the findings within a more appropriate context.

>> We thank you for pointing out the missing information. We have revised the title and made the scope of the study clearer in the manuscript. Please see the title.

Title *"High-abundance rather than rare bacteria dominate diversity patterns, co-occurrence network and ecosystem multifunctionality during a 53-year restoration in a desert"*

2. Interpretation of Results and Causality: A key concern is the interpretation of the results. One of the main research questions posed by the authors is whether rare taxa play a more significant role than abundant taxa in driving ecosystem restoration. However, the manuscript is based on correlation analyses, and the use of language that implies causality is misleading. Correlation does not equate to causation, and the experimental design and statistical analysis employed here do not support the claim that "high-abundance taxa drive a greater number of ecosystem functions through their interactions," nor does it demonstrate whether ecosystem functions drive the interactions of high-abundance taxa. A more cautious interpretation of these findings is necessary.

>> Thank you for pointing out this issue. We completely agree with your perspective. In the revised version, we have adjusted the language and now use "relationship", "correlation" and "associated" (or similar) instead of "effect" and "drive". For example, please see Lines 14–18 in Abstract.

Lines 15–19 *"Notably, we uncovered a dual mechanism underlying the relationships between soil bacterial communities and ecosystem multifunctionality. High-abundance taxa were associated with multiple nutrient cycling-related functions simultaneously through interspecies connections, whereas rare taxa were linked to individual functions independently."*

3. Study Design and Restoration Approach: A detailed description of the study site and restoration methods is lacking. Given that the study employs a space-for-time substitution

design (which is technically a pseudoreplication), it is possible that the observed patterns are driven more by the proximity of the sites than the restoration method itself. Additionally, it is important to reference previous research on straw checkerboard barriers, including their effects on soil properties and vegetation communities, to help readers understand their potential ecological impacts. This would provide a stronger foundation for hypothesizing about the role of microbes in ecosystem restoration.

>> Thanks for your comments. We have added more detailed information describing the study sites and restoration methods. Please see Lines 325–328 for the description of the study site, Lines 331–336 for the description of the straw checkerboard barriers (SCBs) and the vegetation at the study site, Lines 337–351 for the description of experimental design and sample collection. We fully agree on the advantage of long-term succession studies in the same plot. However, the space-for-time substitution approach is commonly used in chronosequence studies of vegetation succession and soil development, due to the difficulty in obtaining long-term continuous tracking empirical data in the same plot. This information has now been highlighted. Please see Lines 352–358. In addition, we have added more information about relevant research on straw checkerboard barriers in the Introduction. Please see Lines 72–78.

Lines 325–328 *“This region is seriously affected by land degradation and desertification caused by wind erosion and is therefore defined as a priority area for implementing ecological conservation and restoration projects in China³⁷.”*

Lines 331–336 *“In practice, the establishment of SCBs involves arranging wheat straw in a checkerboard pattern composed of numerous squares in mobile sandy land, with one half of straw buried in the sand and the other half exposed⁴¹. Plant communities on mobile sandy land are dominated by *Agriophyllum squarrosum*, with a cover of around 2%, while plant communities in non-desertified natural ecosystems are dominated by *Artemisia ordosica*, with a cover of around 45%⁴³.”*

Lines 337–351 *“In this study, a well-documented restoration chronosequence spanning 53 years through the establishment of SCBs was identified. Field data were collected between August and September 2017 from a mobile sandy land plot with a restoration duration of 0 years (i.e., an unrestored desertified ecosystem) and 10 plots restored by SCB treatment, each with different restoration durations (i.e., 1, 3, 4, 5, 6, 11, 20, 23, 26 and 53 years)⁴³. Specifically, a 12 × 12 m quadrat was established in each plot. In each quadrat, five 12-m-long transects (spaced 1.2 m apart) were arranged within to harvest above-ground plant materials following*

the line-intercept protocol; five 1.2 × 1.2 m subplots (spaced at least 4 m apart) were randomly selected to collect soil samples^{43,61}. In each subplot, 5–7 soil cores (0–10 cm depth) were collected randomly, and then thoroughly mixed to form a composite sample in the field. A total of 55 soil samples were collected from the 11 quadrats. The samples were sieved (< 2 mm) and divided into three parts. One part was dried for the determination of physicochemical properties, another part was stored at 4°C for measuring enzyme activities, and the last part was stored at –80°C for DNA extraction (within one week).”

Lines 352–358 “Given the challenge of obtaining long-term data from the same plots, the space-for-time substitution approach is widely employed in temporal studies of ecological succession, and is generally regarded as reliable and scientifically sound^{62,63}. Moreover, in our study, all plots shared similar initial conditions and environmental settings, with the age of the established SCBs serving as the dominant factor driving changes in soil bacterial subcommunities⁴³. Therefore, here we utilized the space-for-time substitution approach to compare plots with different restoration durations.”

Lines 72–78 “Among various restoration measures, straw checkerboard barriers (SCBs) have proven highly effective and are widely acknowledged restoration measure for their ability to stabilize shifting sand and reduce wind erosion^{40,41}. In addition, SCBs can also promote soil nutrient accumulation and facilitate the development of biocrusts⁴². A recent study has shown that the establishment of SCBs triggers temporal changes in the composition and assembly processes of plant and soil microbial communities⁴³.”

References:

37. Li, C. et al. Drivers and impacts of changes in China’s drylands. *Nat. Rev. Earth Environ.* 2, 858–873 (2021).
40. Li, X. R., Xiao, H. L., He, M. Z. & Zhang, J. G. Sand barriers of straw checkerboards for habitat restoration in extremely arid desert regions. *Ecol. Eng.* 28, 149–157 (2006).
41. Zhang, S. et al. Effect of Straw Checkerboards on Wind Proofing, Sand Fixation, and Ecological Restoration in Shifting Sandy Land. *Int. J. Environ. Res. Public Health* 15, 2184 (2018).
42. Li, X. R., Kong, D. S., Tan, H. J. & Wang, X. P. Changes in soil and vegetation following stabilisation of dunes in the southeastern fringe of the Tengger Desert, China. *Plant Soil* 300, 221–231 (2007).

43. Hou, Q. et al. Active restoration efforts drive community succession and assembly in a desert during the past 53 years. *Ecol. Appl.* e3068 (2024).

61. Maestre, F. T. et al. Plant Species Richness and Ecosystem Multifunctionality in Global Drylands. *Science* 335, 214–218 (2012).

62. Walker, L. R., Wardle, D. A., Bardgett, R. D. & Clarkson, B. D. The use of chronosequences in studies of ecological succession and soil development. *J. Ecol.* 98, 725–736 (2010).

63. Hasselquist, E. M. et al. Time for recovery of riparian plants in restored northern Swedish streams: a chronosequence study. *Ecol. Appl.* 25, 1373–1389 (2015).

4. Definition of Abundant, Intermediate, and Rare Taxa: The classification of taxa into abundant, intermediate, and rare categories needs further justification. While I appreciate the authors' efforts to reduce arbitrariness, the current definitions appear biologically inconsistent. Several issues arise:

The categorization is based on the average relative abundance of OTUs across all samples. This approach could misclassify OTUs that are rare in early successional stages but become dominant later—these taxa are potentially the most ecologically significant, yet they are likely categorized as intermediate.

>> We thank you for pointing out this. In the previous manuscript, we did not consider these conditionally abundant/rare OTUs (i.e., OTUs that are abundant and rare in different samples). According to your comments, we have identified conditionally abundant/rare taxa in the revised manuscript. Please see Lines 390–395 in the Methods. We described the results of the identification of the taxa. Please see Lines 115–119 in the Results and Supplementary Fig. 3. In our study, all conditionally abundant/rare OTUs belonged to high-abundance taxa. Therefore, we believe that our categorization approach based on the average relative abundance of OTUs was unlikely to bias our main conclusions in the manuscript. We have highlighted this information as a limitation in the Discussion and suggested that future studies should focus on different types of abundant and rare taxa during restoration. Please see Lines 273–282 in the Discussion.

Lines 390–395 “Furthermore, considering the variations in OTUs across samples, we identified conditionally abundant/rare taxa (i.e., OTUs that are abundant and rare in different samples) on a per-sample basis. To avoid arbitrary classifications, we employed the “sample-specific

rarity cutoff” approach as described by Jia et al.²⁹, setting a rarity cutoff value of 0.3% at the sample level. OTUs with a relative abundance > 0.3% in some samples but < 0.3% in others were defined as conditionally abundant/rare taxa.”

Lines 115–119 “We further identified 355 conditionally abundant/rare OTUs based on a rarity cutoff of 0.3% at the sample level, which were rare during certain successional stages but abundant in others. All conditionally abundant/rare OTUs were classified within high-abundance taxa, with 61.13% categorized as abundant taxa and 38.97% as intermediate taxa (Supplementary Fig. 3).”

Lines 273–282 “Firstly, our categorization approach, based on the average relative abundance of OTUs, may classify conditionally abundant/rare OTUs as intermediate taxa, possibly leading to an overestimation of the performance of high-abundance taxa. Although conditionally abundant/rare OTUs are widespread across all samples, our results confirmed that all of those OTUs belong to high-abundance taxa (see Supplementary Fig. 3). Therefore, we expected the average relative abundance-based categorization approach was less likely to bias our main conclusions, within the context of our datasets. However, given the potential ecological significance of conditionally abundant/rare OTUs, future studies should consider more nuanced classifications of abundant and rare taxa during restoration²⁹.”

Supplementary Figure 3. Conditional abundant/rare taxa identified using a rarity cutoff value

of 0.3%. A total of 355 conditionally abundant/rare OTUs were identified, with 217 classified as abundant taxa and 138 as intermediate taxa.

References:

29. Jia, X., Dini-Andreote, F. & Salles, J. F. Unravelling the interplay of ecological processes structuring the bacterial rare biosphere. *ISME Commun.* 2, 96 (2022).

OTUs are not a taxonomic level, and the comparison of an order with a species can be problematic, as orders would naturally be more abundant than species. This discrepancy should be addressed by standardizing the analysis to a common taxonomic level.

>> Thanks for your comments. We understand your concern regarding the use of OTUs (operational taxonomic units). However, OTUs, referred to as phylotypes in some papers, offer a practical approach for identifying microbial communities based on sequence similarity, especially when many microorganisms remain unclassified. In this study, all available sequences were assigned to OTUs at a 97% similarity threshold, allowing for the comparison of microbial diversity and community composition across samples under consistent criteria. This threshold has been widely adopted in microbiological studies. We acknowledge that whenever the data are available, more accurate methods should be preferred.

Rarefying data can bias results against rare taxa, and I am not convinced it is appropriate in this context. Additionally, singletons could be considered rare taxa, but they seem to be excluded from the analysis, further biasing the results.

>> Thanks for your comments. In our study, the OTU table was rarefied (resampled) to a lowest number of sequences found within an individual sample. This resampling was performed to ensure an equal sampling depth. Additionally, we excluded singletons (those appearing only once in the entire dataset) from the raw sequences, as they may have been caused by sequencing errors. This step helps to reduce noise and enhance the accuracy of results. After the raw sequences were assigned to OTUs and resampling, singletons were not excluded further. We have revised these sentences for clarity. Please see Lines 379–381, Lines 375–377.

Lines 379–381 “*To ensure equal sampling depth⁶⁴, the OTU table was then rarefied to the lowest number of sequences (22,908) found within an individual sample, resulting in a*

resampled dataset of 10,378 bacterial OTUs.”

Lines 375–377 *“All available sequences were assigned to operational taxonomic units (OTUs) at 97% similarity after excluding singletons.”*

References:

64. Hu, W. et al. Aridity-driven shift in biodiversity–soil multifunctionality relationships. *Nat. Commun.* 12, 5350 (2021).

5. Network Analysis and Rare/Abundant Taxa: I also question the suitability of network analysis for examining abundant and rare taxa. Network analysis is based on correlations between taxa, and the patterns observed could be driven by shared environmental factors that shape taxa distributions within the same sample, rather than by ecological interactions. The occurrence of abundant taxa in more samples naturally lead to stronger correlations, but this does not necessarily reflect biologically meaningful interactions. Therefore, the application of network analysis in this context requires more careful consideration and a discussion of its limitations.

>> We thank you for pointing out this important issue. We agree with you. While microbial co-occurrence networks have become a powerful tool to infer interspecies connections, we also acknowledged that they are essentially correlated. For clarity, we have revised the language and now use “interspecific connections” instead of “species interaction” in the revised manuscript. In the Discussion, we have highlighted the limitations of this approach and suggested that further studies should utilize cutting-edge omics technologies, and consider integrating co-occurrence networks and metabolic networks to infer more robust interactions among microbial species and underlying mechanisms. Please see Lines 290–300.

Lines 290–300 *“Lastly, while microbial co-occurrence networks have proven to be a powerful tool for inferring interspecies connections⁵⁵, we also acknowledged that they are inherently correlated, often reflecting common selection patterns driven by shared environmental gradients, rather than providing conclusive evidence of actual ecological interactions^{56,57}. In light of this, recent researches have highlighted the potential of exploring species interactions through the identification of metabolic exchanges and species-specific resource requirements within complex microbial communities⁵⁸. To achieve this, further studies should utilize cutting-edge omics technologies, and consider integrating co-occurrence and*

metabolic networks to infer more robust interactions among microbial species and elucidate the potential underlying ecological mechanisms^{59,60}.”

References:

55. Barberán, A., Bates, S. T., Casamayor, E. O. & Fierer, N. Using network analysis to explore co-occurrence patterns in soil microbial communities. *ISME J.* 6, 343–351 (2012).

56. Blanchet, F. G., Cazelles, K. & Gravel, D. Co-occurrence is not evidence of ecological interactions. *Ecol. Lett.* 23, 1050–1063 (2020).

57. Faust, K. Open challenges for microbial network construction and analysis. *ISME J.* 15, 3111–3118 (2021).

58. Zelezniak, A. et al. Metabolic dependencies drive species co-occurrence in diverse microbial communities. *Proc. Natl. Acad. Sci. U.S.A* 112, 6449–6454 (2015).

59. Peng, X. et al. Metabolic interdependencies in thermophilic communities are revealed using co-occurrence and complementarity networks. *Nat. Commun.* 15, 8166 (2024).

60. Cardona, C., Weisenhorn, P., Henry, C. & Gilbert, J. A. Network-based metabolic analysis and microbial community modeling. *Curr. Opin. Microbiol.* 31, 124–131 (2016).

Conclusion: In summary, while the authors have presented a technically sound analysis, the manuscript overstates its findings, and significant revisions are necessary before it can be considered for publication. Clarification of the study design, justification of taxa definitions, and a more cautious interpretation of the results are essential. Additionally, the title and claims made in the manuscript should be tempered to better reflect the study’s local scale and correlational nature.

>> Thanks for your constructive suggestions on our manuscript. According to your suggestions, we have thoroughly revised the manuscript, particularly the Method and Discussion. Please see the responses above. Thank you again. We believe that these revisions will significantly enhance our manuscript. We appreciate your guidance and look forward to any further feedback you may have.

Reviewer #2 (Remarks to the Author):

This study investigates successional dynamics over a 53-year restoration chronosequence following the implementation of straw checkerboard barriers (SCB) in the Tegger Desert, China. The authors demonstrate that SCB increased the diversity of abundant taxa over time, with negligible effects on rare taxa. Their findings emphasize the pivotal role of abundant taxa in shaping bacterial community composition, as well as enhancing the complexity and stability of co-occurrence networks. Variable selection emerged as the predominant mechanism driving the assembly of both abundant and rare taxa. Notably, the study highlights that abundant taxa facilitated multiple nutrient cycling functions through interactions, while rare taxa contributed independently to single functions.

I commend the authors for undertaking the challenging task of analyzing the dynamics of rare taxa, defined here as taxa with low abundance and limited representation across samples in amplicon datasets. This study reflects significant effort, from meticulous sampling for amplicon sequencing to the thorough characterization of environmental and functional soil properties.

>> We thank you for your positive comments on our work and the importance of the findings reported in our manuscript, as well as your critical view on the manuscript and suggestions for improvement. We have responded to your comments point by point. Please see the responses below.

That said, one aspect that detracts from the study's value is the reliance on predictive databases (FAPROTAX) to infer functional capacities from OTUs. While I acknowledge the economic and technical challenges associated with metagenomic or metatranscriptomic analyses, predictive software introduces inherent limitations that may reduce the robustness of the functional inferences presented in the manuscript. This reviewer would rather remove these predictions and focus on the amplicon data and the rest of the analyses.

>> We thank you for pointing out this. We recognized the limitations of using predictive databases to infer the functional potential of OTUs. In the revised manuscript, we have removed this method and related content.

Additionally, analyzing rare taxa requires substantial sequencing depth, yet the authors do not

provide any statistics on sequencing depth. To address this, the authors should include accumulation (rarefaction) curves for highly abundant, mid-abundant, and rare taxa to ensure proper diversity representation. Without this analysis, the observed lack of association between rare taxa and multifunctionality could be attributed to poor representation of rare communities. Such underrepresentation could cascade across all analyses, including the co-occurrence-based network analysis. The authors mention that rare taxa appear in only 10% of the samples, which they link to the low connectivity of this group in the reconstructed network. However, this claim warrants further support through the inclusion of diversity plateau evidence.

>> Thank you for pointing out the missing information. We have now provided rarefaction curves for the entire dataset, abundant, intermediate, and rare taxa (please see Supplementary Figure 2). This result was described in Lines 107–110. We found that rarefaction curves of most of our samples reached a steady plateau, indicating that the great majority of bacterial OTUs were recovered. However, the rarefaction curves of some rare subcommunities were less saturated (please see Supplementary Fig. 2d). This suggests that the diversity of rare taxa was not fully captured in our samples, which could partly weaken the significance of rare taxa. We have highlighted this information in the Discussion as a limitation of our study and expected future studies would improve the detection of rare taxa by increasing sequencing depth. Please see Lines 282–290.

Lines 107–110 “*The rarefaction curves for most of our samples reached a steady plateau, indicating that the majority of bacterial operational taxonomic units (OTUs) were successfully recovered, providing comprehensive coverage of the bacterial richness in our dataset (Supplementary Fig. 2).*”

Lines 282–290 “*Secondly, we acknowledged that detection of rare taxa is largely depended on sequencing depth. While the pooled data and most of the abundant and intermediate subcommunities showed full rarefaction saturation, the rarefaction curves of some rare subcommunities were less saturated (see Supplementary Fig. 2). This suggests that the diversity of rare taxa was not fully captured in our samples, which could undercut their perceived significance in our findings. Acknowledging this, future studies should aim to increase sequencing depth to improve the detection of rare taxa, despite the inherent challenges in fully characterizing these taxa¹².*”

Supplementary Figure 2. Rarefaction curves of individual samples. a Entire dataset. b Abundant subcommunity. c Intermediate subcommunity. d Rare subcommunity. Lines with different colors indicate samples with different restoration durations.

References:

12. Sogin, M. L. et al. Microbial diversity in the deep sea and the underexplored “rare biosphere”. *Proc. Natl. Acad. Sci. U.S.A* 103, 12115–12120 (2006).

The authors state in the abstract and discussion that high- and mid-abundance taxa contribute to network stability, while rare taxa do not. However, network stability was not directly measured. Instead, only certain network topological properties, such as degree, were assessed and correlated with multifunctionality. Clarifying and justifying the connection between network stability and the selected topological metrics is essential to support the manuscript’s conclusions.

>> Thanks for your suggestion. In the previous manuscript, we only evaluated the topological properties of each node in the network, including degree, closeness centrality, betweenness centrality and clustering coefficient. Nodes with higher degree, closeness centrality, betweenness centrality and clustering coefficient are regarded as key hubs for connection and information transfer, playing a crucial role in maintaining the complexity and functionality of

the network. However, based on this network alone, we cannot clarify the connection between these node-level topological properties and network stability. As a supplement, we constructed networks for abundant, intermediate, and rare taxa separately, and compared their stability in the revised manuscript. Please see Lines 492–496 and Supplementary Figure 7. We found that abundant and intermediate taxa exhibited higher network robustness (stability) compared with rare taxa. Please see Lines 164–166. We expected that newly added information will further support our main conclusions.

Lines 492–496 “As a supplement, we constructed networks for abundant, intermediate, and rare taxa separately, and compared their stability. The stability of the network was characterized by robustness, which is defined as the proportion of remaining nodes after removing 50% of random nodes at 1000 iterations^{79,80}.”

Lines 164–166 “By constructing the networks separately, we further found that abundant and intermediate taxa exhibited higher network robustness compared to rare taxa (Supplementary Fig. 7).”

Supplementary Figure 7. Separate networks and their robustness. a–c Separate networks for abundant, intermediate, and rare taxa. n , the number of nodes; L , the number of edges. The size of each node is proportional to its degree. d Boxplots of the robustness of separate networks for abundant, intermediate, and rare taxa at 1000 iterations. The different letters represent significant differences determined by the Kruskal–Wallis test followed by Dunn post-hoc test (Bonferroni correction).

References:

79. Yuan, M. M. et al. Climate warming enhances microbial network complexity and stability. *Nat. Clim. Change* 11, 343–348 (2021).
80. Wu, H., Gao, T., Hu, A. & Wang, J. Network complexity and stability of microbes enhanced by microplastic diversity. *Environ. Sci. Technol.* 58, 4334–4345 (2024).

Turning to methodology, several key aspects lack sufficient detail. For instance, soil sampling—a critical component of the study—was relegated to the supplementary materials. These details should be brought back into the main text. Similarly, the network reconstruction strategy is not fully described. The authors do not specify the software used, or the rationale behind the selection of the correlation threshold used for network reconstruction. Moreover, only one network combining the three groups (high, mid-abundance, and rare taxa) was reconstructed. This raises questions about the data underlying the boxplots in Figure 5. Were the networks randomized to calculate these metrics, or were multiple networks reconstructed under different conditions to enable replication? Or was centrality and other parameter calculate per node and used for comparison? Including this information is essential for the reproducibility of the study.

>> Thank you for pointing out the missing information. In the revised manuscript, we have included detailed descriptions of the study site and soil sampling. Please see Lines 325–328 for the description of the study site, Lines 331–336 for the description of the straw checkerboard barriers (SCBs) and the vegetation at the study site, Lines 337–351 for the description of experimental design and sample collection. Additionally, in the statistical analyses, we have included a detailed description of network construction and analysis. The network was visualized using Gephi software (v.0.9.2). We calculated all possible Spearman's correlations (ρ) among OTUs and retained robust correlations with absolute values of Spearman's

correlation coefficients (ρ) > 0.75 and FDR-corrected $P < 0.001$. We expected these thresholds to reveal organisms that were strongly correlated with each other. Please see Lines 474–479. To evaluate the importance of nodes within the network, we calculated the topological properties for each node, including degree, closeness centrality, betweenness centrality and clustering coefficient, using the R package “igraph”. We assessed the differences in these properties among abundant, intermediate, and rare nodes using the Kruskal–Wallis test with a Dunn post-hoc test (Bonferroni correction). Please see Lines 479–482, Lines 490–492.

Lines 325–328 “*This region is seriously affected by land degradation and desertification caused by wind erosion and is therefore defined as a priority area for implementing ecological conservation and restoration projects in China*³⁷.”

Lines 331–336 “*In practice, the establishment of SCBs involves arranging wheat straw in a checkerboard pattern composed of numerous squares in mobile sandy land, with one half of straw buried in the sand and the other half exposed*⁴¹. *Plant communities on mobile sandy land are dominated by *Agriophyllum squarrosum*, with a cover of around 2%, while plant communities in non-desertified natural ecosystems are dominated by *Artemisia ordosica*, with a cover of around 45%*⁴³.”

Lines 337–351 “*In this study, a well-documented restoration chronosequence spanning 53 years through the establishment of SCBs was identified. Field data were collected between August and September 2017 from a mobile sandy land plot with a restoration duration of 0 years (i.e., an unrestored desertified ecosystem) and 10 plots restored by SCB treatment, each with different restoration durations (i.e., 1, 3, 4, 5, 6, 11, 20, 23, 26 and 53 years)*⁴³. *Specifically, a 12 × 12 m quadrat was established in each plot. In each quadrat, five 12-m-long transects (spaced 1.2 m apart) were arranged within to harvest above-ground plant materials following the line-intercept protocol; five 1.2 × 1.2 m subplots (spaced at least 4 m apart) were randomly selected to collect soil samples*^{43,61}. *In each subplot, 5–7 soil cores (0–10 cm depth) were collected randomly, and then thoroughly mixed to form a composite sample in the field. A total of 55 soil samples were collected from the 11 quadrats. The samples were sieved (< 2 mm) and divided into three parts. One part was dried for the determination of physicochemical properties, another part was stored at 4°C for measuring enzyme activities, and the last part was stored at –80°C for DNA extraction (within one week).*”

Lines 474–479 “*All possible Spearman’s correlations (ρ) among OTUs were calculated, and only robust correlations with absolute values of Spearman’s correlation coefficients (ρ) >*

0.75 and FDR-corrected $P < 0.001$ were retained. We expected these thresholds to reveal organisms that were strongly correlated with each other^{43,74,75}. In the network, nodes represented bacterial OTUs, and edges represented Spearman's correlations among nodes. The network was visualized using Gephi software (v.0.9.2).”

Lines 479–482 “To evaluate the importance of nodes within the network, we calculated the key topological properties for each node, including degree, closeness centrality, betweenness centrality and clustering coefficient, using the R package “igraph”^{50,75,76}.”

Lines 490–492 “Subsequently, we assessed the differences in these topological properties among abundant, intermediate, and rare nodes using the Kruskal–Wallis test with a Dunn post-hoc test (Bonferroni correction).”

References:

37. Li, C. et al. Drivers and impacts of changes in China's drylands. *Nat. Rev. Earth Environ.* 2, 858–873 (2021).
41. Zhang, S. et al. Effect of Straw Checkerboards on Wind Proofing, Sand Fixation, and Ecological Restoration in Shifting Sandy Land. *Int. J. Environ. Res. Public Health* 15, 2184 (2018).
43. Hou, Q. et al. Active restoration efforts drive community succession and assembly in a desert during the past 53 years. *Ecol. Appl.* e3068 (2024).
50. Xue, Y. et al. Distinct patterns and processes of abundant and rare eukaryotic plankton communities following a reservoir cyanobacterial bloom. *ISME J.* 12, 2263–2277 (2018).
61. Maestre, F. T. et al. Plant Species Richness and Ecosystem Multifunctionality in Global Drylands. *Science* 335, 214–218 (2012).
74. Yang, Y. et al. Deciphering factors driving soil microbial life-history strategies in restored grasslands. *iMeta* 2, e66 (2023).
75. Zhang, Y. et al. Vertical diversity and association pattern of total, abundant and rare microbial communities in deep-sea sediments. *Mol. Ecol.* 30, 2800–2816 (2021).
76. Jiao, S., Wang, J., Wei, G., Chen, W. & Lu, Y. Dominant role of abundant rather than rare bacterial taxa in maintaining agro-soil microbiomes under environmental disturbances. *Chemosphere* 235, 248–259 (2019).

Another methodological concern is the statistical approach. The authors use the Mann-Whitney U test to compare three groups, but this test is designed for pairwise comparisons. Using it to compare three groups would require conducting multiple pairwise comparisons, which could inflate the Type I error rate (false positives) unless corrected (e.g., using a Bonferroni adjustment). Even with corrections, this approach is less efficient and not ideal for multi-group comparisons. A more appropriate statistical test, such as Kruskal-Wallis with a post-hoc analysis, should be applied to allow comparisons across all three groups.

>> We thank you for noticing this error. In the revised manuscript, we have adopted the Kruskal-Wallis test with a Dunn post-hoc test (Bonferroni correction) to compare the diversity, niche breadth, and node-level topological properties of abundant, moderate, and rare subcommunities. We used different letters to represent significant differences among the three groups. For example, please see Fig. 1f, g.

Figure 1. Definition and composition of abundant, intermediate, and rare taxa. a Cutoff values for defining abundant, intermediate, and rare taxa based on the average relative abundance across all samples. b, c Number of OTUs and relative abundance for the three biological groups across all samples. d The proportion of samples occupied for each of abundant, intermediate, and rare OTUs. e Taxonomic distribution of the three biological groups at phylum level. The thickness of each ribbon in the circos plot represents the relative abundance of the three biological groups assigned to different phyla. f, g Boxplots of the niche breadth for each of abundant, intermediate, and rare OTUs estimated using the Levins' index and the tolerance index, respectively. The different letters above the boxes represent significant differences

determined by the Kruskal–Wallis test followed by Dunn post-hoc test (Bonferroni correction).

In conclusion, this study has the potential to contribute meaningfully to the field of microbial ecology. However, addressing the methodological and analytical concerns outlined above is necessary to enhance the rigor and reliability of the findings.

>> Thanks for your constructive suggestions on our manuscript. According to your suggestions, we have thoroughly revised the manuscript, particularly the Method and Discussion. Please see the responses above. Thank you again. We believe that these revisions will significantly enhance our manuscript. We appreciate your guidance and look forward to any further feedback you may have.

Specific comments:

Lines 124-125, are confusing in the current format – authors seem to provide redundant information regarding the prevalence of rare taxa.

>> Thank you for pointing out this issue. That sentence has now been revised. Please see Lines 132–134.

Lines 132–134 “NMDS ordination and ANOSIM based on Bray–Curtis distance revealed distinct differences in the composition of abundant, intermediate and rare subcommunities among samples with varying restoration durations (Fig. 2d–f).”

Line 168 – define the network properties – i.e degree is the measure of the number of links associated with a node, etc.

>> Thanks for your suggestion. In the revised manuscript, we have included definitions of node-level topological properties in the network. Please see Lines 482–487.

Lines 482–487 “Degree refers to the number of edges connected to a node. Closeness centrality is the reciprocal of the average shortest path length from one node to all other nodes. Betweenness centrality quantifies the number of shortest paths passing through a node. Clustering coefficient measures the ratio of the actual connections to all possible connections between the adjacent nodes of a node.”

Improve methods

>> Thanks for your suggestion. We have thoroughly revised the Methods and added more detailed descriptions of the sampling, experimental design, and statistical analyses. Please refer to the response above.

Figure 6. Legend must include a definition of the multifunctionality metrics – what is PC, PN, PP, etc.

>> We thank you for noticing this missing information. We have now added the information. Please see Fig. 6.

Figure 6. Relationships of diversity of abundant, intermediate, and rare taxa with

multifunctionality and individual functions. a Relationships between multifunctionality and diversity of abundant, intermediate, and rare taxa. The black fitted lines are from linear regression. The solid and dotted lines represent statistically significant ($P \leq 0.05$) and nonsignificant ($P > 0.05$) relationships, respectively. b Pearson's correlation coefficients between individual functions and diversity of abundant, intermediate, and rare taxa. Edge width corresponds to the Pearson's r value. Positive and negative correlations are indicated by red and black edges, respectively. Pairwise correlations of these functions are displayed with a color gradient denoting Pearson's correlation coefficient. AGB, aboveground biomass; PC, plant carbon; PN, plant nitrogen; PP, plant phosphorus; DNA, soil DNA concentration; Suc, soil sucrose activity; β -Glu, soil β -glucosidase activity; Cel, soil cellulase activity; Ure, soil urease activity; ALP, soil alkaline phosphatase activity; Cat, soil catalase activity; SOC, soil organic carbon; SNH_4^+ -N, soil ammonium; SNO_3^- -N, soil nitrate; STN, soil total nitrogen; SAP, soil available phosphorus; STP, soil total phosphorus.

Divergent community assembly processes and multifunctionality contributions of abundant and rare soil bacteria during a 53-year restoration in the Tengger Desert, China

COMMSBIO-24-5478A submitted to *Communications Biology*

Dear Referees,

We sincerely appreciate the time and effort dedicated by the editor and reviewers in evaluating our manuscript. We have carefully addressed the key concerns re-highlighted by the reviewers through additional analyses, which are now incorporated into the revised manuscript that we hereby resubmit. A critical update involves replacing operational taxonomic units (OTUs) with amplicon sequence variants (ASVs) for higher resolution, particularly in detecting rare taxa. While this adjustment led to minor changes in some results (e.g., α -diversity and assembly processes of rare taxa), our core conclusion—that abundant and rare taxa mediate ecosystem multifunctionality via distinct mechanisms—remains robust. Please see detailed response below.

Below, we have labeled each reviewer's comments, and we address them one by one. Reviewer comments are in regular font with black color while our response is shown in blue color; *Italic texts* refer to new text added in the manuscript. To benefit review, we provide the main text of the paper with a tracking change version to show where we make revision. We also provide a clean version of the main text. Note that line number within reviewer comments refer to the original manuscript, while line numbers in our reply refer to the clean version of the manuscript resubmitted.

Reviewers' Comments:

Reviewer #1 (Remarks to the Author):

I appreciate the authors' effort in refining their manuscript by moderating their claims, adding details about the experimental design, and addressing limitations. However, the major concerns I raised in the previous review still require substantial revisions through additional analyses or supporting materials rather than just wording adjustments.

1. Justification of pseudo-replication

A detailed experimental design map is needed to properly address concerns about pseudo-

replication. While you have provided more details on the sampling plan and the known effects of straw checkerboard barriers (SCB) on soil properties, your justification that "the space-for-time substitution approach is widely employed in temporal studies of ecological succession and is generally regarded as reliable and scientifically sound" is more applicable to plant succession than to soil microbial communities, given the different spatial scales involved.

As I suggested previously, please include a map that clearly illustrates the spatial scale of your study. Additionally, I recommend consulting Soil DNA chronosequence analysis shows bacterial community re-assembly following post-mining forest rehabilitation by Peddle et al. (Figure 5) as an example of how to demonstrate that observed microbial community variance is not merely an artifact of spatial distance.

>> We thank you for pointing out the missing information. We have provided maps of study areas and sampling plots. Please see Supplementary Fig. 1. It should be noted that in our restoration chronosequence, the 53-year restoration plot is geographically separated from the main sampling region, but remained within the same study area (southern edge of the Tengger Desert) under consistent environmental conditions. This information has now been highlighted in the Methods. Please see Lines 376–381. To test the effects of spatial autocorrelation and restoration duration, we evaluated the relationships between bacterial community composition (including entire dataset, abundant, intermediate, and rare subcommunities based on Bray-Curtis distance) and geographical distance and restoration time interval between samples using the Mantel test based on your suggestions. Our results showed that bacterial community composition was significantly correlated with time interval, but not with geographical distance. Please see Supplementary Table 1. This means that the observed variations in bacterial communities were primarily determined by restoration duration and were not affected by spatial autocorrelation, despite one plot being geographically separate from the others. This result, along with the two reasons mentioned in the previous version, had been used to explain our rational application of the space-for-time substitution approach. Please see Lines 423–438.

Lines 376–381 *"It should be noted that to obtain a longer chronosequence, the 53-year restoration plot was located in the Shapotou restoration area, the earliest site in China to implement SCBs for sand fixation. The other restored plots were situated within the main sampling area of Jingtai County. Nevertheless, all study plots were positioned along the southern edge of the Tengger Desert, ensuring consistent environmental conditions."*

Lines 423–438 *"In this study, we strategically utilized the space-for-time substitution*

approach to assess ecological dynamic across plots with varying restoration durations for the following compelling reasons. First, given the inherent difficulty of obtaining long-term data from the same plot, the space-for-time substitution approach is a widely acceptable and scientifically credible alternative in temporal studies of ecological succession^{81,82}. Second, all selected plots shared similar treatment, initial conditions and environmental settings⁴⁴. Finally, we evaluated the relationships between bacterial community composition (including entire dataset as well as abundant, intermediate, and rare taxa) based on Bray–Curtis distance, geographic distance, and restoration time interval between samples using the Mantel test to evaluate spatial autocorrelation and time effects (Supplementary Table 1)⁸³. This result indicated that the bacterial community composition was significantly correlated with restoration time interval, but not with geographical distance (Supplementary Table 1). This suggests that the observed variations in bacterial communities were primarily determined by the age of the established SCBs, rather than by spatial autocorrelation, despite the 53-year restoration plot in Shapotou was spatially separated from others.”

Supplementary Figure 1. Maps of study area and plots with different restoration durations. **a** The study area (within the red box) is located on the southern edge of the Tengger Desert in northwest China. **b** White circles and numbers indicate plots with different restoration duration (years). It should be noted that a 53-year restoration plot is geographically separated from these plots and is not labeled here.

Supplementary Table 1. Mantel test between bacterial community composition and geographical distance and time interval.

Dataset		Geographical distance		Time interval	
		r	P	r	P
Original data	Entire	0.12	0.169	0.49	0.001

	Abundant	0.01	0.318	0.42	0.001
	Intermediate	0.10	0.277	0.48	0.001
	Rare	0.07	0.292	0.39	0.001
	Entire	0.12	0.188	0.50	0.001
Rarefied data	Abundant	0.02	0.282	0.42	0.001
	Intermediate	0.10	0.283	0.49	0.001
	Rare	0.07	0.272	0.40	0.001

References:

44. Hou, Q. et al. Active restoration efforts drive community succession and assembly in a desert during the past 53 years. *Ecol. Appl.* **35**, e3068 (2025).
81. Walker, L. R., Wardle, D. A., Bardgett, R. D. & Clarkson, B. D. The use of chronosequences in studies of ecological succession and soil development. *J. Ecol.* **98**, 725–736 (2010).
82. Hasselquist, E. M. et al. Time for recovery of riparian plants in restored northern Swedish streams: a chronosequence study. *Ecol. Appl.* **25**, 1373–1389 (2015).
83. Peddle, S. D. et al. Soil DNA chronosequence analysis shows bacterial community re-assembly following post-mining forest rehabilitation. *Restor. Ecol.* **31**, e13706 (2023).

2. Definition of abundant, intermediate, and rare taxa

The classification of taxa based on the average abundance across all samples appears arbitrary and does not account for taxa that shift from rare to abundant over the course of succession. You identified ~140 taxa that follow this pattern and categorized them as intermediate, suggesting that some taxa start as rare but become abundant by the end of restoration. Shouldn't these taxa receive more discussion?

Additionally, the "sample-specific rarity cutoff" remains unclear. According to Figure 1a, OTUs exceeding 0.3% in some samples constitute only a small portion of the most abundant OTUs, yet a large fraction of them ultimately fall into the intermediate category. This suggests that inter-sample abundance variability is common and non-negligible.

To better justify your classification, I suggest calculating the abundance category (abundant/intermediate/rare) for each OTU in each sample and then determining the percentage

of samples in which each OTU falls into each category. Alternatively, quantifying inter-sample abundance variability for each taxon could demonstrate whether most taxa remain stable across samples, supporting the use of an average-based classification.

While I still believe that tracking taxa with significant temporal shifts would provide deeper ecological insights, I acknowledge that this might require more extensive analyses beyond the scope of this revision.

>> Thank you for highlighting this important issue again. As mentioned above, we have completely revised the entire manuscript due to regenerating the ASV table using the raw sequences. We have described the standard procedure in the Methods. Please see Lines 399–407. Compared to OTUs clustered at 97% similarity, the use of ASV allows for the precise identification of differences in each amplicon sequence, providing higher resolution and reducing the omission of rare taxa. In the previous version, a total of 10,378 OTUs were identified, among which 260 OTUs were classified as abundant taxa, 3,656 OTUs as intermediate taxa, and 6,462 OTUs as rare taxa. In the new version, a total of 25,077 ASVs were identified, with 601 ASVs classified as abundant taxa, 4507 ASVs as intermediate taxa, and 19,969 ASVs as rare taxa. This means that a large number of taxa, especially rare taxa, have been detected. Please see Lines 407–410, 114–119. Moreover, the rarefaction curves have confirmed that most ASVs have been successfully recovered, providing comprehensive coverage of bacterial richness for the entire community as well as abundant, intermediate, and rare subcommunities. Please see Lines 111–114, and Supplementary Fig. 3. However, the rarefaction curves of rare taxa did not reach a plateau when using OTUs in the previous version.

According to your suggestion, we have assessed the variability of the relative abundance of each ASV along restoration duration based on Pearson correlation. The results indicated that the majority of ASVs (80.74%) remained stable in relative abundance. Specifically, 85.76% of rare ASVs exhibited nonsignificant temporal variation in relative abundance, whereas the majority of abundant ASVs responded positively (34.28%) or negatively (38.43%) to restoration duration. We also provided the number of ASVs that exhibited significant or nonsignificant changes at the phylum level. Please see Lines 496–498, 145–149, and Supplementary Fig. 6.

We have discussed this result. The majority of rare ASVs were only found in one or a limited number of samples due to their specific habitat preferences and narrow metabolic versatility, which may be the main reason for their relative abundance showing negligible

changes over time. Even so, the composition of rare subcommunities varied significantly, driven primarily by species-specific ecological niches across restoration stages. However, for abundant taxa, the significant changes observed in a greater number of ASVs indicated population fluctuations was prevalent during restoration due to differences in their tolerance and resistance in response to environmental fluctuations. For example, abundant ASVs belonging to Firmicutes, which have high tolerance to extreme drought conditions, exhibited higher relative abundance during the early stages of restoration. Please see Lines 229–242. In addition, intermediate taxa included some ASVs that fluctuate between rare and abundant states, resulting in their community assembly being primarily driven by variable selection due to their ecological fitness is strongly influenced by temporal shifts in environmental conditions. Please see Lines 253–258. Finally, we completely agree that taxa with significant temporal shifts have important ecological significance. We have highlighted this information as a limitation in the revised manuscript and anticipate focusing on these responsive taxa in subsequent studies. Please see Lines 301–310.

Lines 399–407 “*The raw sequences were processed by removing adapter, trimmed, quality filtered, and deduplication using Cutadapt (v.1.18)⁷⁴ and fastp (v.0.23.4)^{75,76}. The resulting high-quality sequence data was analyzed using the Quantitative Insights into Microbial Ecology (QIIME2 v.2024.10) pipeline⁷⁷. The DADA2 plugin was then employed to denoise and generate feature sequence and table of amplicon sequence variants (ASVs)⁷⁸. The taxonomic identity of each ASV was determined using the SILVA database (v.138)⁷⁹. Sequences assigned to mitochondria, chloroplasts, and archaea were excluded from downstream analysis.*”

Lines 407–410 “*Compared to operational taxonomic units (OTUs) clustered at 97% similarity, the ASVs approach allows precise identification of differences in each amplicon sequence, thereby reduces the omission of rare taxa and provides higher resolution at the species level⁸⁰. A total of 2,505,038 sequences and 25,077 ASVs were obtained.*”

Lines 114–119 “*A total of 601 ASVs (2.40%) were classified as abundant taxa, accounting for 55.54% of the total sequences, and 4507 ASVs (17.97%) were classified as intermediate taxa, accounting for 33.59% of the total sequences (Fig. 1b, c). In contrast, although a larger proportion of ASVs (19969, 79.63%) were identified as rare taxa, their relative abundance accounted for only 10.87% of the total sequences (Fig. 1b, c).*”

Lines 111–114 “*The rarefaction curves for most of our samples reached a steady plateau, signifying that the majority of bacterial amplicon sequence variants (ASVs) were successfully*

recovered, providing comprehensive coverage of the bacterial richness in our dataset (Supplementary Fig. 3)."

Lines 496–498 *"Additionally, we evaluated the variability in the relative abundance of each ASV over the restoration duration based on Pearson correlation to test whether their abundance remained relatively stable across samples."*

Lines 145–149 *"Furthermore, we evaluated the temporal variability of the relative abundance of each ASV and found that the substantial proportion (80.74%), predominately rare taxa, remained relatively stable along restoration duration (Supplementary Fig. 6). In contrast, most of the abundant ASVs (72.71%) exhibited either positive (34.28%) or negative (38.43%) responses in relative abundance under the influence of the SCB establishment."*

Lines 229–242 *"The majority of rare taxa possessed a high level of habitat specificity, and were only found in a single or limited number of samples, which may be attributed to their specific habitat preferences and narrow metabolic versatility^{21,51}. Although the relative abundance of these rare ASVs showed negligible changes over time, their community composition displayed substantial variations, driven primarily by species-specific ecological niches across restoration stages²⁰. In contrast, a significant proportion of abundant ASVs, alongside certain intermediate ASVs, exhibited either positive or negative responses in relative abundance to restoration duration, revealing pronounced population fluctuations during restoration. Such dynamics of their relative abundance could be attributed to the differences in their tolerance and resistance in response to environmental fluctuations. For example, a greater number of abundant ASVs belonging to Firmicutes exhibited higher relative abundance during the early stages of restoration, due to their high tolerance to extreme drought conditions⁴⁴."*

Lines 253–258 *"Additionally, the implementation of artificial restoration efforts and the establishment of plants introduce variable selection by altering the soil physicochemical properties throughout restoration process^{37,40}. Intermediate taxa (comprising some ASVs that fluctuate between rare and abundant states) exhibit heightened sensitivity to these changes, as their ecological fitness is strongly influenced by temporal shifts in environmental conditions⁵⁶."*

Lines 301–310 *"Firstly, our categorization approach, based on the average relative abundance of ASVs, may fail to capture those taxa that exhibit changes in relative abundance during restoration (i.e., conditionally abundant/rare ASVs). Although significant temporal variations in the composition of abundant and rare subcommunities were observed, our results*

indicated that the relative abundance of most ASVs remained stable across restoration stages (see supplementary Figure 6). Therefore, we expected the average relative abundance-based categorization approach was less likely to bias our main conclusions, within the context of our datasets. However, given the potential ecological significance of conditionally abundant/rare ASVs, future studies should consider more nuanced classifications of abundant and rare taxa during restoration⁴⁰.”

Supplementary Figure 3. Rarefaction curves of individual samples. a Entire dataset; **b** Abundant subcommunity; **c** Intermediate subcommunity; **d** Rare subcommunity. Lines with different colors indicate samples with different restoration durations.

Supplementary Figure 6. Number of ASVs with significant or nonsignificant changes in relative abundance along restoration duration. **a** The number and proportion of ASVs with significant or insignificant changes in relative abundance along restoration duration in the entire community, as well as in the abundant, intermediate, and rare subcommunities. **b–e** The number of these ASVs at the phylum level in the entire community, as well as in the abundant, intermediate, and rare subcommunities. Statistically significant positive and negative ($P \leq 0.05$), and nonsignificant ($P > 0.05$) changes based on Pearson correlation are indicated by red, black, and gray, respectively.

References:

20. Wang, X. et al. Abundant and rare fungal taxa exhibit different patterns of phylogenetic niche conservatism and community assembly across a geographical and environmental gradient. *Soil Boil. Biochem.* **186**, 109167 (2023).
21. Jiao, S. & Lu, Y. Abundant fungi adapt to broader environmental gradients than rare fungi in agricultural fields. *Glob. Chang. Biol.* **26**, 4506–4520 (2020).
37. Dini-Andreote, F., Stegen, J. C., van Elsas, J. D. & Salles, J. F. Disentangling mechanisms that mediate the balance between stochastic and deterministic processes in microbial succession. *Proc. Natl. Acad. Sci. U.S.A* **112**, E1326–E1332 (2015).
40. Jia, X., Dini-Andreote, F. & Salles, J. F. Unravelling the interplay of ecological processes structuring the bacterial rare biosphere. *ISME Commun.* **2**, 96 (2022).

44. Hou, Q. et al. Active restoration efforts drive community succession and assembly in a desert during the past 53 years. *Ecol. Appl.* **35**, e3068 (2025).
51. Barberán, A. et al. Why are some microbes more ubiquitous than others? Predicting the habitat breadth of soil bacteria. *Ecol. Lett.* **17**, 794–802 (2014).
56. Jia, X., Dini-Andreote, F. & Falcão Salles, J. Community assembly processes of the microbial rare biosphere. *Trends Microbiol.* **26**, 738–747 (2018).
74. Martin, M. CUTADAPT removes adapter sequences from high-throughput sequencing reads. *EMBnet J.* **17**, 10–12 (2011).
75. Chen, S., Zhou, Y., Chen, Y. & Gu, J. fastp: an ultra-fast all-in-one FASTQ preprocessor. *Bioinformatics* **34**, 884–890 (2018).
76. Chen, S. Ultrafast one-pass FASTQ data preprocessing, quality control, and deduplication using fastp. *iMeta* **2**, e107 (2023).
77. Caporaso, J. G. et al. QIIME allows analysis of high-throughput community sequencing data. *Nat. Methods* **7**, 335–336 (2010).
78. Callahan, B. J. et al. DADA2: High-resolution sample inference from Illumina amplicon data. *Nat. Methods* **13**, 581–583 (2016).
79. Quast, C. et al. The SILVA ribosomal RNA gene database project: improved data processing and web-based tools. *Nucleic Acids Res.* **41**, D590–596 (2013).
80. Callahan, B. J., McMurdie, P. J. & Holmes, S. P. Exact sequence variants should replace operational taxonomic units in marker-gene data analysis. *ISME J.* **11**, 2639–2643 (2017).

3. Taxonomic resolution in abundance classification

Since OTUs can represent various taxonomic levels (phylum, class, family, genus, species), comparing abundance across these levels is problematic. While OTU-based analysis is common, its combination with rare/abundant classification is less so. A “class” with an unknown order could be classified as abundant, while an OTU at the species level could be categorized as rare. To address this, I recommend reporting taxonomic resolution for each class (% of OTUs annotated to genus vs. phylum for abundant/intermediate/rare categories). If the resolution is

consistent across categories, your approach should be robust. However, if there are significant discrepancies, you might consider restricting the analysis to OTUs with consistent taxonomic resolution (e.g., genus-level) to reduce ambiguity.

>> Thanks for your comments. We are sorry that this critical issue was not substantially addressed in the previous version. The amplicon sequences were clustered into OTUs at 97% similarity, which inherently introduced taxonomic ambiguity across different levels (e.g., a single OTU may span species to family level). We fully recognize the limitations of the OTU-based approach in distinguishing between rare and abundant taxa. To fundamentally address this issue, we have implemented significant methodological improvements by adopting DADA2-derived ASVs instead of traditional OTU clustering (please see response above). This approach provides single-nucleotide resolution, enabling more precise detection of 25,077 ASVs (a 142% increase from the original 10,378 OTUs).

In addition, the taxonomic identity of each ASV was determined using the SILVA database (v.138). We found that 61.26% of ASVs were annotated at the species level, and 92.77% were annotated at the genus level. The remaining unclassified ASVs likely represent either: (1) database coverage limitations, or (2) uncultured microbial dark matter lacking formal taxonomic designation at the species level. Nevertheless, these unannotated ASVs still capture genuine biological variation through single-nucleotide resolution, potentially representing distinct microbial strains. We believe that the use of ASVs enables the revised manuscript to more accurately reflect the diversity and structure of both abundant and rare subcommunities under the same high-resolution standard.

4. Methodological concerns regarding rarefaction, network analysis, and community assembly indices

This is my most critical concern. There needs to be a clear justification that rarefaction, network analysis, and community assembly indices are appropriate for a dataset that has been divided into abundant and rare taxa. While these methods are widely used in microbial ecology, they are typically applied to entire communities rather than split into categories. I am concerned that the observed patterns may be driven by mathematical artifacts rather than ecological processes.

Rarefaction disproportionately affects rare taxa, introducing stochasticity. The higher stochasticity observed in rare taxa's community assembly could be an artifact of rarefaction

rather than an ecological signal. I suggest rarefying only for richness calculations and using unrarefied data for all other analyses.

Network analysis may be biased because abundant taxa are more frequently detected across samples, making them more likely to co-occur with other taxa. This can artificially inflate their degree (number of connections) and centrality (importance in the network), leading to an overestimation of their ecological significance. To test for such biases, I recommend performing a supplementary analysis on a completely randomized microbial dataset, categorizing abundant/rare groups as you did, and checking whether similar patterns emerge.

Community assembly indices may also be affected, as dominant phyla could skew deterministic signals, while rare taxa—being absent in many samples—could inflate stochasticity. More clarification is needed on how your dichotomization might introduce mathematical biases in these methods.

Additionally, given the strength of your temporal dataset, why not analyze network structure and community assembly across different restoration stages? This would provide more valuable insights into microbial succession in the context of restoration ecology.

>> Thank you for these very useful suggestions, which have significantly strengthened our manuscript. Some results have been modified, while others remain consistent with the previous version, which we have mentioned below. First, when calculating the diversity, we employed a rarefied ASV table to ensure that diversity comparisons between samples were made at the same sequencing depth. Please see Lines 410–413. The diversity of abundant and intermediate taxa still exhibited a consistent nonlinear increase over time as before. With more rare ASVs being detected, we found that the diversity of rare taxa significantly increased linearly with restoration duration, whereas the pattern that was not apparent in the previous version. Please see Fig. 2. Importantly, despite this enhanced detection of rare taxa, their diversity remains unassociated with ecosystem multifunctionality and most individual functions, consistent with our previous results. Please see Lines 187–193 and Fig. 6. For all other analyses, we used the unrarefied data. The temporal changes in the composition of abundant, intermediate, and rare subcommunities were consistent with the previous results.

Second, we have reconstructed the co-occurrence network using ASV table with the same standards as before and calculated the topological properties at the node level. In the previous version, abundant taxa had higher degree, centrality, and clustering coefficient compared to rare taxa, which may have led to your concerns. As you pointed out, abundant taxa were more

frequently detected across samples, making them more likely to co-occur with other taxa, thus potentially overestimating their ecological significance. However, the ASV-based analysis revealed important new insights. The proportion of rare taxa in the network significantly increased with improved ASV detection, and their topological properties showed no significant differences from abundant taxa. Please see Lines 172–175, and Figure 5. In separated networks, rare taxa exhibited higher network robustness than abundant taxa. Please see Supplementary Fig. 7. We have also constructed the network at each restoration stage. Temporal network analysis across restoration stages revealed increasing network complexity, with nodes showing higher degree and betweenness centrality over time. Please see Supplementary Fig. 8. In the networks of each restoration stage, the number of intermediate ASVs was higher due to the low diversity of abundant taxa and the low occurrence frequency of rare taxa. Rare taxa consistently exhibited higher betweenness centrality but lower clustering coefficients than abundant taxa across all restoration stages, while showing comparable degree and closeness centrality. Please see Lines 177–184, and Supplementary Fig. 9. These results demonstrate that ASV-level analysis provides a more balanced representation of network roles of rare taxa. For our co-occurrence network, abundant and rare taxa exhibited similar topological properties, indicating that they played an equally important roles in maintaining network structure and stability, regardless of their frequency of occurrence in the samples. Please see Lines 283–286. While we maintained our discussion about network analysis limitations, we did not perform randomized community analyses as the revised results already addressed the key concerns about detection bias.

More importantly, despite the network was reconstructed, we found that abundant taxa with higher degree displayed stronger correlations with ecosystem multifunctionality, while rare taxa showed an opposite trend. This robust pattern aligned with the previous version, reinforcing the divergent functional roles of microbial abundance groups. Moreover, the proportion of ASVs associated with multiple functions increased for abundant taxa and decreased for rare taxa, suggesting that abundant ASVs were more likely to be associated with multiple functions simultaneously, whereas rare ASVs were typically linked to single function. These significant results remained robust across both OTU- and ASV-based analyses, further reinforcing the divergent functional roles of abundant and rare taxa. Abundant taxa were associated with multiple functions simultaneously through interspecies connections, whereas rare taxa were linked to individual functions independently.

Finally, the assembly processes of abundant, intermediate, and rare subcommunities were re-evaluated due to structural differences between ASV- and OTU-based classifications. In the

previous version, the assembly of all three subcommunities was dominated by deterministic processes. However, the revised analysis revealed a key distinction: abundant subcommunities were primarily governed by stochastic processes, whereas deterministic processes remained the dominant driver for intermediate and rare taxa. Please see Lines 152–160, and Fig. 4. We consider these updated findings more biologically plausible. Abundant taxa, typically ecological generalists with broad niche breadth, can exploit a wider range of resources. Due to their insensitivity to environmental selection, their distribution patterns are primarily shaped by stochastic processes such as dispersal limitation and ecological drift. In contrast, rare taxa, as specialists, have narrower range of habitats and specific environmental fitness, making their community assembly predominantly deterministic. More specifically, homogeneous environmental conditions exert strong selective pressure (i.e. homogeneous selection) on rare taxa, resulting in their perpetually low relative abundance and convergent community composition. The implementation of artificial restoration efforts also introduced variable selection by altering the soil physicochemical properties, with intermediate taxa being most responsive to these changes. These points were emphasized in the Discussion. Please see Lines 243–258. Following your suggestions, we further analyzed temporal shifts in assembly processes of abundant and rare subcommunities across restoration stages. At all restoration stages, stochastic processes dominated the assembly of abundant subcommunities, while homogeneous selection had a greater influence on the assembly of intermediate and rare subcommunities. Homogenizing dispersal was more pronounced in the early restoration stages, as wind erosion and homogenized environmental conditions promoted the stochastic dispersal of species. Please see Lines 160–167, 258–262, and Figure 4. Additionally, the null-modeling-based quantitative framework is widely used to assess the relative importance of assembly processes of microbial communities, which heavily relies on phylogenetic signals, as well as the number and distribution of individuals. Therefore, the assessment of the assembly processes of different subcommunities in this study may introduce mathematical biases. For example, stochasticity may be overestimated in abundant taxa due to low richness; the assembly processes of rare taxa that are susceptible to competition may be misestimated because the classification overlooks interactions between different types of species. We have emphasized this information as a limitation of the manuscript in the Discussion. Please see Lines 311–323.

Lines 410–413 “*To standardize richness comparison among samples at an equal sampling depth⁷², the ASV table was rarefied to the lowest number of sequences (24,911) found within an individual sample for richness calculations.*”

Lines 187–193 “A significant and positive relationship between soil bacterial ASV richness and ecosystem multifunctionality (EMF) was found for abundant and intermediate taxa, but not for rare taxa (Fig. 6a). Consistent results were observed for plant productivity, decomposition rate, and soil nutrient pool indices (Supplementary Fig. 10). The great majority of individual functions were significantly and positively associated with ASV richness of abundant and intermediate taxa, whereas only 4 functions were associated with rare taxa (Fig. 6b).”

Lines 172–175 “The topological properties of abundant ASVs, including degree, betweenness centrality, and clustering coefficient, did not significantly differ from those of rare ASVs, while their closeness centrality was notably lower than that of rare ASVs (Fig. 5b).”

Lines 177–184 “In addition, we constructed correlation networks at each restoration stage and found a progressive increase in network complexity over time, as evidenced by noticeable elevations in the degree and betweenness centrality of the nodes (Supplementary Figs. 8 and 9). Throughout all restoration stages, rare taxa exhibited higher betweenness centrality and lower clustering coefficient than abundant and intermediate taxa, whereas their degree surpassed that of latter two during the late restoration stages (Supplementary Fig. 9).”

Lines 283–286 “For co-occurrence network, both abundant and rare taxa exhibited similar topological properties, indicating that they play equally important roles in maintaining network structure and stability, irrespective of their frequency of occurrence in the samples.”

Lines 152–160 “Leveraging the null-modeling-based quantitative framework, we elucidated that distinct ecological processes underpinning the assembly of abundant and rare subcommunities (Fig. 4). Specifically, the assembly of abundant subcommunities was mainly modulated by stochastic processes (69.3%), especially dispersal limitation (45.19%), while variable selection exerted a moderate deterministic influence (26.6%) on the abundant subcommunities (Fig. 4a). However, the assembly of intermediate (70.37%) and rare (73.53%) subcommunities was mainly governed by deterministic processes, with variable (43.43% for intermediate subcommunities) and homogenous selection (54.14% for rare subcommunities) being the main factors, respectively (Fig. 4b, c).”

Lines 243–258 “Our findings indicated that the successional dynamics of abundant and rare taxa were governed by contrasting community assembly processes during restoration, which aligns with previous studies^{21,41}. Abundant taxa, typically ecological generalists with broad niche breadth, can exploit a wider range of resources. Due to their insensitivity to

environmental selection, their distribution patterns are primarily shaped by stochastic processes such as dispersal limitation and ecological drift⁵²⁻⁵⁴. In contrast, rare taxa, as specialists, have narrower range of habitats and specific environmental fitness, making their community assembly predominantly deterministic, particularly through homogeneous selection^{18,52,55}. This is because homogeneous environmental conditions exert strong selective pressure on rare taxa, resulting in their perpetually low relative abundance and convergent community composition^{40,56}. Additionally, the implementation of artificial restoration efforts and the establishment of plants introduce variable selection by altering the soil physicochemical properties throughout restoration process^{37,40}. Intermediate taxa (comprising some ASVs that fluctuate between rare and abundant states) exhibit heightened sensitivity to these changes, as their ecological fitness is strongly influenced by temporal shifts in environmental conditions⁵⁶.”

Lines 160–167 “On the other hand, the temporal variations in the assembly processes of abundant and rare subcommunities were further delineated. Across all restoration stages, homogenizing dispersal and undominated processes had a greater influence on the temporal assembly of abundant subcommunities, while homogeneous selection dominated intermediate and rare subcommunities (Fig. 4d–f). Notably, homogenizing dispersal was more accentuated during the initial stages of restoration for the assembly of both abundant and intermediate subcommunities.”

Lines 258–262 “Notably, the importance of homogenizing dispersal on both abundant and intermediate taxa progressively diminished as restoration proceeds. This trend is partially attributed to the prevalence of wind erosion and environmental homogeneity during the early stages of restoration, which facilitated the stochastic dispersal of species^{44,57}.”

Lines 311–323 “Secondly, the null-modeling-based quantitative framework is widely applied to assess the relative influence of assembly processes in microbial communities^{37,62,63}. However, this approach heavily relies on assumptions regarding phylogenetic signals, as well as the number and distribution of individuals within a community. Therefore, applying this framework to different subcommunities in our study may introduce mathematical biases. For example, stochastic processes may be overestimated in abundant taxa due to their low richness⁶⁴. Additionally, categorical separation of taxa may obscure inter-group interactions, potentially leading to a misestimation of the assembly processes affecting rare taxa, which are often vulnerable to competition from abundant taxa⁵⁶. These limitations highlight the need for integrating complementary approaches in future studies, but our results nonetheless provide a

robust baseline for understanding how assembly processes differentially structure microbial subcommunities during ecological restoration.”

Figure 2. Temporal changes in the α -diversity and composition of abundant, intermediate, and rare subcommunities. **a–c** Relationships between restoration duration and the α -diversity of abundant, intermediate, and rare taxa. The black fitted lines are from linear and nonlinear regression. The solid and dotted lines represent statistically significant ($P \leq 0.05$) and nonsignificant ($P > 0.05$) relationships, respectively. **d–f** NMDS ordination of abundant, intermediate, and rare subcommunities based on Bray–Curtis distance. ANOSIM was performed to test the differences in abundant, intermediate, and rare subcommunities among restoration durations.

Figure 4. Relative importance of ecological processes mediating the assembly of abundant, intermediate, and rare subcommunities. **a–c** Relative importance of the assembly processes of abundant, intermediate, and rare subcommunities throughout the entire restoration process. **d–f** Relative importance of the assembly processes of abundant, intermediate, and rare subcommunities across different restoration stages. The relative importance of these five ecological processes was estimated using null model analysis.

Figure 5. Co-occurrence network of abundant, intermediate, and rare taxa. **a** The co-occurrence network constructed with abundant, intermediate, and rare taxa. The size of each node is proportional to its degree. The values below the network represent the number proportion of nodes in each group. **b** Boxplots of node-level topological properties of abundant, intermediate, and rare taxa, including degree, closeness centrality, betweenness centrality and clustering coefficient. The different letters represent significant differences determined by the Kruskal–Wallis test followed by Dunn post-hoc test (Bonferroni correction). **c–e** Relationships between the degree of ASVs and their correlation strength ($|r|$) with multifunctionality for abundant, intermediate, and rare taxa. The black fitted lines are from linear regression. EMF, ecosystem multifunctionality.

Figure 6. Relationships of diversity of abundant, intermediate, and rare taxa with multifunctionality and individual functions. **a** Relationships between multifunctionality and diversity of abundant, intermediate, and rare taxa. The black fitted lines are from linear regression. The solid and dotted lines represent statistically significant ($P \leq 0.05$) and nonsignificant ($P > 0.05$) relationships, respectively. **b** Pearson's correlation coefficients between individual functions and diversity of abundant, intermediate, and rare taxa. Edge width corresponds to the Pearson's r value. Positive and negative correlations are indicated by red and black edges, respectively. Pairwise correlations of these functions are displayed with a color gradient denoting Pearson's correlation coefficient. AGB, aboveground biomass; PC, plant carbon; PN, plant nitrogen; PP, plant phosphorus; DNA, soil DNA concentration; Suc, soil sucrose activity; β -Glu, soil β -glucosidase activity; Cel, soil cellulase activity; Ure, soil urease activity; ALP, soil alkaline phosphatase activity; Cat, soil catalase activity; SOC, soil organic carbon; $\text{SNH}_4^+\text{-N}$, soil ammonium; $\text{SNO}_3^-\text{-N}$, soil nitrate; STN, soil total nitrogen; SAP, soil

available phosphorus; STP, soil total phosphorus.

Supplementary Figure 7. Separate networks and their robustness. a–c Separate networks for abundant, intermediate, and rare taxa. N , the number of nodes; L , the number of edges. The size of each node is proportional to its degree. **d** Boxplots of the robustness of separate networks for abundant, intermediate, and rare taxa at 1000 iterations. The different letters represent significant differences determined by the Kruskal–Wallis test followed by Dunn post-hoc test (Bonferroni correction).

Supplementary Figure 8. Networks of abundant, intermediate, and rare taxa at each restoration stage. *N*, the number of nodes; *L*, the number of edges. The size of each node is proportional to its degree.

Supplementary Figure 9. Node-level topological properties of abundant, intermediate, and rare taxa in networks at each restoration stage. a Degree; **b** Closeness centrality; **c** Betweenness centrality; **d** Clustering coefficient. The error bars represent standard error.

References:

18. Pedrós-Alió, C. The Rare Bacterial Biosphere. *Annu. Rev. Mar. Sci.* **4**, 449–466 (2012).
21. Jiao, S. & Lu, Y. Abundant fungi adapt to broader environmental gradients than rare fungi in agricultural fields. *Glob. Chang. Biol.* **26**, 4506–4520 (2020).
37. Dini-Andreote, F., Stegen, J. C., van Elsas, J. D. & Salles, J. F. Disentangling mechanisms that mediate the balance between stochastic and deterministic processes in microbial succession. *Proc. Natl. Acad. Sci. U.S.A* **112**, E1326–E1332 (2015).
40. Jia, X., Dini-Andreote, F. & Salles, J. F. Unravelling the interplay of ecological processes structuring the bacterial rare biosphere. *ISME Commun.* **2**, 96 (2022).
41. Peng, Z. et al. Soil phosphorus determines the distinct assembly strategies for abundant and rare bacterial communities during successional reforestation. *Soil Ecol. Lett.* **3**, 342–355 (2021).
44. Hou, Q. et al. Active restoration efforts drive community succession and assembly in a desert during the past 53 years. *Ecol. Appl.* **35**, e3068 (2025).
52. Liu, L., Yang, J., Yu, Z. & Wilkinson, D. M. The biogeography of abundant and rare bacterioplankton in the lakes and reservoirs of China. *ISME J.* **9**, 2068–2077 (2015).
53. Xu, Q. et al. Microbial generalists and specialists differently contribute to the community diversity in farmland soils. *J. Adv. Res.* **40**, 17–27 (2022).
54. Hu, W. et al. Continental-scale niche differentiation of dominant topsoil archaea in drylands. *Environ. Microbiol.* **24**, 5483–5497 (2022).
55. Shu, D. et al. Rare prokaryotic sub-communities dominate the complexity of ecological networks and soil multinutrient cycling during long-term secondary succession in China's Loess Plateau. *Sci. Total Environ.* **774**, 145737 (2021).
56. Jia, X., Dini-Andreote, F. & Falcão Salles, J. Community assembly processes of the microbial rare biosphere. *Trends Microbiol.* **26**, 738–747 (2018).
57. Ferrenberg, S. et al. Changes in assembly processes in soil bacterial communities following

a wildfire disturbance. *ISME J.* **7**, 1102–1111 (2013).

62. Stegen, J. C. et al. Quantifying community assembly processes and identifying features that impose them. *ISME J.* **7**, 2069–2079 (2013).

63. Stegen, J. C., Lin, X., Konopka, A. E. & Fredrickson, J. K. Stochastic and deterministic assembly processes in subsurface microbial communities. *ISME J.* **6**, 1653–1664 (2012).

64. Gao, C. et al. Fungal community assembly in drought-stressed sorghum shows stochasticity, selection, and universal ecological dynamics. *Nat. Commun.* **11**, 34 (2020).

72. Hu, W. et al. Aridity-driven shift in biodiversity–soil multifunctionality relationships. *Nat. Commun.* **12**, 5350 (2021).

Detailed comments

Title: Why not specify "Tengger Desert" directly?

>> Thanks. We have revised the title due to changes in some results and specified "Tengger Desert". Please see the title.

Title “*Divergent community assembly processes and multifunctionality contributions of abundant and rare soil bacteria during a 53-year restoration in the Tengger Desert, China*”

Introduction, Paragraphs 1 & 2: These sections would benefit from a clearer justification for why separating abundant and rare taxa is critical to understanding or improving ecological restoration.

One possible approach is to emphasize the ecological rationale: abundant taxa may be crucial for short-term functional recovery, while rare taxa contribute to ecosystem resilience through functional redundancy. Thus, studying them separately provides insights into different restoration goals.

Alternatively, you could frame the abundant-rare dichotomy as a trade-off in restoration strategies. Should restoration prioritize inoculating soils with rare taxa to enhance stress resilience, or focus on rebuilding abundant taxa to accelerate ecosystem productivity?

Introduction, Paragraph 3 (Line 69): Did you investigate temporal variation in community

assembly? The current figures do not seem to incorporate a temporal component.

Introduction, Paragraph 4: This section effectively explains the importance of dryland restoration and SCBs, but it does not clearly justify why microbial communities—especially abundant and rare taxa—should be considered separately. A reasonable revision would be to merge this paragraph with Paragraph 1 and move SCB-related details to the Methods section. Paragraphs 2 and 3 should also be reframed with a stronger focus on restoration.

>> Thanks for your suggestions. We have systematically revised the Introduction accordingly. First, we integrated the content on dryland ecosystem restoration from the original paragraph 4 into the paragraph 1 to enhance the coherence of background discussion. Please see Lines 25–34. And we moved the content related to SCBs to the Methods. Second, we emphasized the differential ecological importance of abundant and rare taxa during restoration in the paragraph 2. Please see Lines 46–58. Finally, we supplemented the paragraph 3 with the temporal changes in community assembly. Please see Lines 70–81.

Lines 25–34 *“Drylands, home to more than 38% of the global population, provide a wide array of ecosystem services^{1,2}. However, these regions are increasingly vulnerable to desertification due to increased aridity driven by climate change and anthropogenic pressures, jeopardizing global ecological security and sustainable socio-economic development^{1,3,4}. In response, ecological restoration efforts have been widely implemented globally, effectively enhancing biodiversity as well as ecosystem functions and services^{5–7}. Over the past seven decades, large-scale artificial restoration initiatives in China, including the Three-North Shelterbelt Development Program and the Grain for Green Program, have significantly contributed to the greening of drylands^{8–10}.”*

Lines 46–58 *“Abundant and rare taxa may play distinct roles in promoting ecosystem restoration owing to their divergent ecological and functional characteristics^{23,24}. Abundant taxa can utilize a wide range of resources, and are highly active in driving biogeochemical cycles due to their broad niche occupancy and metabolic versatility^{18,21,25}. As a result, they can significantly contribute to the rapid restoration of multiple functions in degraded ecosystems²⁶. In contrast, rare taxa have specific habitat preferences and narrow niche breadth and limited competitive ability, possess diverse genetic and metabolic lineages²⁷. This empowers rare taxa to promptly respond to environmental disturbance, however, their inherently high diversity may reinforce the resilience and long-term stability of ecosystem functioning through functional redundancy and complementary effects^{28,29}. Thus, analyzing abundant and rare taxa separately*

offers critical insights into divergent restoration goals and serves as guidance for the development of targeted restoration strategies.”

Lines 70–81 “*Previous studies have shown that stochastic processes have a greater impact on structuring abundant subcommunities, whereas rare taxa are primarily mediated by deterministic processes in reforestation⁴¹, farmland²¹ and dryland ecosystems²⁰. However, other studies have reported contrasting findings^{25,42,43}. Together, these results suggest that the assembly processes of abundant and rare subcommunities may be contingent upon the specific ecosystem types. Furthermore, there is increasing evidence that ecological restoration can drive a time-dependent shift in the balance between stochastic and deterministic processes in microbial succession^{37,44,45}. For instance, previous studies have indicated that within successional stages microbial communities are initially governed by stochastic processes, but the importance of deterministic processes intensifies as succession proceeded³⁷.”*

References:

1. Reynolds, J. F. et al. Global Desertification: Building a Science for Dryland Development. *Science* **316**, 847–851 (2007).
2. Maestre, F. T. et al. Structure and Functioning of Dryland Ecosystems in a Changing World. *Annu. Rev. Ecol. Evol. Syst.* **47**, 215–237 (2016).
3. Huang, J., Yu, H., Guan, X., Wang, G. & Guo, R. Accelerated dryland expansion under climate change. *Nat. Clim. Change* **6**, 166–171 (2016).
4. Huang, J. et al. Declines in global ecological security under climate change. *Ecol. Indic.* **117**, 106651 (2020).
5. Benayas, J. M. R., Newton, A. C., Diaz, A. & Bullock, J. M. Enhancement of Biodiversity and Ecosystem Services by Ecological Restoration: A Meta-Analysis. *Science* **325**, 1121–1124 (2009).
6. Atkinson, J. et al. Terrestrial ecosystem restoration increases biodiversity and reduces its variability, but not to reference levels: A global meta-analysis. *Ecol. Lett.* **25**, 1725–1737 (2022).
7. Resch, M. C. et al. Evaluating long-term success in grassland restoration: an ecosystem multifunctionality approach. *Ecol. Appl.* **31**, e02271 (2021).
8. Li, C. et al. Drivers and impacts of changes in China’s drylands. *Nat. Rev. Earth Environ.* **2**, 858–873 (2021).

9. Bryan, B. A. et al. China's response to a national land-system sustainability emergency. *Nature* **559**, 193–204 (2018).
10. Ouyang, Z. et al. Improvements in ecosystem services from investments in natural capital. *Science* **352**, 1455–1459 (2016).
18. Pedrós-Alió, C. The Rare Bacterial Biosphere. *Annu. Rev. Mar. Sci.* **4**, 449–466 (2012).
20. Wang, X. et al. Abundant and rare fungal taxa exhibit different patterns of phylogenetic niche conservatism and community assembly across a geographical and environmental gradient. *Soil Boil. Biochem.* **186**, 109167 (2023).
21. Jiao, S. & Lu, Y. Abundant fungi adapt to broader environmental gradients than rare fungi in agricultural fields. *Glob. Chang. Biol.* **26**, 4506–4520 (2020).
23. Zhang, Z., Lu, Y., Wei, G. & Jiao, S. Rare Species-Driven Diversity–Ecosystem Multifunctionality Relationships are Promoted by Stochastic Community Assembly. *mBio* **13**, e00449–00422 (2022).
24. Wang, Y. et al. Biogeographic pattern of bacterioplanktonic community and potential function in the Yangtze River: Roles of abundant and rare taxa. *Sci. Total Environ.* **747**, 141335 (2020).
25. Jiao, S., Chen, W. & Wei, G. Biogeography and ecological diversity patterns of rare and abundant bacteria in oil-contaminated soils. *Mol. Ecol.* **26**, 5305–5317 (2017).
26. Rivett, D. W. & Bell, T. Abundance determines the functional role of bacterial phylotypes in complex communities. *Nat. Microbiol.* **3**, 767–772 (2018).
27. Jousset, A. et al. Where less may be more: how the rare biosphere pulls ecosystems strings. *ISME J.* **11**, 853–862 (2017).
28. Xiong, C. et al. Rare taxa maintain the stability of crop mycobiomes and ecosystem functions. *Environ. Microbiol.* **23**, 1907–1924 (2021).
29. Chen, W. et al. Number of global change factors alters the relative roles of abundant and rare microbes in driving soil multifunctionality resistance. *Curr. Biol.* **35**, 373–382 (2025).
37. Dini-Andreote, F., Stegen, J. C., van Elsas, J. D. & Salles, J. F. Disentangling mechanisms that mediate the balance between stochastic and deterministic processes in microbial succession. *Proc. Natl. Acad. Sci. U.S.A* **112**, E1326–E1332 (2015).

41. Peng, Z. et al. Soil phosphorus determines the distinct assembly strategies for abundant and rare bacterial communities during successional reforestation. *Soil Ecol. Lett.* **3**, 342–355 (2021).
42. Hou, J. et al. Biogeography and diversity patterns of abundant and rare bacterial communities in rice paddy soils across China. *Sci. Total Environ.* **730**, 139116 (2020).
43. Li, P. et al. Distinct Successions of Common and Rare Bacteria in Soil Under Humic Acid Amendment – A Microcosm Study. *Front. Microbiol.* **10**, 2271 (2019).
44. Hou, Q. et al. Active restoration efforts drive community succession and assembly in a desert during the past 53 years. *Ecol. Appl.* **35**, e3068 (2025).
45. Tripathi, B. M. et al. Soil pH mediates the balance between stochastic and deterministic assembly of bacteria. *ISME J.* **12**, 1072–1083 (2018).

I will stop here, as the Results and Discussion sections may require significant revisions if you incorporate the additional analyses I suggested.

>> We sincerely appreciate your thoughtful and constructive feedback on our manuscript. We have carefully addressed all of your suggestions and have thoroughly revised the manuscript accordingly. Your insights have been invaluable in strengthening the clarity, rigor, and overall impact of our work. We believe these revisions have significantly improved the manuscript, and we are grateful for your guidance throughout this process. Please let us know if you have any additional feedback. We would be happy to incorporate any further suggestions.

Reviewer #2 (Remarks to the Author):

Dear authors, thank you for your detailed answers to this reviewer's comments and questions. The manuscript has improved largely.

I have some small edits that I suggest in the following lines:

Lines 123-125: Please acknowledge the need for deeper sequencing depth for rare groups. It is not a negative aspect but something that must be acknowledge to inform future studies.

>> We thank you for pointing out this. We completely agree with you that increasing sequencing depth and using precise sequence identification methods will help us to accurately investigate rare taxa. In the revised version, the use of ASV data detected a significantly large number of rare taxa, resulting in the rarefaction curves reaching a plateau in most samples, which addresses the shortcomings of the previous version. Please see the response above to the second concern of Reviewer #1.

Line 172: Change "positive connections" to "positive correlations".

Line 173: Instead of "indicating higher prevalence of synergistic" say "indicating higher putative synergistic relationships"

Remember, correlation networks provide a view of correlations and provide insights for further hypotheses development.

>> Thanks. Indeed, network analysis is often used to infer interspecific correlations. However, it does not provide conclusive evidence of actual ecological interactions. We have emphasized this point in the Discussions. This sentence has been revised for clarity. Please see Lines 170–172.

Lines 170–172 *“Within the network, positive correlations predominated (98.55%), indicating higher putative synergistic relationships among bacterial ASVs during the restoration of desertified ecosystems.”*

Divergent community assembly processes and multifunctionality contributions of abundant and rare soil bacteria during a 53-year restoration in the Tengger Desert, China

COMMSBIO-24-5478B submitted to *Communications Biology*

Dear Referees,

We sincerely appreciate the time and effort the reviewer has dedicated to evaluating our manuscript. We are grateful for the constructive comments, which have helped us significantly improved the overall quality of the manuscript. We also apologize for any shortcomings or oversights that may have remained in the earlier version and appreciate the opportunity to improve our work based on your suggestions.

Below, we have labeled each reviewer's comments, and we address them one by one. Reviewer comments are in regular black font while our responses are shown in blue; *Italic texts* refer to new text added in the manuscript. To facilitate review process, we have included both a tracked-changes version of the revised manuscript highlighting all modifications, and a clean version for ease of reading. Note that line number within reviewer comments refer to the original manuscript, while line numbers in our reply refer to the clean version of the manuscript resubmitted.

Reviewers' Comments:

Reviewer #1 (Remarks to the Author):

General comments:

This round of revision has significantly improved the manuscript. I now have enough context to not recommend rejection. However, the manuscript still requires substantial revision to meet the standards of *Communications Biology*. Most importantly, I strongly recommend incorporating structural equation modeling (SEM) to synthesize your analyses and deliver a clear, integrative take-home message. Right now, the manuscript reads like a set of preliminary analyses without a compelling narrative. The discussion remains surface-level, especially in areas where your data could support deeper, testable insights. The network analysis, in particular, feels underdeveloped—please start by clearly defining its purpose in your study.

>> Thank you for your recognition of our work and your valuable suggestions. We have

thoroughly revised the manuscript based on your detailed comments below. For example, we replaced certain terms, recalculated the two niche breadth indices, and provided more detailed descriptions of some methods. A structural equation model has been constructed to infer the direct and indirect relationships among restoration duration, plant cover, soil pH, bacterial ASV richness, network degree, and ecosystem multifunctionality (Please see Figure 6b). This has significantly enriched and improved our manuscript. Additionally, in our manuscript, network analysis was used to compare the differences in topological roles among abundant, intermediate, and rare taxa based on their node-level properties. We fully acknowledge your concerns regarding this analysis, but it still provides valuable insights into potential indirect connections among microorganisms or their shared responses in abundance to environmental and temporal factors using sequencing data. We have further elaborated the node-level properties, identified keystone nodes, and revised the relevant content more clearly in the Results and Discussion. Please see our detailed point-by-point responses below. Once again, thank you very much.

The data itself is highly promising, but I encourage you to take a step back and reflect more deeply on the knowledge gap you're addressing and the practical value of your findings. This is a rare 53-year restoration chronosequence—what lessons can you offer future restoration practitioners? Have microbial communities fully recovered with the current restoration approach? If not, how does your work help address that gap? If recovery is complete, which subcommunities restored more slowly, and what does that imply? Based on your assembly mechanism results, how could restoration outcomes be improved?

>> We are grateful for your raising these critical questions regarding the practical value of our 53-year restoration dataset. These perspectives align perfectly with our goal of bridging microbial ecology and restoration practice. In fact, these issues have been discussed in our recent paper published in *Ecological Applications* (Hou et al. 2025). Using this 53-year restoration dataset, our results demonstrated that both plant and soil microbial (bacterial and fungal) communities gradually approached reference (non-desertified natural) communities over time since restoration, but have not fully recovered within 53 years. By identifying critical temporal thresholds for relatively rapid changes in community composition, we found that plant communities exhibited a more delayed response to restoration effects compared to soil microbial communities. Additionally, we identified both positive and negative indicator species within plant and microbial communities during restoration. Monitoring these indicator species in natural ecosystems could help detect early warning signals of desertification.

In this study, we primarily focused on the differences among abundant, intermediate, and rare bacterial taxa. Our current findings also provide practical insights for improving dryland restoration. First, conventional afforestation and grass-planting actions can promote plant cover and the richness of high-abundance microbial taxa, while soil environmental factors (i.e., environmental selection) likely shape rare subcommunities. Therefore, additional regulation of soil properties (e.g., soil pH) is necessary to meet the niche requirements of rare taxa. Such targeted measures could enhance the functional redundancy of microbial communities, thus improving ecosystem multifunctionality and long-term ecological stability. Second, the key ASVs identified may serve as bioindicators for monitoring dryland restoration, due to their role as ecological engineers and their ability to orchestrate microbial communities in performing ecosystem functions. Therefore, successful restoration may be achieved with the introduction of these ASVs, either through targeted isolation and cultivation in artificial environments or via the application of existing commercial microbial inoculants. These insights have been emphasized in the Discussion. Please see Lines 331–343.

Lines 331–343 *“This study provides practical insights for advancing dryland restoration. First, conventional afforestation and grass-planting effectively enhance plant cover and the richness of high-abundance microbial taxa (Fig. 6b). However, additional regulation of soil properties (e.g., soil pH) is necessary to meet the niche requirements of rare microbial taxa, and thereby promoting the functional redundancy of microbial communities during ecological restoration. Such enhancements are instrumental in bolstering ecosystem multifunctionality and ensuring long-term ecological stability²⁹. Furthermore, the keystone ASVs identified may serve as reliable bioindicators for monitoring dryland restoration, given their pivotal role as ecosystem engineers and their ability to orchestrate microbial communities in performing ecosystem functions⁶⁵. Therefore, successful restoration may be achieved with the introduction of these ASVs, either by targeted isolation and cultivation in artificial environments or via the deployment of commercially available microbial inoculants.”*

References:

29. Jousset, A. et al. Where less may be more: how the rare biosphere pulls ecosystems strings. *ISME J.* **11**, 853–862 (2017).
65. Banerjee, S., Schlaeppi, K. & van der Heijden, M. G. A. Keystone taxa as drivers of microbiome structure and functioning. *Nat. Rev. Microbiol.* **16**, 567–576 (2018).

Detailed comments:

Lines 106–109: This section discussing shelterbelt forest restoration (SCBs) reads disconnected from the surrounding content. I suggest moving it earlier in the introduction (e.g., around Lines 38–41), potentially replacing or following the Three-North Shelterbelt example. Alternatively, consider moving this content to the Methods section. In the remainder of the introduction, use a more general term like “restoration” rather than focusing specifically on SCBs.

>> Thanks for your suggestion. We have relocated the statement regarding SCBs to the opening paragraph and refined its wording to enhance clarity. Please see Lines 35–38.

Lines 35–38 *“Among the ecological restoration strategies widely employed in these projects, straw checkerboard barriers (SCBs) stand out as one of the most cost-effective and demonstrably highly effective in stabilizing shifting sand and curbing wind erosion^{11,12}.”*

References:

11. Li, X. R., Xiao, H. L., He, M. Z. & Zhang, J. G. Sand barriers of straw checkerboards for habitat restoration in extremely arid desert regions. *Ecol. Eng.* **28**, 149–157 (2006).

12. Zhang, S. et al. Effect of Straw Checkerboards on Wind Proofing, Sand Fixation, and Ecological Restoration in Shifting Sandy Land. *Int. J. Environ. Res. Public Health* **15**, 2184 (2018).

I will jump to method before results:

Lines 461–463: Reference 44 is highly relevant. Please summarize its key findings explicitly here, rather than vaguely stating that it “triggers changes.”

>> We appreciate your attention to this omitted detail. The sentence has been carefully amended to address this point. Please see Lines 390–395.

Lines 390–395 *“Our recent study has demonstrated that the establishment of SCBs triggers a progressive convergence of plant and soil microbial communities toward those characteristic of non-desertified natural states, albeit complete restoration remains unrealized even after 53 years⁴⁶. During this process, plant-soil microbe correlations and elevated soil organic carbon content emerged as key drivers of community succession and ecological restoration.”*

References:

46. Hou, Q. et al. Active restoration efforts drive community succession and assembly in a desert during the past 53 years. *Ecol. Appl.* **35**, e3068 (2025).

Line 474 & Supplementary Figure 1: Why don't you include the 53-year restoration plot in panel a? Including it would improve clarity and completeness.

>> Thanks. We have now revised Supplementary Figure 1a and further clarified the details in its legend.

Supplementary Figure 1. Maps of study area and plots with different restoration durations. **a** The study area is located on the southern edge of the Tengger Desert in northwest China. White circle indicates the main sampling area of Jingtai County; red circle represents the 53-year restoration plot in the Shapotou restoration area. **b** White circles and numbers indicate plots with different restoration duration (years) within the main sampling area. It should be noted that the 53-year restoration plot is geographically separated from these plots.

Line 530: Please briefly explain the MultiCoLA method and cite the original reference. Without this context, Supplementary Figure 2 is difficult to interpret. Also clarify the rationale for selecting the two cutoff values (0.028% and 0.002%).

>> Thanks for your suggestions. The missing details regarding the MultiCoLA method and threshold selection criteria have now been systematically supplemented. Please see Lines 448–467.

Lines 448–467 “Initially, ASV dataset was ranked in ascending order on their sequences count. Low-abundance ASVs were removed based on the continuous percentage (1–95%) of the total number of sequences to obtain abundant (i.e., rare ASVs were removed) and rare (i.e., rare ASVs were retained) truncated datasets (Supplementary Fig. 2a, b). Pairwise distance matrices were then calculated for the original and truncated datasets using the Bray–Curtis dissimilarity index. Subsequently, community structure and the main patterns of community variation (using non-metric multidimensional scaling) of the original and truncated datasets were compared using the non-parametric Spearman rho correlation coefficient and the Procrustes method (Supplementary Fig. 2c–f)⁴⁸. While the abundant dataset maintained a high degree of structural consistency with the original dataset, the rare dataset demonstrated marked divergence in community composition. Finally, we precisely define thresholds for delineating abundant and rare truncated datasets based on the Spearman and Procrustes correlation values, and calculated the corresponding relative abundances. The results showed that the abundant truncated dataset retained nearly identical community structure and variation compared to the original dataset until 45% (corresponding to an average relative abundance of 0.028%) of rare ASVs were removed, while the rare truncated dataset showed no obvious divergence from the original dataset when the threshold exceeded 11% (corresponding to an average relative abundance of 0.002%) (Supplementary Fig. 2).”

References:

48. Gobet, A., Quince, C. & Ramette, A. Multivariate Cutoff Level Analysis (MultiCoLA) of large community data sets. *Nucleic Acids Res.* **38**, e155 (2010).

Line 552: A scatter plot showing geographic distance versus Bray–Curtis dissimilarity would be helpful. I definitely not meant to accusing you are deliberately hiding something but it’s possible that the 53-year plot is geographically distant from all others, potentially acting as an outlier and lowering the overall correlation. Given that geographic distance and restoration age seem correlated (per Supplementary Figure 1), it’s surprising to see such a discrepancy in their respective effects.

>> Thank you for your suggestion. We have now included scatter plots illustrating the relationships between the differences in bacterial community composition and geographical distance and time interval. Please see Supplementary Figure 3. Just as you said, the 53-year plot is geographically distant from all others, which led to the weaker correlation. However, we

retained this plot to obtain a longer chronosequence. Importantly, time interval showed stronger correlations with bacterial community dissimilarity than geographic distance, suggesting that the observed variations in bacterial community composition were primarily determined by restoration duration rather than spatial autocorrelation.

Supplementary Figure 3. Relationships between the differences in bacterial community composition and geographical distance and time interval. a Geographical distance; **b** Time interval. The solid and dotted lines represent statistically significant ($P \leq 0.05$) and nonsignificant ($P > 0.05$) relationships, respectively. Significant P values are represented by *** when $P < 0.001$, ** when $P < 0.01$ and * when $P < 0.05$.

Lines 555–556: Consider rephrasing to: “These results indicate that differences in bacterial community composition were significantly associated with restoration time intervals, but not with geographical distance (Supplementary Table 1).” This wording better reflects that you are discussing beta diversity, not absolute composition.

>> Thanks. This sentence has been revised in accordance with your suggestion. Please see Lines 484–486.

Lines 484–486 “These results indicated that differences in bacterial community composition were significantly associated with restoration time interval, but not with geographical distance (Supplementary Fig. 3).”

Line 597: Please indicate which software or platform and version was used for data analysis.

>> Thanks. This information has been added. Please see Lines 527–528.

Lines 527–528 *“All statistical analyses were carried out using R software (v.4.1.1) unless otherwise indicated.”*

Lines 598–599: A reference is missing here. Also, specify whether any R packages were used to calculate the niche indices. More importantly, explain why these two indices are necessary. For instance, the `levins.Bn` function in the `MicroNiche` package requires you to define “environments.” What environments were compared here?

If you treated each sample as a unique environment (as implied), I suggest reconsidering the use of the Levin’s index. If your “abundant taxa” are defined by both high abundance and widespread occurrence, the Levin’s index based on sample occurrence may be redundant. If your goal is to show that abundant taxa occupy broader niches, consider using your ecosystem function measurements instead. For example, you could cluster samples into distinct “environmental” types, then compute the niche breadth based on that clustering. A similar logic likely applies to your tolerance index—please clarify its computation and rationale.

>> We thank you for pointing out this important issue. In our previous version, we treated each sample as a distinct environment for calculating Levin’s index and computed tolerance index based on restoration duration gradient, which may lead to inevitable results due to our definitions of abundant, intermediate, and rare taxa. Following your suggestions, we have now recalculated both niche breadth indices. Here we defined 17 measured ecosystem functions as environmental variables. To calculate Levin’s index, we first clustered samples into different environmental types using the K-means clustering algorithm, then evaluated the distribution of ASVs across these environmental types using the `levins.Bn` function in the `MicroNiche` package. For the tolerance index, we first extracted the main characteristics of multiple environmental variables using principal component analysis, followed by assessment of the distribution range of ASVs and the variation in their relative abundances along the environmental gradient using the `ade4` package. We employed these two indices to jointly demonstrate the differences in niche breadth among abundant, intermediate, and rare taxa. Please see Lines 528–534. The corresponding code has been updated in the public repository `figshare`. The new results were consistent with the previous version, demonstrating that abundant taxa occupy broader niches compared to rare taxa. Please see Figure 1.

Lines 528–534 “To evaluate the niche breadth of each bacterial ASV, we treated 17 measured ecosystem functions as environmental variables and calculated two complementary metrics: Levins’ index and tolerance index. The Levins’ index estimates niche breadth in terms of the distribution of ASVs across different environment conditions using the R package “MicroNiche”⁸⁵, whereas the tolerance index account for both the distribution range of ASVs and the variation in their relative abundances along environmental gradients (i.e., environmental tolerance), using the R package “ade4”⁸⁶.”

Figure 1. Definition and composition of abundant, intermediate, and rare taxa. **a** Cutoff values for defining abundant, intermediate, and rare taxa based on the average relative abundance across all samples. **b, c** Number of ASVs and relative abundance for the three biological groups across all samples. **d** The proportion of samples occupied for each of abundant, intermediate, and rare ASVs. **e** Taxonomic distribution of the three biological groups at phylum level. The thickness of each ribbon in the circos plot represents the relative abundance of the three biological groups assigned to different phyla. **f, g** Boxplots of the niche breadth for each ASV within abundant, intermediate, and rare taxa estimated using the Levins’ index and the tolerance index, respectively. The different letters above the boxes represent significant differences determined by the Kruskal–Wallis test followed by Dunn post-hoc test (Bonferroni correction).

References:

85. Feinsinger, P., Spears, E. E. & Poole, R. W. A simple measure of niche breadth. *Ecology* **62**, 27–32 (1981).

86. Dolédec, S., Chessel, D. & Gimaret-Carpentier, C. Niche separation in community analysis: a new method. *Ecology* **81**, 2914–2927 (2000).

Lines 604–606: The combined use of regression and Kruskal–Wallis tests is unclear. Are you aiming to predict temporal trends or assess differences between time points? These methods serve distinct purposes. I recommend emphasizing the temporal trend. Consider fitting and comparing different regression models (e.g., linear, quadratic, logarithmic, spline-based) and reporting R² or AIC values to support your choice of model.

>> Thanks. We are sorry that the previous description is not clear and confusing you. Regression analyses were employed to predict temporal trends of ASV richness for abundant, intermediate, and rare taxa. Following your suggestion, we fitted linear and nonlinear (including quadratic, logarithmic and restoration model) regressions and provided their AIC values to support our selection of the model. The restoration model we employed was developed by Poorter et al. (2021, Science) and effectively captures the typical restoration trajectory of ecological attributes, exhibiting an initial increase followed by stabilization. The temporal patterns of ASV richness of abundant and intermediate taxa were better described by this restoration model, whereas the rare taxa exhibited a linear increase with time. Please see Lines 537–542 and Supplementary Table 1. Additionally, here the Kruskal–Wallis test was used solely to assess the differences in ASV richness (including all samples) among abundant, intermediate, and rare taxa, and the differences among different time points were not considered.

Lines 537–542 “*We evaluated the responses of ASV richness of abundant, intermediate, and rare taxa across the restoration duration by fitting linear and nonlinear (quadratic, logarithmic and restoration model) regression models, and selected the optimal model for each dataset based on lowest AIC value. The restoration model developed by Poorter et al.⁸⁷ can capture the typical restoration trajectory of ecological attributes, exhibiting an initial increase followed by stabilization.*”

Supplementary Table 1. Best models for richness of abundant, intermediate and rare taxa.

Model	Abundant		Intermediate		Rare	
	r ²	AIC	r ²	AIC	r ²	AIC
Linear	0.08	623	0.19***	735	0.15**	752
Quadratic	0.49***	592	0.52***	709	0.13*	754

Logarithmic	0.41***	599	0.40***	718	0.04	758
Restoration model	0.62***	578	0.54***	707	-	-

Significant P values are represented by *** when $P < 0.001$, ** when $P < 0.01$ and * when $P < 0.05$. Lower AIC values indicate a better fit of the model.

References:

87. Poorter, L. et al. Multidimensional tropical forest recovery. *Science* **374**, 1370–1376 (2021).

Line 647: While I'm relatively new to network analysis and Gephi, I would appreciate a clearer ecological interpretation of the topological properties you report. For instance, overall network metrics such as centralization of betweenness (rather than individual node values) have been used to infer microbial community stability—see Long et al. (2025, *Global Change Biology*, Fig. 1c) and Chen et al. (2024, *Communications Earth & Environment*, Fig. 1b). These may provide more meaningful contrasts than node-level metrics. Otherwise, please explain the ecological significance of each metric more clearly in the Methods section.

(Upon further reading, I noticed similar graphs are included in the supplementary material. In that case, my revised suggestion is to add more explanation in the Methods section regarding why these specific network metrics were chosen and what ecological insights they are intended to provide.)

>> Thank you for pointing out this important issue. We acknowledge the limitations of co-occurrence network, yet they remain valuable for understanding microbial community structure and potential interspecies connections. Our goal was to elucidate the roles and importance of abundant, intermediate, and rare taxa by calculating key node-level topological properties (e.g., degree, closeness centrality, betweenness centrality and clustering coefficient), which are widely used and potentially linked to community stability and function. We have provided more explicit ecological explanations for these topological properties in the Methods. Please see Lines 583–602. We further distinguished the keystone ASVs in the network based on the within-module connectivity (Z_i) and among-module connectivity (P_i), with the expectation of providing more definite evidence for the differences in topological roles among abundant, intermediate, and rare taxa. Please see Lines 605–608, Lines 180–183 and Supplementary Figure 8. While network-level metrics can infer microbial community structure and stability, they offer limited resolution for ASV-level comparisons in our study. Notably, certain network-

level metrics are derived from node-level properties (such as averages), underscoring their analytical interdependence. We therefore prioritized node-level properties in this study.

Lines 583–602 “*We calculated the key topological properties for each node, including degree, closeness centrality, betweenness centrality and clustering coefficient, using the R package “igraph”. These metrics are widely employed in microbial network analysis to assess the importance of nodes, and are potentially associated with community stability and function^{90–92}. Degree refers to the number of edges connected to a node. Nodes with high degree co-vary with many other nodes, often acting as key drivers of community function, resource distribution, and resilience. Closeness centrality measures the proximity of a node to all other nodes in a network via the average shortest path length, reflecting the speed and efficiency with which a node can propagate changes or respond across the network. Betweenness centrality quantifies the number of shortest paths passing through a node. Nodes with high betweenness centrality can effectively regulate the flow of information or metabolites, and are crucial for maintaining network stability. Clustering coefficient measures the degree to which a node’s neighboring nodes are interconnected, calculated as the ratio of the actual to all possible connections among adjacent nodes. A higher clustering coefficient indicates that neighboring nodes are more tightly interconnected, suggesting localized clustering. Thus, nodes with higher degree, closeness centrality, betweenness centrality and clustering coefficient are regarded as key hubs for information transfer and connectivity, playing a crucial role in maintaining the complexity and functionality of the network^{65,92,93}.”*

Lines 605–608 “*The keystone ASVs were further determined based on the within-module connectivity (Z_i) and among-module connectivity (P_i), including network hubs ($Z_i > 2.5$ and $P_i > 0.62$), module hubs ($Z_i > 2.5$ and $P_i \leq 0.62$), and connectors ($Z_i \leq 2.5$ and $P_i > 0.62$)^{94,95}.”*

Lines 180–183 “*A total of 97 ASVs were identified as keystone nodes in the network, comprising 4 module hubs (including 3 abundant and 1 intermediate ASVs) and 93 connectors (with 8 abundant, 44 intermediate, and 41 rare ASVs) (Supplementary Fig. 8).”*

Supplementary Figure 8. Topological roles of nodes determined by their within-module connectivity and among-module connectivity. Each dot represents a node. The module hubs, connectors and network hubs are regarded as keystone nodes.

References:

65. Banerjee, S., Schlaeppi, K. & van der Heijden, M. G. A. Keystone taxa as drivers of microbiome structure and functioning. *Nat. Rev. Microbiol.* **16**, 567–576 (2018).
90. Zhang, Y. et al. Vertical diversity and association pattern of total, abundant and rare microbial communities in deep-sea sediments. *Mol. Ecol.* **30**, 2800–2816 (2021).
91. Xue, Y. et al. Distinct patterns and processes of abundant and rare eukaryotic plankton communities following a reservoir cyanobacterial bloom. *ISME J.* **12**, 2263–2277 (2018).
92. Jiao, S., Wang, J., Wei, G., Chen, W. & Lu, Y. Dominant role of abundant rather than rare bacterial taxa in maintaining agro-soil microbiomes under environmental disturbances. *Chemosphere* **235**, 248–259 (2019).
93. Newman, M. E. J. The Structure and Function of Complex Networks. *SIAM Rev.* **45**, 167–256 (2003).
94. Guimerà, R. & Nunes Amaral, L. A. Functional cartography of complex metabolic networks.

Nature **433**, 895–900 (2005).

95. Olesen, J. M., Bascompte, J., Dupont, Y. L. & Jordano, P. The modularity of pollination networks. *Proc. Natl. Acad. Sci. U.S.A* **104**, 19891–19896 (2007).

Line 668 and final analysis: Ending with a simple correlation analysis feels somewhat underwhelming given the depth of your earlier explorations. I encourage you to develop a hypothesis-driven (hypotheses built on your results above) structural equation model (SEM) to synthesize your findings and strengthen the paper's impact. A good example is provided by Chen et al. (2024, *Communications Earth & Environment*), which effectively ties together multiple ecological layers using SEM.

For instance, your SEM could include restoration age and soil chemical properties as input variables, microbial attributes (e.g., richness of rare taxa, network properties of abundant taxa) as mediating variables, and ecosystem functions (e.g., multifunctionality, enzyme activity, or plant growth) as outcome variables. Your earlier results suggest that richness of rare taxa and the average degree of abundant taxa may be key mediators, while intermediate taxa could be indirectly linked through soil chemistry. I understand this may require additional work, but I believe this approach would align well with your paper's title on "Divergent multifunctionality contributions" and make your study valuable to a broader ecological and restoration audience.

>> Thank you for your valuable suggestion. We have now constructed a structural equation model to infer the hypothesized direct and indirect relationships between restoration duration, plant cover, soil pH, bacterial ASV richness, subnetwork degree, and ecosystem multifunctionality. To prevent overfitting, before modeling, the richness and degree of abundant, intermediate, and rare taxa were each consolidated into a composite variable using linear regression models, with standardized coefficients serving as their weights. We then employed the chi-square test (χ^2), comparative fit index (CFI) and Fisher's C test to evaluate the goodness-of-fit statistics of the model. Unlike the referenced example that focused on network-level properties, our study specifically evaluated node-level properties. To ensure data compatibility, we used the average node degree of abundant, intermediate, and rare taxa within subnetworks at each restoration stage in the SEM, without involving the co-occurrence network across samples. Our results indicated that restoration duration was positively associated with multifunctionality both directly and indirectly, mediated by increasing plant cover and decreasing soil pH. The abundant and intermediate taxa had positive weights while the rare taxa

had a negative weight for the composite richness. Increased plant cover significantly enhanced bacterial ASV richness, particularly for abundant and intermediate taxa. Notably, bacterial ASV richness exhibited a direct and positive association with multifunctionality, while the relationship between average degree and multifunctionality was not significant. Please see Lines 618–628, Lines 203–212 and Figure 6b.

Lines 618–628 “*Subsequently, a structural equation model (SEM) was constructed to infer the hypothesized direct and indirect relationships between restoration duration, plant cover, soil pH, bacterial ASV richness, network properties (i.e., average degree in subnetworks at each restoration stage), and multifunctionality across samples. To avoid model overfitting, ASV richness and degree of abundant, intermediate, and rare taxa were each consolidated into a composite variable using linear regression models prior to SEM construction, using standardized coefficients serving as their weights. The abundant and intermediate taxa had positive weights while the rare taxa had a negative weight for the composite richness. We then employed the chi-square test (χ^2), comparative fit index (CFI) and Fisher's C test to evaluate the goodness-of-fit statistics of the model.*”

Lines 203–212 “*To complement these bivariate correlation analyses, we employed a structural equation model (SEM) to infer the hypothesized direct and indirect pathways connecting restoration duration, plant cover, soil pH, bacterial ASV richness, average degree and multifunctionality (Fig. 6b). Our results revealed that restoration duration was positively associated with multifunctionality both directly and indirectly, mediated through enhanced plant cover and reduced soil pH. Increased plant cover significantly augmented bacterial ASV richness, particularly for abundant and intermediate taxa. Notably, bacterial ASV richness maintained a direct and positive association with multifunctionality, while the relationship between average degree and multifunctionality was not statistically significant.*”

Figure 6. Relationships of abundant, intermediate, and rare taxa richness with ecosystem multifunctionality. **a** Relationships between multifunctionality and richness of abundant, intermediate, and rare taxa. The black fitted lines are from linear regression. The solid and dotted lines represent statistically significant ($P \leq 0.05$) and nonsignificant ($P > 0.05$) relationships, respectively. **b** Structural equation model depicting the hypothesized direct and indirect relationships among restoration duration, plant cover, soil pH, bacterial ASV richness, average degree and multifunctionality. The richness and degree of abundant, intermediate, and rare taxa were each consolidated into a composite variable using the standardized coefficients (as shown in the values below, representing their weights) from a linear regression model. Numbers adjacent to arrows (path coefficients) represent standardized effect sizes. Significant P values are represented by *** when $P < 0.001$, ** when $P < 0.01$ and * when $P < 0.05$. Arrow width is proportional to the magnitude of standardized path coefficients. Red and blue arrows represent positive and negative relationships, respectively, with solid and dashed arrows denote statistically significant and non-significant relationships, respectively. R^2 is the proportion of variance explained by the model. Goodness-of-fit of SEM are evaluated by Chi-square test, CFI and Fisher's C test. Time, restoration duration.

Results and Discussion Review Comments

L141–165: Consider moving this section to the supplementary materials if you are constrained by word limits. While I’ve repeatedly asked for more details on how you define abundant, intermediate, and rare microbes, the results here are not central to your main narrative.

>> Thank you for your approval of our previous revisions. Should it be necessary, we will move this section to the supplementary materials.

L166–167: The trend description is ambiguous. For abundant or intermediate taxa, the increase from year 0 to 10 is evident, but I wouldn’t characterize the subsequent trend as a nonlinear increase. Please rephrase for clarity.

>> Thank you for highlighting this issue. This sentence has been revised for improved clarity. Please see Lines 130–133.

Lines 130–133 *“ASV richness of both abundant and intermediate taxa exhibited marked increases with restoration duration, reaching asymptotic stability after approximately 15 years (Fig. 2a, b; Supplementary Table 1), whereas rare taxa richness showed a sustained, linear increase over time (Fig. 2c; Supplementary Table 1).”*

L168–170: This sentence is hard to follow and appears internally contradictory. If it doesn’t contribute directly to your main points, consider removing it. Alternatively, a clearer version might be: “Although rare taxa include more ASVs across all samples, intermediate ASVs contribute greater richness at the sample level.” Also, avoid the term “alpha-diversity” unless you’re using a specific index—just state the name (e.g., richness) for precision.

>> Thanks for your suggestions and sorry for the inappropriate expression. The sentence has been revised based on your suggestions. Please see Lines 133–135. For clarity, we have replaced “ α -diversity” with “richness” throughout the manuscript.

Lines 133–135 *“Although rare taxa included more ASVs across all samples, intermediate ASVs contributed greater richness at sample level ($P < 0.001$; Supplementary Fig. 5a).”*

L173: Comparing contributions to community dissimilarity among the three groups is

interesting, but I question the logic of comparing 1 vs. 2, 2 vs. 3, etc. Wouldn't comparisons like 0 vs. 1, 0 vs. 2, ..., 0 vs. 53 be more ecologically informative? That way, you can directly discuss which groups recover early and which remain unrecovered after 53 years.

>> We thank you for noticing this insufficient analysis. Our recently published paper has demonstrated that plant and soil microbial communities progressively approximate natural communities (i.e., the reference state) since the establishment of SCBs, with identifiable temporal thresholds for community transitions. However, even after 53 years of restoration, these communities remain significantly distinct from their corresponding natural communities. These main findings have been mentioned in the Methods. Please see the response above. The temporal variations among different taxa have also been discussed. These results collectively underscore the inherent complexity and prolonged duration of biological community restoration processes. This study focuses on the restoration chronosequence and does not include natural communities, as the classification of abundant and rare taxa in restored and natural plots may be an important concern for us. The samples with 0 years of restoration represent the mobile sand (i.e., the initial unrestored state), rather than serving as reference natural communities. Here, we performed the similarity percentage (SIMPER) analysis based on Bray–Curtis distance to quantify and compare the contributions of abundant, intermediate, and rare taxa to community dissimilarity. In the previous version, we only examined sequential temporal comparisons between adjacent restoration stages. To address this limitation, we have now implemented comprehensive pairwise comparisons across all restoration stages, while quantitatively assessing the relative contributions of abundant, intermediate, and rare taxa to community dissimilarity. Please see Supplementary Figure 5b and Supplementary Table 2. The results emphasized that abundant and intermediate taxa contribute more significantly to community composition differences compared to rare taxa.

Supplementary Figure 5. ASV richness and cumulative contributions to community differences in abundant, intermediate, and rare taxa. **a** Boxplots of the ASV richness of abundant, intermediate, and rare taxa. The different letters represent significant differences determined by the Kruskal–Wallis test followed by Dunn post-hoc test (Bonferroni correction). **b** Boxplots of the cumulative contributions of abundant, intermediate, and rare taxa to pairwise composition differences using SIMPER analysis based on Bray–Curtis distance. The different letters represent significant differences determined by the Kruskal–Wallis test followed by Dunn post-hoc test (Bonferroni correction). The specific values were presented in Supplementary Table 2.

Supplementary Table 2. Cumulative contributions of abundant, intermediate, and rare taxa to pairwise composition differences based on Bray–Curtis distance.

Taxa	Time	0 years	1 year	3 years	4 years	5 years	6 years	11 years	20 years	23 years	26 years
Abundant	1 year	46.58									
	3 years	50.22	51.28								
	4 years	40.67	51.46	57.23							
	5 years	44.24	48.93	55.89	47.12						
	6 years	39.46	47.34	54.58	42.63	43.75					
	11 years	42.06	50.10	57.55	47.31	46.42	43.41				
	20 years	38.50	49.42	55.44	46.76	46.16	43.36	37.45			
	23 years	41.25	51.76	57.81	48.72	48.05	45.47	39.33	29.80		

	26 years	41.65	52.40	58.38	49.53	48.98	46.09	39.57	29.34	31.18	
	53 years	39.97	50.51	56.04	46.76	46.85	44.30	39.35	33.37	36.41	35.98
	1 year	37.22									
	3 years	34.32	38.17								
	4 years	39.21	37.45	32.57							
	5 years	38.03	40.24	34.45	39.16						
Intermediate	6 years	40.64	40.54	34.63	41.45	41.65					
	11 years	39.52	38.59	32.79	39.10	40.21	41.74				
	20 years	41.98	38.32	33.56	39.28	40.39	41.73	44.43			
	23 years	41.55	38.43	33.58	39.64	40.90	42.01	44.74	47.90		
	26 years	41.24	37.86	33.07	38.95	40.06	41.46	44.66	48.27	49.46	
	53 years	40.06	36.48	32.04	38.14	38.60	39.73	42.23	44.37	44.73	44.90
	1 year	16.20									
	3 years	15.46	10.55								
	4 years	20.12	11.10	10.21							
	5 years	17.72	10.83	9.66	13.72						
Rare	6 years	19.91	12.12	10.79	15.92	14.60					
	11 years	18.42	11.31	9.67	13.58	13.37	14.85				
	20 years	19.52	12.26	11.00	13.96	13.46	14.92	18.12			
	23 years	17.20	9.80	8.60	11.63	11.05	12.53	15.93	22.30		
	26 years	17.11	9.74	8.55	11.51	10.96	12.45	15.76	22.39	19.35	
	53 years	19.97	13.01	11.92	15.10	14.55	15.97	18.42	22.26	18.86	19.12

L175: Include the relevant statistics here or in the figure to substantiate the claim of a significant difference.

>> Thanks. We have used the Kruskal-Wallis test with a Dunn post-hoc test (Bonferroni correction) to assess the differences in the relative contributions of abundant, intermediate, and rare taxa to community composition differences. Please see Supplementary Figure 5b above.

L178: Consider using “longer” instead of “greater” for temporal descriptions.

>> Thanks. This sentence has now been revised. Please see Lines 142–144.

Lines 142–144 “Moreover, the composition differences of abundant, intermediate, and rare subcommunities increased significantly with longer restoration duration interval (Fig. 3a–c).”

L179: It’s unclear how the triangle plot demonstrates temporal variability, and the figure reference may be incorrect. Also, please explain how the triangle plot is constructed—is it based on total beta diversity or some partitioned component?

>> We thank you for pointing out this vague expression. The triangle plots specifically visualize the two principal components of β -diversity for abundant, intermediate, and rare taxa, rather than reflecting their temporal dynamics. Each axis of the triangle plot represents a distinct aspect of community composition: one axis corresponds to community similarity, while the other two axes quantify the two components of β -diversity (i.e., species turnover and richness difference). The plotted points (projected positions along the axes) represent the triples of average values of similarity, turnover, and richness difference matrices, with the sum of these three average values equaling 1. Our results indicated that species turnover dominates the community variations for abundant, intermediate, and rare taxa. We have revised the figure title and further provided more details and the specific values of triples of abundant, intermediate, and rare taxa in the figure legend. Please see Figure 3.

Figure 3. Temporal variations in β -diversity and the visualization of its compositional components. **a–c** Relationships between restoration duration dissimilarity and β -diversity of abundant, intermediate, and rare subcommunities based on the Mantel test. The black fitted lines are from linear regression. Significant P values are represented by *** when $P < 0.001$, ** when $P < 0.01$ and * when $P < 0.05$. **d–f** Triangular plots of the components of β -diversity of abundant, intermediate, and rare subcommunities. The three axes represent compositional similarity and the two components of β -diversity (i.e., species turnover and richness difference) based on Bray–Curtis dissimilarity. Each point represents a triplet of average values from the corresponding similarity, turnover and richness difference matrices, constrained such that their sum equals 1. The average similarity, species turnover, and richness difference were 50%, 46%, and 4% for abundant subcommunities; 18%, 80%, and 1% for intermediate subcommunities; and 2%, 98%, and 0% for rare subcommunities.

L184: The current heatmap visualization is ineffective. Please revise the color scheme to better highlight relative abundance differences.

>> Thanks for your suggestion. We have now revise the color scheme of the heatmap for clarity. Please see Supplementary Figure 6.

Supplementary Figure 6. Heatmaps of the relative abundance of abundant, intermediate, and rare ASVs. **a–c** The arrangement of relative abundance for each abundant, intermediate, and rare ASV along the 53-year restoration chronosequence. The gradient of red color intensity represents the relative abundance levels, with darker shades indicating higher abundance and

lighter shades denoting lower abundance.

L204–205: Avoid referring only to “abundant and rare subcommunities” here. Omitting “intermediate” can be misleading and imply exclusion from your analysis.

>> We thank you for pointing out this issue. The sentence has now been corrected in the revised manuscript. Please see Lines 164–166.

Lines 164–166 *“On the other hand, the temporal variations in the assembly processes of abundant, intermediate and rare subcommunities were further delineated.”*

L223: While positive correlations are often interpreted as synergistic interactions, that assumption generally applies within a shared ecological context. In your case, microbes from year 0 and year 53 are unlikely to interact directly. The observed correlations likely reflect shared responses to environmental gradients rather than true biotic interactions. Given your study design, it would be more informative to construct subnetworks for each restoration stage, rather than categorizing nodes by abundance. Supplementary Figures 8 and 9 are more insightful and might be better suited for inclusion in the main text.

>> We thank you for pointing out this, and sorry for the inappropriate expression. We fully agree that the correlations in the co-occurrence network reflect the shared responses of the abundance of strains to environmental gradients. We conducted statistical analysis on the sequencing data to identify significant correlations in the abundance of strains across samples. In the resulting correlation network, nodes represent strains (not those that only appeared at certain time points, such as 0 years and 53 years), and edges indicate correlations that reflect parallel changes in the abundance of strains across samples or time points, rather than actual ecological interactions between strains. Positive correlations in the abundance of strains can simply be due to shared environmental preferences among species or indirect interactions within the community. Despite these limitations, microbial association networks remain valuable for generating hypotheses about ecological interactions. Calculating topological properties and identifying keystone nodes can provide critical insights into the structural organization of microbial communities. In the previous version, our description of the co-occurrence networks was not sufficiently clear. For clarity and rigorous, we have now further emphasized the above information in the Methods and revised the wording throughout the

manuscript, using terms such as “shared/coordinated responses” or “parallel changes” or similar to describe the results of the network analysis.

Additionally, constructing subnetworks for each restoration stages may provide more information about temporal changes in the network, but this has limited relevance to our primary objective of investigating differences among microbial taxa categorized by abundance. In this study, we still aim to evaluate the topological properties and roles of ASVs/nodes in the correlation network across all samples (Figure 5) to reveal the co-occurrence patterns and functional contributions of abundant, intermediate, and rare taxa, while subnetworks serve as supplementary figures to provide additional insights. In fact, another ongoing study based on this experimental design primarily focuses on the temporal dynamics of soil microbial networks (including both bacteria and fungi) and their impact on ecosystem multifunctionality by extracting subnetworks for each sample. Multiple network-level properties were assessed over time, such as network complexity, stability, bacteria-fungi links, and the proportion of positive correlations. We also employed multiple regression and structural equation model to evaluate the direct and indirect effects of restoration duration, microbial diversity, and network properties on ecosystem multifunctionality. We sincerely hope to receive your expert guidance in our future work, which would significantly enhance the quality of our findings.

L239: With the richness of your dataset, I strongly encourage replacing the final section with a structural equation model (SEM). For example: What are the known effects of SCB (your restoration method) from prior literature? Does it increase soil carbon or vegetation productivity? SEM would allow you to test whether restoration time improves microbial indices indirectly via changes in soil properties or vegetation, and which microbial features best explain ecological functions such as decomposition.

>> Thank you for raising this point again. Our restoration method (SCBs) is a combination of fixing shifting sand and adding organic material (straw) simultaneously to expedite vegetation and soil restoration. As noted in our responses above, we employed a structural equation model (SEM) to examine the direct and indirect relationships among restoration duration, plant cover, soil pH, bacterial ASV richness, subnetwork degree, and ecosystem multifunctionality (please see Figure 6b). The results Our results indicated that restoration duration was positively associated with multifunctionality both directly and indirectly, mediated by increasing plant cover and decreasing soil pH. Increased plant cover significantly enhanced bacterial ASV

richness, particularly for abundant and intermediate taxa. Bacterial ASV richness exhibited a direct and positive association with multifunctionality, while the relationship between average degree and multifunctionality was not significant. Although the SEM provides clear and comprehensive insights compared to simple correlation analyses (e.g., relationships between richness and individual function/multifunctionality), we believe that ASV-level analyses should still be retained, as they also provide critical information (e.g., Figures 5c–e and 7).

L285–286: The claim “The increase can potentially be attributed to the improvement of vegetation and soil conditions” is exactly the type of hypothesis that SEM could test more rigorously.

>> We fully agree with you. The SEM has provided clearer evidence demonstrating the facilitative effects of vegetation (cover) and soil (pH) conditions (please see Figure 6b). Since most plant and soil attributes in this study were defined as ecosystem functions, they did not appear as environmental variables in the SEM. Once again, we sincerely appreciate your critical suggestion to employ SEM in our study.

L294: Similarly, the idea that patterns are “driven primarily by species-specific ecological niches across restoration stages” can also be empirically evaluated using SEM. It could help confirm whether different ecological processes govern the assembly of the three subcommunities.

>> The idea that the temporal variation in the composition of rare subcommunities was primarily driven by species-specific ecological niches across restoration stages can be supported by the relevant analyses in our manuscript. (1) Turnover accounted for 98% of the temporal variation in rare subcommunity composition (please see Figure 3f), highlighting the dominance of species/ASV replacement; (2) 70.92% of rare ASVs were detected in only one sample. However, since the SEM was constructed using richness data at the sample level, it remains challenging to provide clear support for patterns in community composition and assembly processes.

L341–342: This statement contradicts your earlier assertion (L309) that abundant taxa, as ecological generalists with broad niches, are shaped mainly by stochastic processes like dispersal limitation. If abundant taxa are insensitive to environmental selection, how can their

increase be associated with increased niche differentiation? Please clarify your position.

>> We thank you for pointing out this, and sorry again for the inappropriate expression. The abundant taxa, as generalists with broad niches, are primarily governed by stochastic processes in their distribution patterns. However, our results demonstrated that variable selection (26.6%) imposed by environmental changes during restoration serves as a secondary yet critical process shaping the community turnover in abundant taxa. This process may have expanded ecological niches to some extent, thereby enhancing the richness of abundant taxa. We have carefully revised the wording and emphasized these pieces of information in the Discussion. Please see Lines 267–270, Lines 277–279 and Lines 287–290.

Lines 267–270 “*Therefore, their distribution patterns are primarily shaped by stochastic processes (e.g., dispersal limitation or ecological drift), as generalists tend to be less influenced by environmental selection^{52–54}.*”

Lines 277–279 “*Thus, variable selection served as a secondary yet critical process orchestrating the compositional turnover of both abundant and rare subcommunities⁴².*”

Lines 287–290 “*In this study, the increased richness of high-abundance taxa may reflect the diversification ecological niches shaped by variable selection with restoration duration, which promotes stability and functionality of ecosystems, thus the richness of high-abundance taxa was positively correlated with ecosystem multifunctionality.*”

References:

42. Jia, X., Dini-Andreote, F. & Salles, J. F. Unravelling the interplay of ecological processes structuring the bacterial rare biosphere. *ISME Commun.* **2**, 96 (2022).
52. Liu, L., Yang, J., Yu, Z. & Wilkinson, D. M. The biogeography of abundant and rare bacterioplankton in the lakes and reservoirs of China. *ISME J.* **9**, 2068–2077 (2015).
53. Xu, Q. et al. Microbial generalists and specialists differently contribute to the community diversity in farmland soils. *J. Adv. Res.* **40**, 17–27 (2022).
54. Hu, W. et al. Continental-scale niche differentiation of dominant topsoil archaea in drylands. *Environ. Microbiol.* **24**, 5483–5497 (2022).

L350–351: The assumption that abundant taxa are necessarily fast-growing or highly

competitive is oversimplified. Fierer (2017) points out substantial variation in traits even among closely located soil microbes. Furthermore, Actinobacteria are often abundant yet are considered stress-tolerant, not necessarily fast-growing or competitive. Please ground your interpretations in trait-based ecology and acknowledge this complexity.

>> Thank you for your suggestion. We have realized that the previous expression was oversimplified and inappropriate. This sentence has now been revised. Our finding can be explained by the observation that while rare subcommunities have greater richness, the high-abundance taxa occupying the majority of niches typically possess higher metabolic versatility, thereby driving variations in multiple ecosystem functions associated with nutrient cycling. A recent study (Peng et al. 2025, *Advanced Science*) supports our perspective, demonstrating that bacterial taxa with broader niche breadth maximize the completeness and diversity of metabolic pathways essential for growth and resource acquisition, including carbon fixation, ATP synthesis, as well as carbohydrate and nitrogen metabolism. Please see Lines 295–302.

Lines 295–302 “*This divergence can be attributed to the fact that, despite greater overall richness, rare subcommunities are less metabolically versatile than high-abundance taxa, which typically dominate niche space and possess broader metabolic repertoires that underpin key nutrient cycling functions^{23,28}. Supporting this interpretation, a recent study demonstrated that bacterial taxa with broader niche breadth maximize the completeness and diversity of metabolic pathways essential for growth and resource acquisition, such as carbon fixation, ATP synthesis, and carbohydrate and nitrogen metabolism⁵⁸.*”

References:

23. Jiao, S. & Lu, Y. Abundant fungi adapt to broader environmental gradients than rare fungi in agricultural fields. *Glob. Chang. Biol.* **26**, 4506–4520 (2020).

28. Rivett, D. W. & Bell, T. Abundance determines the functional role of bacterial phylotypes in complex communities. *Nat. Microbiol.* **3**, 767–772 (2018).

58. Peng, Z. et al. Trait-Based Life History Strategies Shape Bacterial Niche Breadth. *Adv. Sci.* **12**, 2405947 (2025).

L360: The interpretations in this paragraph are problematic:

1. Your data show significant differences in topological properties (e.g., higher degree for

abundant, higher closeness centrality for intermediate, etc.), so the conclusion that they are similar is unsupported.

2. Even if metrics were similar, that doesn't justify the claim that they play equally important roles in maintaining network structure. Concepts like “keystone species” or “modularity” would be more appropriate indicators.

3. Given that the network spans all restoration stages, it's difficult to interpret how node-level metrics (like degree) relate to ecosystem multifunctionality.

4. Finally, the implication that microbes from year 0 and year 53 can interact lacks ecological plausibility.

>> Thanks for your suggestions. We identified keystone nodes in the network based on the within-module connectivity and among-module connectivity (please see response above). Abundant taxa served as module hubs within the network, whereas rare and intermediate taxa exhibited higher closeness centrality, acting as keystone connectors that bridged modules. This suggests distinct ecological roles for these taxa in maintaining network structure and stability. As mentioned above, the observed correlations in the network reflect the shared responses of nodes in abundance during restoration. Nodes do not represent strains present at specific time points (e.g., 0 or 53 years). In this context, higher node-level properties (e.g., degree) indicate that a given node undergoes parallel changes/responds in abundance with more nodes throughout the restoration process. Crucially, we did not simply fit the relationship between degree and ecosystem multifunctionality. Instead, we evaluated the relationships between the node degree and their association strength with multifunctionality at the node level for abundant, intermediate and rare taxa (Fig. 5c–d; each point represents an ASV). The results indicated that the relative abundances of both high-degree abundant ASVs and low-degree rare ASVs exhibited stronger correlations with multifunctionality. Collectively, these findings suggest that bacterial communities may mediate ecosystem multifunctionality through a dual mechanism involving complementary roles of these key taxa. Specifically, highly connected abundant taxa—likely due to shared environmental preferences—functioned as module hubs, exhibiting extensive co-occurrence patterns with other ASVs. These taxa thereby act as key contributors to multiple ecological processes via mass ratio effects, functional integration, and metabolic versatility, enabling comprehensive and efficient resource utilization to maintain high functional performance. Conversely, rare taxa occupy unique niches and perform specific functions, resulting in their independent association with multifunctionality that relies little on

interspecies connections or shared environmental responses. These findings align with previous observations in agricultural ecosystems (Zhang et al. mBio 2022). We have carefully revised this paragraph to ensure our interpretation accurately reflects our results. Please see Lines 310–330.

Lines 310–330 “*Co-occurrence network analysis has been emerged as a powerful tool for deciphering interspecies connections that significantly shape biodiversity patterns and ecosystem functioning^{62–64}. In this study, abundant taxa served as module hubs within the network, whereas rare and intermediate taxa exhibited elevated closeness centrality, serving as keystone connectors that bridged distinct modules. This delineates discrete yet distinct ecological roles for these taxa in maintaining network structure and stability. Although our SEM analysis revealed no significant correlation between the average subnetwork degree and ecosystem multifunctionality, subsequent ASV-level analysis indicated that the relative abundances of both high-degree abundant ASVs and low-degree rare ASVs exhibited stronger correlations with multifunctionality. Collectively, these insights point toward a dual mechanistic framework through which bacterial communities regulate ecosystem multifunctionality via complementary roles of these key taxa. Specifically, highly connected abundant taxa presumably due to shared environmental preferences functioned as module hubs, exhibiting extensive co-occurrence patterns with other ASVs. These taxa thereby act as key contributors to multiple ecological processes via mass ratio effects, functional integration, and metabolic versatility, enabling comprehensive and efficient resource utilization to maintain high functional performance²⁵. Conversely, rare taxa occupied unique niches and perform specific functions, resulting in their independent association with multifunctionality that relies little on interspecies connections or shared environmental responses^{25,37}.”*

References:

25. Zhang, Z., Lu, Y., Wei, G. & Jiao, S. Rare Species-Driven Diversity–Ecosystem Multifunctionality Relationships are Promoted by Stochastic Community Assembly. *mBio* 13, e00449–00422 (2022).
37. Pester, M., Bittner, N., Deevong, P., Wagner, M. & Loy, A. A ‘rare biosphere’ microorganism contributes to sulfate reduction in a peatland. *ISME J.* 4, 1591–1602 (2010).
62. Barberán, A., Bates, S. T., Casamayor, E. O. & Fierer, N. Using network analysis to explore co-occurrence patterns in soil microbial communities. *ISME J.* 6, 343–351 (2012).

63. Blanchet, F. G., Cazelles, K. & Gravel, D. Co-occurrence is not evidence of ecological interactions. *Ecol. Lett.* **23**, 1050–1063 (2020).

64. Oña, L., Shreekar, S. K. & Kost, C. Disentangling microbial interaction networks. *Trends Microbiol.* **33**, 619–634 (2025).

L379: Acknowledging limitations is useful when you have a clear take-home message. It's meant to help readers understand how broadly your conclusions apply. But in your case, I've raised concerns about your experimental design and whether your analyses are the best way to support your story. So just adding a "limitations" section won't really fix those deeper issues.

>> We sincerely appreciate your rigorous review, which has prompted us to thoroughly re-examine our analytical framework. We have carefully revised the entire manuscript in accordance with your detailed suggestions (please refer to the responses above). In this section, we have streamlined the content and primarily emphasized the limitations of the network analysis and community assembly. Please see Lines 344–359. Should further refinements be needed, we would be pleased to address them promptly. Thank you again for your valuable contributions to strengthening this work.

Lines 344–359 *"Despite the valuable insights provided by our study into the distinct characteristics of abundant and rare taxa, some limitations warrant attention and should be addressed in future research. For instance, microbial co-occurrence networks reflect common selection patterns driven by shared environmental gradients, rather than providing conclusive evidence of actual ecological interactions^{63,66}. In light of this, further studies should utilize cutting-edge omics technologies, and consider integrating co-occurrence and metabolic networks to infer more robust interactions among microbial species and elucidate the potential underlying ecological mechanisms^{67,68}. Additionally, the null-modeling-based quantitative framework used to assess the relative contributions of microbial community assembly processes is inherently relies on assumptions regarding phylogenetic signals, as well as the abundance and distribution of individuals within a community^{69,70}. Applying this framework to different subcommunities in our study may introduce mathematical biases, potentially obscuring nuanced inter-group interactions⁵⁶. Despite these methodological caveats, our results still provide a robust empirical foundation for the understanding how assembly processes differentially structure microbial subcommunities during ecological restoration."*

References:

56. Jia, X., Dini-Andreote, F. & Falcão Salles, J. Community assembly processes of the microbial rare biosphere. *Trends Microbiol.* **26**, 738–747 (2018).
63. Blanchet, F. G., Cazelles, K. & Gravel, D. Co-occurrence is not evidence of ecological interactions. *Ecol. Lett.* **23**, 1050–1063 (2020).
66. Faust, K. Open challenges for microbial network construction and analysis. *ISME J.* **15**, 3111–3118 (2021).
67. Peng, X. et al. Metabolic interdependencies in thermophilic communities are revealed using co-occurrence and complementarity networks. *Nat. Commun.* **15**, 8166 (2024).
68. Cardona, C., Weisenhorn, P., Henry, C. & Gilbert, J. A. Network-based metabolic analysis and microbial community modeling. *Curr. Opin. Microbiol.* **31**, 124–131 (2016).
69. Stegen, J. C. et al. Quantifying community assembly processes and identifying features that impose them. *ISME J.* **7**, 2069–2079 (2013).
70. Stegen, J. C., Lin, X., Konopka, A. E. & Fredrickson, J. K. Stochastic and deterministic assembly processes in subsurface microbial communities. *ISME J.* **6**, 1653–1664 (2012).